# Implicit Regularization in Deep Learning May Not Be Explainable by Norms

**Noam Razin**
Tel Aviv University
noam.razin@cs.tau.ac.il

**Nadav Cohen**
Tel Aviv University
cohennadav@cs.tau.ac.il

## Abstract

Mathematically characterizing the implicit regularization induced by gradient-based optimization is a longstanding pursuit in the theory of deep learning. A widespread hope is that a characterization based on minimization of norms may apply, and a standard test-bed for studying this prospect is matrix factorization (matrix completion via linear neural networks). It is an open question whether norms can explain the implicit regularization in matrix factorization. The current paper resolves this open question in the negative, by proving that there exist natural matrix factorization problems on which the implicit regularization drives *all* norms (and quasi-norms) *towards infinity*. Our results suggest that, rather than perceiving the implicit regularization via norms, a potentially more useful interpretation is minimization of rank. We demonstrate empirically that this interpretation extends to a certain class of non-linear neural networks, and hypothesize that it may be key to explaining generalization in deep learning.[1]

## 1 Introduction

A central mystery in deep learning is the ability of neural networks to generalize when having far more learnable parameters than training examples. This generalization takes place even in the absence of any explicit regularization (see [88]), thus a view by which gradient-based optimization induces an *implicit regularization* has arisen (see, *e.g.*, [64]). Mathematically characterizing this implicit regularization is regarded as a major open problem in the theory of deep learning (*cf.* [66]). A widespread hope (initially articulated in [65]) is that a characterization based on *minimization of norms* (or quasi-norms[2]) may apply. Namely, it is known that for linear regression, gradient-based optimization converges to solution with minimal $\ell_2$ norm (see for example Section 5 in [88]), and the hope is that this result can carry over to neural networks if we allow $\ell_2$ norm to be replaced by a different (possibly architecture- and optimizer-dependent) norm (or quasi-norm).

A standard test-bed for studying implicit regularization in deep learning is *matrix completion* (*cf.* [34, 8]): given a randomly chosen subset of entries from an unknown matrix $W^*$, the task is to recover the unseen entries. This may be viewed as a prediction problem, where each entry in $W^*$ stands for a data point: observed entries constitute the training set, and the average reconstruction error over the unobserved entries is the test error, quantifying generalization. Fitting the observed entries is obviously an underdetermined problem with multiple solutions. However, an extensive body of work (see [26] for a survey) has shown that if $W^*$ is low-rank, certain technical assumptions (*e.g.* "incoherence") are satisfied and sufficiently many entries are observed, then various algorithms can achieve approximate or even exact recovery. Of these, a well-known method based upon convex optimization finds the minimal nuclear norm matrix among those fitting observations (see [15]).

One may try to solve matrix completion using shallow neural networks. A natural approach, *matrix factorization*, boils down to parameterizing the solution as a product of two matrices — $W = W_2 W_1$ — and optimizing the resulting (non-convex) objective for fitting observations. Formally, this can be viewed as training a depth 2 linear neural network. It is possible to explicitly constrain the rank of the produced solution by limiting the shared dimension of $W_1$ and $W_2$. However, Gunasekar *et al.* have shown in [34] that in practice, even when the rank is unconstrained, running gradient descent with small learning rate (step size) and initialization close to the origin (zero) tends to produce low-rank solutions, and thus allows accurate recovery if $W^*$ is low-rank. Accordingly, they conjectured that the implicit regularization in matrix factorization boils down to minimization of nuclear norm:

**Conjecture 1** (from [34], informally stated). *With small enough learning rate and initialization close enough to the origin, gradient descent on a full-dimensional matrix factorization converges to a minimal nuclear norm solution.*

In a subsequent work — [8] — Arora *et al.* considered *deep matrix factorization*, obtained by adding depth to the setting studied in [34]. Namely, they considered solving matrix completion by training a depth $L$ linear neural network, *i.e.* by running gradient descent on the parameterization $W = W_L W_{L-1} \cdots W_1$, with $L \in \mathbb{N}$ arbitrary (and the dimensions of $\{W_l\}_{l=1}^{L}$ set such that rank is unconstrained). It was empirically shown that deeper matrix factorizations (larger $L$) yield more accurate recovery when $W^*$ is low-rank. Moreover, it was conjectured that the implicit regularization, for any depth $L \geq 2$, can *not* be described as minimization of a mathematical norm (or quasi-norm):

**Conjecture 2** (based on [8], informally stated). *Given a (shallow or deep) matrix factorization, for any norm (or quasi-norm) $\|\cdot\|$, there exists a set of observed entries with which small learning rate and initialization close to the origin can* not *ensure convergence of gradient descent to a minimal (in terms of $\|\cdot\|$) solution.*

Conjectures 1 and 2 contrast each other, and more broadly, represent opposing perspectives on the question of whether norms may be able to explain implicit regularization in deep learning. In this paper, we resolve the tension between the two conjectures by affirming the latter. In particular, we prove that there exist natural matrix completion problems where fitting observations via gradient descent on a depth $L \geq 2$ matrix factorization leads — with probability $0.5$ or more over (arbitrarily small) random initialization — *all* norms (and quasi-norms) to *grow towards infinity*, while the rank essentially decreases towards its minimum. This result is in fact stronger than the one suggested by Conjecture 2, in the sense that: *(i)* not only is each norm (or quasi-norm) disqualified by some setting, but there are actually settings that jointly disqualify all norms (and quasi-norms); and *(ii)* not only are norms (and quasi-norms) not necessarily minimized, but they can grow towards infinity. We corroborate the analysis with empirical demonstrations.

Our findings imply that, rather than viewing implicit regularization in (shallow or deep) matrix factorization as minimizing a norm (or quasi-norm), a potentially more useful interpretation is *minimization of rank*. As a step towards assessing the generality of this interpretation, we empirically explore an extension of matrix factorization to *tensor factorization*.[3] Our experiments show that in analogy with matrix factorization, gradient descent on a tensor factorization tends to produce solutions with low rank, where rank is defined in the context of tensors.[4] Similarly to how matrix factorization corresponds to a linear neural network whose input-output mapping is represented by a matrix, it is known (see [22]) that tensor factorization corresponds to a *convolutional arithmetic circuit* (certain type of *non-linear* neural network) whose input-output mapping is represented by a tensor. We thus obtain a second exemplar of a neural network architecture whose implicit regularization strives to lower a notion of rank for its input-output mapping. This leads us to believe that the phenomenon may be general, and formalizing notions of rank for input-output mappings of contemporary models may be key to explaining generalization in deep learning.

The remainder of the paper is organized as follows. Section 2 presents the deep matrix factorization model. Section 3 delivers our analysis, showing that its implicit regularization can drive all norms to infinity. Experiments, with both the analyzed setting and tensor factorization, are given in Section 4. For conciseness, we defer our summary to Appendix A, and review related work in Appendix B.

## 2   Deep matrix factorization

Suppose we would like to complete a $d$-by-$d'$ matrix based on a set of observations $\{b_{i,j} \in \mathbb{R}\}_{(i,j) \in \Omega}$, where $\Omega \subset \{1, 2, \ldots, d\} \times \{1, 2, \ldots, d'\}$. A standard (underdetermined) loss function for the task is:

$$\ell : \mathbb{R}^{d,d'} \to \mathbb{R}_{\geq 0} \quad , \quad \ell(W) = \frac{1}{2} \sum_{(i,j) \in \Omega} \left( (W)_{i,j} - b_{i,j} \right)^2 . \tag{1}$$

Employing a depth $L$ matrix factorization, with hidden dimensions $d_1, d_2, \ldots, d_{L-1} \in \mathbb{N}$, amounts to optimizing the *overparameterized objective*:

$$\phi(W_1, W_2, \ldots, W_L) := \ell(W_{L:1}) = \frac{1}{2} \sum_{(i,j) \in \Omega} \left( (W_{L:1})_{i,j} - b_{i,j} \right)^2 , \tag{2}$$

where $W_l \in \mathbb{R}^{d_l, d_{l-1}}$, $l = 1, 2, \ldots, L$, with $d_L := d$, $d_0 := d'$, and:

$$W_{L:1} := W_L W_{L-1} \cdots W_1 , \tag{3}$$

referred to as the *product matrix* of the factorization. Our interest lies on the implicit regularization of gradient descent, *i.e.* on the type of product matrices (Equation (3)) it will find when applied to the overparameterized objective (Equation (2)). Accordingly, and in line with prior work (*cf.* [34, 8]), we focus on the case in which the search space is unconstrained, meaning $\min\{d_l\}_{l=0}^{L} = \min\{d_0, d_L\}$ (rank is not limited by the parameterization).

As a theoretical surrogate for gradient descent with small learning rate and near-zero initialization, similarly to [34] and [8] (as well as other works analyzing linear neural networks, *e.g.* [75, 6, 53, 7]), we study *gradient flow* (gradient descent with infinitesimally small learning rate):

$$\dot{W}_l(t) := \frac{d}{dt} W_l(t) = -\frac{\partial}{\partial W_l} \phi(W_1(t), W_2(t), \ldots, W_L(t)) \quad , \ t \geq 0 \ , \ l = 1, 2, \ldots, L, \tag{4}$$

and assume *balancedness* at initialization, *i.e.*:

$$W_{l+1}(0)^\top W_{l+1}(0) = W_l(0) W_l(0)^\top \quad , \ l = 1, 2, \ldots, L-1 . \tag{5}$$

In particular, when considering random initialization, we assume that $\{W_l(0)\}_{l=1}^{L}$ are drawn from a joint probability distribution by which Equation (5) holds almost surely. This is an idealization of standard random near-zero initializations, *e.g.* Xavier ([31]) and He ([40]), by which Equation (5) holds approximately with high probability (note that the equation holds exactly in the standard "residual" setting of identity initialization — *cf.* [38, 10]). The condition of balanced initialization (Equation (5)) played an important role in the analysis of [6], facilitating derivation of a differential equation governing the product matrix of a linear neural network (see Lemma 4 in Subappendix G.2.1). It was shown in [6] empirically (and will be demonstrated again in Section 4) that there is an excellent match between the theoretical predictions of gradient flow with balanced initialization, and its practical realization via gradient descent with small learning rate and near-zero initialization. Other works (*e.g.* [7, 45]) have supported this match theoretically, and we provide additional support in Appendix D by extending our theory to the case of unbalanced initialization (Equation (5) holding approximately).

Formally stated, Conjecture 1 from [34] treats the case $L = 2$, where the product matrix $W_{L:1}$ (Equation (3)) holds $\alpha \cdot W_{init}$ at initialization, $W_{init}$ being a fixed arbitrary full-rank matrix and $\alpha$ a varying positive scalar. Taking time to infinity ($t \to \infty$) and then initialization size to zero ($\alpha \to 0^+$), the conjecture postulates that if the limit product matrix $\bar{W}_{L:1} := \lim_{\alpha \to 0^+} \lim_{t \to \infty} W_{L:1}$ exists and is a global optimum for the loss $\ell(\cdot)$ (Equation (1)), *i.e.* $\ell(\bar{W}_{L:1}) = 0$, then it will be a global optimum with minimal nuclear norm, meaning $\bar{W}_{L:1} \in \arg\min_{W:\ell(W)=0} \|W\|_{nuclear}$. In contrast to Conjecture 1, Conjecture 2 from [8] can be interpreted as saying that for any depth $L \geq 2$ and any norm or quasi-norm $\|\cdot\|$, there exist observations $\{b_{i,j}\}_{(i,j) \in \Omega}$ for which global optimization of loss ($\lim_{\alpha \to 0^+} \lim_{t \to \infty} \ell(W_{1:L}) = 0$) does not imply minimization of $\|\cdot\|$ (*i.e.* we may have $\lim_{\alpha \to 0^+} \lim_{t \to \infty} \|W_{1:L}\| \neq \min_{W:\ell(W)=0} \|W\|$). Due to technical subtleties (for example the requirement of Conjecture 1 that a double limit of the product matrix with respect to time and initialization size exists), Conjectures 1 and 2 are not necessarily contradictory. However, they are in direct opposition in terms of the stances they represent — one supports the prospect of norms being able to explain implicit regularization in matrix factorization, and the other does not. The current paper seeks a resolution.

## 3   Implicit regularization can drive all norms to infinity

In this section we prove that for matrix factorization of depth $L \geq 2$, there exist observations $\{b_{i,j}\}_{(i,j) \in \Omega}$ with which optimizing the overparameterized objective (Equation (2)) via gradient flow (Equations (4) and (5)) leads — with probability 0.5 or more over random ("symmetric") initializa-

tion — *all* norms and quasi-norms of the product matrix (Equation (3)) to *grow towards infinity*, while its rank essentially decreases towards minimum. By this we not only affirm Conjecture 2, but in fact go beyond it in the following sense: *(i)* the conjecture allows chosen observations to depend on the norm or quasi-norm under consideration, while we show that the same set of observations can apply jointly to all norms and quasi-norms; and *(ii)* the conjecture requires norms and quasi-norms to be larger than minimal, while we establish growth towards infinity.

For simplicity of presentation, the current section delivers our construction and analysis in the setting $d = d' = 2$ (*i.e.* 2-by-2 matrix completion) — extension to different dimensions is straightforward (see Appendix E). We begin (Subsection 3.1) by introducing our chosen observations $\{b_{i,j}\}_{(i,j)\in\Omega}$ and discussing their properties. Subsequently (Subsection 3.2), we show that with these observations, decreasing loss often increases all norms and quasi-norms while lowering rank. Minimization of loss is treated thereafter (Subsection 3.3). Finally (Subsection 3.4), robustness of our construction to perturbations is established.

### 3.1 A simple matrix completion problem

Consider the problem of completing a 2-by-2 matrix based on the following observations:
$$\Omega = \{(1,2),(2,1),(2,2)\} \quad , \quad b_{1,2} = 1 \, , \, b_{2,1} = 1 \, , \, b_{2,2} = 0 \, . \tag{6}$$
The solution set for this problem (*i.e.* the set of matrices obtaining zero loss) is:
$$\mathcal{S} = \left\{ W \in \mathbb{R}^{2,2} : (W)_{1,2} = 1, (W)_{2,1} = 1, (W)_{2,2} = 0 \right\} \, . \tag{7}$$
Proposition 1 below states that minimizing a norm or quasi-norm along $W \in \mathcal{S}$ requires confining $(W)_{1,1}$ to a bounded interval, which for Schatten-$p$ (quasi-)norms (in particular for nuclear, Frobenius and spectral norms)[5] is simply the singleton $\{0\}$.

**Proposition 1.** *For any norm or quasi-norm over matrices $\|\cdot\|$ and any $\epsilon > 0$, there exists a bounded interval $I_{\|\cdot\|,\epsilon} \subset \mathbb{R}$ such that if $W \in \mathcal{S}$ is an $\epsilon$-minimizer of $\|\cdot\|$ (i.e. $\|W\| \leq \inf_{W'\in\mathcal{S}} \|W'\| + \epsilon$) then necessarily $(W)_{1,1} \in I_{\|\cdot\|,\epsilon}$. If $\|\cdot\|$ is a Schatten-$p$ (quasi-)norm, then in addition $W \in \mathcal{S}$ minimizes $\|\cdot\|$ (i.e. $\|W\| = \inf_{W'\in\mathcal{S}} \|W'\|$) if and only if $(W)_{1,1} = 0$.*

*Proof sketch (for complete proof see Subappendix G.3).* The (weakened) triangle inequality allows us to lower bound $\|\cdot\|$ by $|(W)_{1,1}|$ (up to multiplicative and additive constants). Thus, the set of $(W)_{1,1}$ values corresponding to $\epsilon$-minimizers must be bounded. If $\|\cdot\|$ is a Schatten-$p$ (quasi-)norm, a straightforward analysis shows it is monotonically increasing with respect to $|(W)_{1,1}|$, implying it is minimized if and only if $(W)_{1,1} = 0$. $\qquad\square$

In addition to norms and quasi-norms, we are also interested in the evolution of rank throughout optimization of a deep matrix factorization. More specifically, we are interested in the prospect of rank being implicitly minimized, as demonstrated empirically in [34, 8]. The discrete nature of rank renders its direct analysis unfavorable from a dynamical perspective (the rank of a matrix implies little about its proximity to low-rank), thus we consider the following surrogate measures: *(i) effective rank* (Definition 1 below; from [74]) — a continuous extension of rank used for numerical analyses; and *(ii) distance from infimal rank* (Definition 2 below) — (Frobenius) distance from the minimal rank that a given set of matrices may approach. According to Proposition 2 below, these measures independently imply that, although all solutions to our matrix completion problem — *i.e.* all $W \in \mathcal{S}$ (see Equation (7)) — have rank 2, it is possible to essentially minimize the rank to 1 by taking $|(W)_{1,1}| \to \infty$. Recalling Proposition 1, we conclude that in our setting, there is a direct contradiction between minimizing norms or quasi-norms and minimizing rank — the former requires confinement to some bounded interval, whereas the latter demands divergence towards infinity. This is the critical feature of our construction, allowing us to deem whether the implicit regularization in deep matrix factorization favors norms (or quasi-norms) over rank or vice versa.

**Definition 1** (from [74]). The *effective rank* of a matrix $0 \neq W \in \mathbb{R}^{d,d'}$ with singular values $\{\sigma_r(W)\}_{r=1}^{\min\{d,d'\}}$ is defined to be $\mathrm{erank}(W) := \exp\{H(\rho_1(W), \rho_2(W), \ldots, \rho_{\min\{d,d'\}}(W))\}$, where $\{\rho_r(W) := \sigma_r(W)/\sum_{r'=1}^{\min\{d,d'\}} \sigma_{r'}(W)\}_{r=1}^{\min\{d,d'\}}$ is a distribution induced by the singular values,

and $H(\rho_1(W), \rho_2(W), \ldots, \rho_{\min\{d,d'\}}(W)) := -\sum_{r=1}^{\min\{d,d'\}} \rho_r(W) \cdot \ln \rho_r(W)$ is its (Shannon) entropy (by convention $0 \cdot \ln 0 = 0$).

**Definition 2.** For a matrix space $\mathbb{R}^{d,d'}$, we denote by $D(\mathcal{S}, \mathcal{S}')$ the (Frobenius) distance between two sets $\mathcal{S}, \mathcal{S}' \subset \mathbb{R}^{d,d'}$ (i.e. $D(\mathcal{S}, \mathcal{S}') := \inf\{\|W - W'\|_{Fro} : W \in \mathcal{S}, W' \in \mathcal{S}'\}$), by $D(W, \mathcal{S}')$ the distance between a matrix $W \in \mathbb{R}^{d,d'}$ and the set $\mathcal{S}'$ (i.e. $D(W, \mathcal{S}') := \inf\{\|W - W'\|_{Fro} : W' \in \mathcal{S}'\}$), and by $\mathcal{M}_r$, for $r = 0, 1, \ldots, \min\{d, d'\}$, the set of matrices with rank $r$ or less (i.e. $\mathcal{M}_r := \{W \in \mathbb{R}^{d,d'} : \mathrm{rank}(W) \leq r\}$). The *infimal rank of the set $\mathcal{S}$* — denoted $\mathrm{irank}(\mathcal{S})$ — is defined to be the minimal $r$ such that $D(\mathcal{S}, \mathcal{M}_r) = 0$. The *distance of a matrix $W \in \mathbb{R}^{d,d'}$ from the infimal rank of $\mathcal{S}$* is defined to be $D(W, \mathcal{M}_{\mathrm{irank}(\mathcal{S})})$.

**Proposition 2.** *The effective rank (Definition 1) takes the values $(1, 2]$ along $\mathcal{S}$ (Equation (7)). For $W \in \mathcal{S}$, it is maximized when $(W)_{1,1} = 0$, and monotonically decreases to 1 as $|(W)_{1,1}|$ grows. Correspondingly, the infimal rank (Definition 2) of $\mathcal{S}$ is 1, and the distance of $W \in \mathcal{S}$ from this infimal rank is maximized when $(W)_{1,1} = 0$, monotonically decreasing to 0 as $|(W)_{1,1}|$ grows.*

*Proof sketch (for complete proof see Appendix G.4).* Analyzing the singular values of $W \in \mathcal{S}$ — $\sigma_1(W) \geq \sigma_2(W) \geq 0$ — reveals that: *(i)* $\sigma_1(W)$ attains a minimal value of 1 when $(W)_{1,1} = 0$, monotonically increasing to $\infty$ as $|(W)_{1,1}|$ grows; and *(ii)* $\sigma_2(W)$ attains a maximal value of 1 when $(W)_{1,1} = 0$, monotonically decreasing to 0 as $|(W)_{1,1}|$ grows. The results for effective rank, infimal rank and distance from infimal rank readily follow from this characterization. $\qquad\square$

### 3.2 Decreasing loss increases norms

Consider the process of solving our matrix completion problem (Subsection 3.1) with gradient flow over a depth $L \geq 2$ matrix factorization (Section 2). Theorem 1 below states that if the product matrix (Equation (3)) has positive determinant at initialization, lowering the loss leads norms and quasi-norms to increase, while the rank essentially decreases.

**Theorem 1.** *Suppose we complete the observations in Equation (6) by employing a depth $L \geq 2$ matrix factorization, i.e. by minimizing the overparameterized objective (Equation (2)) via gradient flow (Equations (4) and (5)). Denote by $W_{L:1}(t)$ the product matrix (Equation (3)) at time $t \geq 0$ of optimization, and by $\ell(t) := \ell(W_{L:1}(t))$ the corresponding loss (Equation (1)). Assume that $\det(W_{L:1}(0)) > 0$. Then, for any norm or quasi-norm over matrices $\|\cdot\|$:*

$$\|W_{L:1}(t)\| \geq a_{\|\cdot\|} \cdot \frac{1}{\sqrt{\ell(t)}} - b_{\|\cdot\|} \quad , \, t \geq 0, \tag{8}$$

*where $b_{\|\cdot\|} := \max\{\sqrt{2}a_{\|\cdot\|}, 8c_{\|\cdot\|}^2 \max_{i,j\in\{1,2\}} \|\mathbf{e}_i\mathbf{e}_j^\top\|\}$, $a_{\|\cdot\|} := \|\mathbf{e}_1\mathbf{e}_1^\top\| / (\sqrt{2}c_{\|\cdot\|})$, the vectors $\mathbf{e}_1, \mathbf{e}_2 \in \mathbb{R}^2$ form the standard basis, and $c_{\|\cdot\|} \geq 1$ is a constant with which $\|\cdot\|$ satisfies the weakened triangle inequality (see Footnote 2). On the other hand:*

$$\mathrm{erank}(W_{L:1}(t)) \leq \inf_{W'\in\mathcal{S}} \mathrm{erank}(W') + \tfrac{2\sqrt{12}}{\ln(2)} \cdot \sqrt{\ell(t)} \quad , \, t \geq 0, \tag{9}$$

$$D(W_{L:1}(t), \mathcal{M}_{\mathrm{irank}(\mathcal{S})}) \leq 3\sqrt{2} \cdot \sqrt{\ell(t)} \quad , \, t \geq 0, \tag{10}$$

*where $\mathrm{erank}(\cdot)$ stands for effective rank (Definition 1), and $D(\cdot, \mathcal{M}_{\mathrm{irank}(\mathcal{S})})$ represents distance from the infimal rank (Definition 2) of the solution set $\mathcal{S}$ (Equation (7)).*

*Proof sketch (for complete proof see Subappendix G.5).* Using a dynamical characterization from [8] for the singular values of the product matrix (restated in Subappendix G.2.1 as Lemma 5), we show that the latter's determinant does not change sign, *i.e.* it remains positive. This allows us to lower bound $|(W_{L:1})_{1,1}(t)|$ by $1/\sqrt{\ell(t)}$ (up to multiplicative and additive constants). Relating $|(W_{L:1})_{1,1}(t)|$ to (quasi-)norms, effective rank and distance from infimal rank then leads to the desired bounds. $\qquad\square$

An immediate consequence of Theorem 1 is that, if the product matrix (Equation (3)) has positive determinant at initialization, convergence to zero loss leads *all* norms and quasi-norms to *grow to infinity*, while the rank is essentially minimized. This is formalized in Corollary 1 below.

**Corollary 1.** *Under the conditions of Theorem 1, global optimization of loss, i.e. $\lim_{t\to\infty} \ell(t) = 0$, implies that for any norm or quasi-norm over matrices $\|\cdot\|$ we have that $\lim_{t\to\infty} \|W_{L:1}(t)\| = \infty$, where $W_{L:1}(t)$ is the product matrix of the deep factorization (Equation (3)) at time $t$ of optimization. On the other hand: $\lim_{t\to\infty} \mathrm{erank}(W_{L:1}(t)) = \inf_{W'\in\mathcal{S}} \mathrm{erank}(W')$ and*

$\lim_{t\to\infty} D(W_{L:1}(t), \mathcal{M}_{\mathrm{irank}(\mathcal{S})}) = 0$, *where* $\mathrm{erank}(\cdot)$ *stands for effective rank (Definition 1), and* $D(\cdot, \mathcal{M}_{\mathrm{irank}(\mathcal{S})})$ *represents distance from the infimal rank (Definition 2) of the solution set* $\mathcal{S}$ *(Equation (7)).*

*Proof.* Taking the limit $\ell(t) \to 0$ in the bounds given by Theorem 1 establishes the results. □

Theorem 1 and Corollary 1 imply that in our setting (Subsection 3.1), where minimizing norms (or quasi-norms) and minimizing rank contradict each other, the implicit regularization of deep matrix factorization is willing to completely give up on the former in favor of the latter, at least on the condition that the product matrix (Equation (3)) has positive determinant at initialization. How probable is this condition? By Proposition 3 below, it holds with probability $0.5$ if the product matrix is initialized by any one of a wide array of common distributions, including matrix Gaussian distribution with zero mean and independent entries, and a product of such. We note that rescaling (multiplying by $\alpha > 0$) initialization does not change the sign of product matrix's determinant, therefore as postulated by Conjecture 2, initialization close to the origin (along with small learning rate) can *not* ensure convergence to solution with minimal norm or quasi-norm.

**Proposition 3.** *If* $W \in \mathbb{R}^{d,d}$ *is a random matrix whose entries are drawn independently from continuous distributions, each symmetric about the origin, then* $\Pr(\det(W) > 0) = \Pr(\det(W) < 0) = 0.5$. *Furthermore, for* $L \in \mathbb{N}$, *if* $W_1, W_2, \ldots, W_L \in \mathbb{R}^{d,d}$ *are random matrices drawn independently from continuous distributions, and there exists* $l \in \{1, 2, \ldots, L\}$ *with* $\Pr(\det(W_l) > 0) = 0.5$, *then* $\Pr(\det(W_L W_{L-1} \cdots W_1) > 0) = \Pr(\det(W_L W_{L-1} \cdots W_1) < 0) = 0.5$.

*Proof sketch (for complete proof see Subappendix G.6).* Multiplying a row of $W$ by $-1$ keeps its distribution intact while flipping the sign of its determinant. This implies $\Pr(\det(W) > 0) = \Pr(\det(W) < 0)$. The first result then follows from the fact that a matrix drawn from a continuous distribution is almost surely non-singular. The second result is an outcome of the same fact, as well as the multiplicativity of determinant and the law of total probability. □

### 3.3 Convergence to zero loss

It is customary in the theory of deep learning (*cf.* [34, 36, 8]) to distinguish between implicit regularization — which concerns the type of solutions found in training — and the complementary question of whether training loss is globally optimized. We supplement our implicit regularization analysis (Subsection 3.2) by addressing this complementary question in two ways: *(i)* in Section 4 we empirically demonstrate that on the matrix completion problem we analyze (Subsection 3.1), gradient descent over deep matrix factorizations (Section 2) indeed drives training loss towards global optimum, *i.e.* towards zero; and *(ii)* in Proposition 4 below we theoretically establish convergence to zero loss for the special case of depth 2 and scaled identity initialization (treatment of additional depths and initialization schemes is left for future work). We note that when combined with Corollary 1, Proposition 4 affirms that in the latter special case, all norms and quasi-norms indeed grow to infinity while rank is essentially minimized.

**Proposition 4.** *Consider the setting of Theorem 1 in the special case of depth* $L = 2$ *and initial product matrix (Equation (3))* $W_{L:1}(0) = \alpha \cdot I$, *where* $I$ *stands for the identity matrix and* $\alpha \in (0, 1]$. *Under these conditions* $\lim_{t\to\infty} \ell(t) = 0$, *i.e. the training loss is globally optimized.*

*Proof sketch (for complete proof see Subappendix G.7).* We first establish that the product matrix is positive definite for all $t$. This simplifies a dynamical characterization from [6] (restated as Lemma 4 in Subappendix G.2), yielding lucid differential equations governing the entries of the product matrix. Careful analysis of these equations then completes the proof. □

### 3.4 Robustness to perturbations

Our analysis (Subsection 3.2) has shown that when applying a deep matrix factorization (Section 2) to the matrix completion problem defined in Subsection 3.1, if the product matrix (Equation (3)) has positive determinant at initialization — a condition that holds with probability $0.5$ under the wide variety of random distributions specified by Proposition 3 — then the implicit regularization drives *all* norms and quasi-norms *towards infinity*, while rank is essentially driven towards its minimum. A natural question is how common this phenomenon is, and in particular, to what extent does it persist if the observed entries we defined (Equation (6)) are perturbed. Theorem 2 in Appendix C generalizes Theorem 1 (from Subsection 3.2) to the case of arbitrary non-zero values for the off-diagonal observations $b_{1,2}, b_{2,1}$, and an arbitrary value for the diagonal observation $b_{2,2}$. In this generalization, the assumption (from Theorem 1) of the product matrix's determinant at initialization being positive

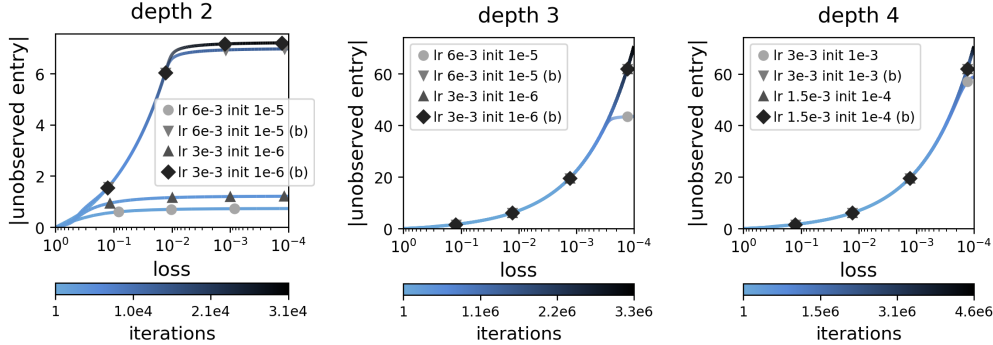

Figure 1: Implicit regularization in matrix factorization can drive *all* norms (and quasi-norms) *towards infinity*. For the matrix completion problem defined in Subsection 3.1, our analysis (Subsection 3.2) implies that with small learning rate and initialization close to the origin, when the product matrix (Equation (3)) is initialized to have positive determinant, gradient descent on a matrix factorization leads absolute value of unobserved entry to increase (which in turn means norms and quasi-norms increase) as loss decreases, *i.e.* as observations are fit. This is demonstrated in the plots above, which for representative runs, show absolute value of unobserved entry as a function of the loss (Equation 1), with iteration number encoded by color. Each plot corresponds to a different depth for the matrix factorization, and presents runs with varying configurations of learning rate and initialization (abbreviated as "lr" and "init", respectively). Both balanced (Equation 5) and unbalanced (layer-wise independent) random initializations were evaluated (former is marked by "(b)"). Independently for each depth, runs were iteratively carried out, with both learning rate and standard deviation for initialization decreased after each run, until the point where further reduction did not yield a noticeable change (presented runs are those from the last iterations of this process). Notice that depth, balancedness, and small learning rate and initialization, all contribute to the examined effect (absolute value of unobserved entry increasing as loss decreases), with the transition from depth 2 to 3 or more being most significant. Notice also that all runs initially follow the same curve, differing from one another in the point at which they diverge (enter a phase where examined effect is lesser). While a complete investigation of these phenomena is left for future work, we provide a partial theoretical explanation in Appendix D. For further implementation details, and similar experiments with different matrix dimensions, as well as perturbed and repositioned observations, see Appendix F.

is modified to an assumption of it having the same sign as $b_{1,2} \cdot b_{2,1}$ (the probability of which is also 0.5 under the random distributions covered by Proposition 3). Conditioned on the modified assumption, the smaller $|b_{2,2}|$ is compared to $|b_{1,2} \cdot b_{2,1}|$, the higher the implicit regularization is guaranteed to drive norms and quasi-norms, and the lower it is guaranteed to essentially drive the rank. Two immediate implications of Theorem 2 are: *(i)* if the diagonal observation is unperturbed ($b_{2,2} = 0$), the off-diagonal ones ($b_{1,2}, b_{2,1}$) can take on *any* non-zero values, and the phenomenon of implicit regularization driving norms and quasi-norms towards infinity (while essentially driving rank towards its minimum) will persist; and *(ii)* this phenomenon gracefully recedes as the diagonal observation is perturbed away from zero. We note that Theorem 2 applies even if the unobserved entry is repositioned, thus our construction is robust not only to perturbations in observed values, but also to an arbitrary change in the observed locations. See Subappendix F.1 for empirical demonstrations.

## 4 Experiments

This section presents our empirical evaluations. We begin in Subsection 4.1 with deep matrix factorization (Section 2) applied to the settings we analyzed (Section 3). Then, we turn to Subsection 4.2 and experiment with an extension to tensor (multi-dimensional array) factorization. For brevity, many details behind our implementation, as well as some experiments, are deferred to Appendix F.

### 4.1 Analyzed settings

In [34], Gunasekar *et al.* experimented with matrix factorization, arriving at Conjecture 1. In the following work [8], Arora *et al.* empirically evaluated additional settings, ultimately arguing against Conjecture 1, and raising Conjecture 2. Our analysis (Section 3) affirmed Conjecture 2, by providing a setting in which gradient descent (with infinitesimally small learning rate and initialization arbitrarily close to the origin) over (shallow or deep) matrix factorization provably drives *all* norms (and quasi-norms) *towards infinity*. Specifically, we established that running gradient descent on the overparameterized matrix completion objective in Equation (2), where the observed entries are those defined in Equation (6), leads the unobserved entry to diverge to infinity as loss converges to zero. Figure 1 demonstrates this phenomenon empirically. Figures 4 and 5 in Subappendix F.1 extend the experiment by considering, respectively: different matrix dimensions (see Appendix E); and

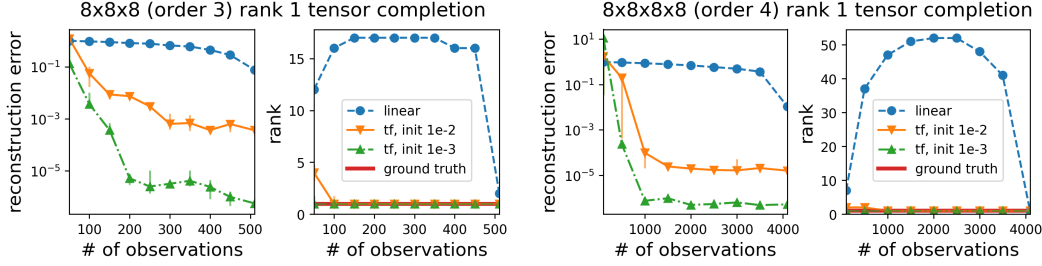

Figure 2: Gradient descent over tensor factorization exhibits an implicit regularization towards low (tensor) rank. Plots above report results of tensor completion experiments, comparing: *(i)* minimization of loss (Equation (11)) via gradient descent over tensor factorization (Equation (12) with $R$ large enough for expressing any tensor) starting from (small) random initialization (method is abbreviated as "tf"); against *(ii)* trivial baseline that matches observations while holding zeros in unobserved locations — equivalent to minimizing loss via gradient descent over linear parameterization (*i.e.* directly over $\mathcal{W}$) starting from zero initialization (hence this method is referred to as "linear"). Each pair of plots corresponds to a randomly drawn low-rank ground truth tensor, from which multiple sets of observations varying in size were randomly chosen. The ground truth tensors corresponding to left and right pairs both have rank 1 (for results obtained with additional ground truth ranks see Figure 6 in Subappendix F.1), with sizes 8-by-8-by-8 (order 3) and 8-by-8-by-8-by-8 (order 4) respectively. The plots in each pair show reconstruction errors (Frobenius distance from ground truth) and ranks (numerically estimated) of final solutions as a function of the number of observations in the task, with error bars spanning interquartile range (25'th to 75'th percentiles) over multiple trials (differing in random seed for initialization), and markers showing median. For gradient descent over tensor factorization, we employed an adaptive learning rate scheme to reduce run times (see Subappendix F.2 for details), and iteratively ran with decreasing standard deviation for initialization, until the point at which further reduction did not yield a noticeable change (presented results are those from the last iterations of this process, with the corresponding standard deviations annotated by "init"). Notice that gradient descent over tensor factorization indeed exhibits an implicit tendency towards low rank (leading to accurate reconstruction of low-rank ground truth tensors), and that this tendency is stronger with smaller initialization. For further details and experiments see Appendix F.

perturbations and repositionings applied to observations (*cf.* Subsection 3.4). The figures confirm that the inability of norms (and quasi-norms) to explain implicit regularization in matrix factorization translates from theory to practice.

## 4.2 From matrix to tensor factorization

At the heart of our analysis (Section 3) lies a matrix completion problem whose solution set (Equation (7)) entails a direct contradiction between minimizing norms (or quasi-norms) and minimizing rank. We have shown that on this problem, gradient descent over (shallow or deep) matrix factorization is willing to completely give up on the former in favor of the latter. This suggests that, rather than viewing implicit regularization in matrix factorization through the lens of norms (or quasi-norms), a potentially more useful interpretation is *minimization of rank*. Indeed, while global minimization of rank is in the worst case computationally hard (*cf.* [73]), it has been shown in [8] (theoretically as well as empirically) that the dynamics of gradient descent over matrix factorization promote sparsity of singular values, and thus they may be interpreted as searching for low rank locally. As a step towards assessing the generality of this interpretation, we empirically explore an extension of matrix factorization to *tensor factorization*.

In the context of matrix completion, (depth 2) matrix factorization amounts to optimizing the loss in Equation (1) by applying gradient descent to the parameterization $W = \sum_{r=1}^{R} \mathbf{w}_r \otimes \mathbf{w}_r'$, where $R \in \mathbb{N}$ is a predetermined constant, $\otimes$ stands for outer product,[6] and $\{\mathbf{w}_r \in \mathbb{R}^d\}_{r=1}^R$, $\{\mathbf{w}_r' \in \mathbb{R}^{d'}\}_{r=1}^R$ are the optimized parameters. The minimal $R$ required for this parameterization to be able to express a given $\bar{W} \in \mathbb{R}^{d,d'}$ is precisely the latter's rank. Implicit regularization towards low rank means that even when $R$ is large enough for expressing any matrix (*i.e.* $R \geq \min\{d, d'\}$), solutions expressible (or approximable) with small $R$ tend to be learned.

A generalization of the above is obtained by switching from matrices (tensors of *order* 2) to tensors of arbitrary order $N \in \mathbb{N}$. This gives rise to a *tensor completion* problem, with corresponding loss:

$$\ell : \mathbb{R}^{d_1, d_2, \dots, d_N} \to \mathbb{R}_{\geq 0} \quad , \quad \ell(\mathcal{W}) = \frac{1}{2} \sum_{(i_1, i_2, \dots, i_N) \in \Omega} \left( (\mathcal{W})_{i_1, i_2, \dots, i_N} - b_{i_1, i_2, \dots, i_N} \right)^2 , \quad (11)$$

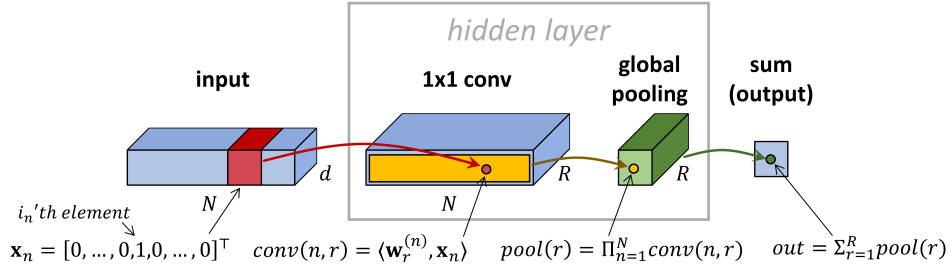

$$\mathbf{x}_n = [0, ..., 0, 1, 0, ..., 0]^\top \quad conv(n, r) = \langle \mathbf{w}_r^{(n)}, \mathbf{x}_n \rangle \quad pool(r) = \Pi_{n=1}^N conv(n, r) \quad out = \Sigma_{r=1}^R pool(r)$$

Figure 3: Tensor factorizations correspond to convolutional arithmetic circuits (class of *non-linear* neural networks studied extensively), analogously to how matrix factorizations correspond to linear neural networks. Specifically, the tensor factorization in Equation (12) corresponds to the convolutional arithmetic circuit illustrated above (illustration assumes $d_1 = d_2 = \cdots = d_N = d$ to avoid clutter). The input to the network is a tuple $(i_1, i_2, \ldots, i_N) \in \{1, 2, \ldots, d_1\} \times \{1, 2, \ldots, d_2\} \times \cdots \times \{1, 2, \ldots, d_N\}$, represented via one-hot vectors $(\mathbf{x}_1, \mathbf{x}_2, \ldots, \mathbf{x}_N) \in \mathbb{R}^{d_1} \times \mathbb{R}^{d_2} \times \cdots \times \mathbb{R}^{d_N}$. These vectors are processed by a hidden layer comprising: *(i)* locally connected linear operator with $R$ channels, the $r$'th one computing inner products against filters $(\mathbf{w}_r^{(1)}, \mathbf{w}_r^{(2)}, \ldots, \mathbf{w}_r^{(N)}) \in \mathbb{R}^{d_1} \times \mathbb{R}^{d_2} \times \cdots \times \mathbb{R}^{d_N}$ (this operator is referred to as "1×1 conv", appealing to the case of weight sharing, *i.e.* $\mathbf{w}_r^{(1)} = \mathbf{w}_r^{(2)} = \ldots = \mathbf{w}_r^{(N)}$); followed by *(ii)* global pooling computing products of all activations in each channel. The result of the hidden layer is then reduced through summation to a scalar — output of the network. Overall, given input tuple $(i_1, i_2, \ldots, i_N)$, the network outputs $(\mathcal{W})_{i_1, i_2, \ldots, i_N}$, where $\mathcal{W} \in \mathbb{R}^{d_1, d_2, \ldots, d_N}$ is given by the tensor factorization in Equation (12). Notice that the number of terms ($R$) and the tunable parameters ($\{\mathbf{w}_r^{(n)}\}_{r,n}$) in the factorization respectively correspond to the width and the learnable filters of the network. Our tensor factorization (Equation (12)) was derived as an extension of a shallow (depth 2) matrix factorization, and accordingly, the convolutional arithmetic circuit it corresponds to is shallow (has a single hidden layer). Endowing the factorization with hierarchical structures would render it equivalent to *deep* convolutional arithmetic circuits (see [22] for details) — investigation of the implicit regularization in these models is viewed as a promising avenue for future research.

where $\{b_{i_1, i_2, \ldots, i_N} \in \mathbb{R}\}_{(i_1, i_2, \ldots, i_N) \in \Omega}, \Omega \subset \{1, 2, \ldots, d_1\} \times \{1, 2, \ldots, d_2\} \times \cdots \times \{1, 2, \ldots, d_N\}$, stands for the set of observed entries. One may employ a tensor factorization by minimizing the loss in Equation (11) via gradient descent over the parameterization:

$$\mathcal{W} = \sum_{r=1}^R \mathbf{w}_r^{(1)} \otimes \mathbf{w}_r^{(2)} \otimes \cdots \otimes \mathbf{w}_r^{(N)} \quad , \quad \mathbf{w}_r^{(n)} \in \mathbb{R}^{d_n} \ , \ r = 1, 2, \ldots, R \ , \ n = 1, 2, \ldots, N \ , \quad (12)$$

where again, $R \in \mathbb{N}$ is a predetermined constant, $\otimes$ stands for outer product, and $\{\mathbf{w}_r^{(n)}\}_{r=1 \, n=1}^{R \quad N}$ are the optimized parameters. In analogy with the matrix case, the minimal $R$ required for this parameterization to be able to express a given $\mathcal{W} \in \mathbb{R}^{d_1, d_2, \ldots, d_N}$ is defined to be the latter's *(tensor) rank*. An implicit regularization towards low rank here would mean that even when $R$ is large enough for expressing any tensor, solutions expressible (or approximable) with small $R$ tend to be learned.

Figure 2 displays results of tensor completion experiments, in which tensor factorization (optimization of loss in Equation (11) via gradient descent over parameterization in Equation (12)) is applied to observations (*i.e.* $\{b_{i_1, i_2, \ldots, i_N}\}_{(i_1, i_2, \ldots, i_N) \in \Omega}$) drawn from a low-rank ground truth tensor. As can be seen in terms of both reconstruction error (distance from ground truth tensor) and (tensor) rank of the produced solutions, tensor factorizations indeed exhibit an implicit regularization towards low rank. The phenomenon thus goes beyond the special case of matrix (order 2 tensor) factorization. Theoretically supporting this finding is regarded as a promising direction for future research.

As discussed in Section 1, matrix completion can be seen as a prediction problem, and matrix factorization as its solution with a *linear neural network*. In a similar vein, tensor completion may be viewed as a prediction problem, and tensor factorization as its solution with a *convolutional arithmetic circuit* — see Figure 3. Convolutional arithmetic circuits form a class of *non-linear* neural networks that has been studied extensively in theory (*cf.* [22, 19, 20, 23, 77, 54, 24, 9, 55]), and has also demonstrated promising results in practice (see [18, 21, 78]). Analogously to how the input-output mapping of a linear neural network is naturally represented by a matrix, that of a convolutional arithmetic circuit admits a natural representation as a tensor. Our experiments (Figure 2 and Figure 6 in Subappendix F.1) show that (at least in some settings) when learned via gradient descent, this tensor tends to have low rank. We thus obtain a second exemplar of a neural network architecture whose implicit regularization strives to lower a notion of rank for its input-output mapping. This leads us to believe that the phenomenon may be general, and formalizing notions of rank for input-output mappings of contemporary models may be key to explaining generalization in deep learning.

## Broader Impact

The application of deep learning in practice is based primarily on trial and error, conventional wisdom and intuition, often leading to suboptimal performance, as well as compromise in important aspects such as safety, privacy and fairness. Developing rigorous theoretical foundations behind deep learning may facilitate a more principled use of the technology, alleviating aforementioned shortcomings. The current paper takes a step along this vein, by addressing the central question of implicit regularization induced by gradient-based optimization. While theoretical advances — particularly those concerned with explaining widely observed empirical phenomena — oftentimes do not pose apparent societal threats, a potential risk they introduce is misinterpretation by scientific readership. We have therefore made utmost efforts to present our results as transparently as possible.

## Acknowledgments and Disclosure of Funding

This work was supported by Len Blavatnik and the Blavatnik Family foundation, as well as the Yandex Initiative in Machine Learning. The authors thank Nathan Srebro and Jason D. Lee for their illuminating comments which helped improve the manuscript.

## Footnotes

[1]Due to lack of space, a significant portion of the paper is deferred to the appendices. We refer the reader to [72] for a self-contained version of the text.

[2]A *quasi-norm* $\|\cdot\|$ on a vector space $\mathcal{V}$ is a function from $\mathcal{V}$ to $\mathbb{R}_{\geq 0}$ that satisfies the same axioms as a norm, except for the triangle inequality $\forall v_1, v_2 \in \mathcal{V} : \|v_1 + v_2\| \leq \|v_1\| + \|v_2\|$, which is replaced by the weaker requirement $\exists c \geq 1 \ \ s.t. \ \ \forall v_1, v_2 \in \mathcal{V} : \|v_1 + v_2\| \leq c \cdot (\|v_1\| + \|v_2\|)$.

[3]For the sake of this paper, *tensors* can be thought of as $N$-dimensional arrays, with $N \in \mathbb{N}$ arbitrary (matrices correspond to the special case $N = 2$).

[4]The *rank of a tensor* is the minimal number of summands required to express it, where each summand is an outer product between vectors.

[5]For $p \in (0, \infty]$, the *Schatten-$p$ (quasi-)norm* of a matrix $W \in \mathbb{R}^{d,d'}$ with singular values $\{\sigma_r(W)\}_{r=1}^{\min\{d,d'\}}$ is defined as $\left(\sum_{r=1}^{\min\{d,d'\}} \sigma_r^p(W)\right)^{1/p}$ if $p < \infty$ and as $\max\{\sigma(W)\}_{r=1}^{\min\{d,d'\}}$ if $p = \infty$. It is a norm if $p \geq 1$ and a quasi-norm if $p < 1$. Notable special cases are nuclear (trace), Frobenius and spectral norms, corresponding to $p = 1, 2$ and $\infty$ respectively.

[6] Given $\{\mathbf{v}^{(n)} \in \mathbb{R}^{d_n}\}_{n=1}^N$, the outer product $\mathbf{v}^{(1)} \otimes \mathbf{v}^{(2)} \otimes \cdots \otimes \mathbf{v}^{(N)} \in \mathbb{R}^{d_1, d_2, \dots, d_N}$ — an order $N$ tensor — is defined by $(\mathbf{v}^{(1)} \otimes \mathbf{v}^{(2)} \otimes \cdots \otimes \mathbf{v}^{(N)})_{i_1, i_2, \dots, i_N} = (\mathbf{v}^{(1)})_{i_1} \cdot (\mathbf{v}^{(2)})_{i_2} \cdots (\mathbf{v}^{(N)})_{i_N}$.

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
