[Supplementary Material]

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

[7]As opposed to works studying the relation between implicit regularization and generalization (*cf.* [64, 66, 76]), or ones analyzing other sources of implicit regularization such as dropout (*e.g.* [83, 5]).

[8]Limiting such treatments to particular models is necessary, as in general one can not expect a gradient-based optimizer to yield minimal norm solutions over all possible objectives. Indeed, [80] and [25] have shown that there exist (carefully crafted) objectives over which variants of gradient descent do not produce minimal norm solutions. This accords with the conventional wisdom by which implicit regularization does not stem from an optimizer alone, but from its combination with a model (class of objectives).

[9]For a case related to (yet different from) that of [56], it was shown in [28] that the set of matrices fitting observations is in fact a singleton, meaning Conjecture 1 holds trivially.

[10] That is, for any $\theta_1, \theta_2 \in \mathcal{D}$ it holds that $\|\nabla f(\theta_2) - \nabla f(\theta_1)\|_2 \leq \beta \cdot \|\theta_2 - \theta_1\|_2$.

[11] The Frobenius norm of a matrix tuple is defined as the Euclidean norm of their concatenation as a vector, so for example $\|\theta(t)\|_{Fro} = (\sum_{l=1}^{L} \|W_l(t)\|_{Fro}^2)^{0.5}$.

[12]These assumptions are technical in nature; we defer their relaxation to future work.

[13]We did not attempt to optimize those; doing so is regarded as a potential direction for future work.

[14]Note that this phenomenon takes place even when initializations are perfectly balanced (*i.e.* have unbalancedness magnitude zero), the reason being the discrepancy between gradient descent with small learning rate and gradient flow. Specifically, while the latter would have conserved the balancedness throughout (Lemma 3), the former will (generically) lead to positive unbalancedness magnitude immediately after its commencement.

[15]That is, if the matrix to complete had size $d$-by-$d'$ with $d \neq d'$, we cleared rows $d' + 1$ to $d$ of $W_L(0)$ if $d > d'$, and columns $d + 1$ to $d'$ of $W_1(0)$ if $d' > d$.

[16]Positive for the experiments reported by Figures 1 and 4, and negative for those reported by Figure 5.

[17]As shown in [37], for any $d_1, d_2, \ldots, d_N \in \mathbb{N}$, using $R = (\Pi_{n=1}^N d_n)/\max\{d_n\}_{n=1}^N$ suffices for expressing all tensors in $\mathbb{R}^{d_1, d_2, \ldots, d_N}$.

[18] An infinitely differentiable function $f : \mathcal{D} \to \mathbb{R}$ is *analytic* if at every $x \in \mathcal{D}$ its Taylor series converges to it on some neighborhood of $x$ (see [52] for further details). Specifically, the matrix completion loss considered (Equation (1)) is analytic.

[19] As discussed in Section 2, mounting empirical and theoretical evidence suggest a close match between the predictions of gradient flow with balanced initialization, and its practical realization via gradient descent with small learning rate and near-zero initialization (*cf.* [6, 7, 45]). It was recently argued in [70] that certain aspects of balancedness do not transfer from gradient flow to gradient descent. However, the definitions in [70] deviate from the conventional ones, hence its conclusions are not applicable to standard settings.

[20]The claim relies on the fact that the Schatten-$p$ quasi-norm of $W_x$ is continuous with respect to $x$ for all $p \in (0, \infty)$. We note, however, that quasi-norms in general may be discontinuous.

[21] A technical subtlety is that, since analytic singular values can change order, it is not guaranteed in general that the minimal singular value is analytic in $t$, *i.e.* the characterization given in Lemma 5 does not necessarily apply to the minimal singular value. However, in our case, over $[t_0, T]$ the maximal singular value is at least $(1 - \sqrt{\ell_{init}})/\sqrt{2}$ (Equation (58)), while the minimal singular value is at most $(1 - \sqrt{\ell_{init}})/2$. Hence, no order change can occur in that time interval, and the movement of the minimal singular is indeed given by Lemma 5.

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

## Notes

a The *nuclear norm* (also known as *trace norm*) of a matrix is the sum of its singular values, regarded as a convex relaxation of rank.

b A technical subtlety of optimization in continuous time is that in principle, it is possible to asymptote (diverge to infinity) after finite time. In such a case, the asymptote is regarded as the end of optimization, and time tending to infinity ($t \to \infty$) is to be interpreted as tending towards that point.

c The formal statement in [34] applies to symmetric matrix factorization and positive definite $W_{init}$, but it is claimed thereafter that affirming the conjecture would imply the same for the asymmetric setting considered in this paper. We also note that the conjecture is stated in the context of matrix sensing, thus in particular applies to matrix completion (a special case).

d As stated in Section 2, we consider full-dimensional factorizations, in this case meaning that hidden dimensions $d_1, d_2, \ldots, d_{L-1}$ are all greater than or equal to 2.

e Recall that gradient flow corresponds to gradient descent with infinitesimally small learning rate.

f Notice that under (positively) scaled identity initialization the determinant of the product matrix (Equation (3)) is positive, as required by Corollary 1.

g The reader is referred to [51] and [37] for an introduction to tensor factorizations.

h To see that this parameterization is equivalent to the usual form $W = W_2 W_1$, simply view $R$ as the dimension shared between $W_1$ and $W_2$, $\{\mathbf{w}_r\}_{r=1}^R$ as the columns of $W_2$, and $\{\mathbf{w}_r'\}_{r=1}^R$ as the rows of $W_1$.

i There exist many types of tensor factorizations (*cf.* [51, 37]). We treat here the classic and most basic one, known as *CANDECOMP/PARAFAC (CP)*.

j Observed entries in the matrix to recover stand for training examples, and unobserved entries for test set.

## A    Summary

The extent to which norms (and quasi-norms) can explain the implicit regularization induced by gradient-based optimization is a central question in the theory of deep learning. A standard test-bed for its study is matrix factorization — matrix completion via linear neural networks trained by gradient descent — which in practice tends to produce low-rank solutions. It is an open problem whether the implicit regularization in matrix factorization can be characterized as minimization of a norm (or quasi-norm) — Conjecture 1 from [34] supports this supposition, whereas Conjecture 2 from [8] opposes it. We presented a simple (and robust to perturbations) matrix completion setting for which, with probability $0.5$ or more over random initialization of gradient descent, the implicit regularization in matrix factorization provably drives *all* norms (and quasi-norms) to *grow towards infinity*, while rank is essentially minimized. This affirms Conjecture 2, and although it does not formally refute Conjecture 1 (the latter's technical assumptions are not necessarily satisfied by our setting), we believe that in essence our result implies that norm (or quasi-norm) minimization cannot explain implicit regularization in matrix factorization, let alone in deep learning altogether.

The crux behind the matrix completion setting we defined is that its solution set entails a direct contradiction between minimizing norms (or quasi-norms) and minimizing rank. The fact that the former is given up on in favor of the latter suggests that, rather than viewing implicit regularization in matrix factorization through the lens of norms (or quasi-norms), a potentially more useful interpretation is minimization of rank. As a step towards assessing the generality of this interpretation, we experimented with an extension of matrix factorization to tensor factorization, and found that it too exhibits an implicit regularization towards low rank, where rank is defined in the context of tensors. Similarly to how matrix factorization corresponds to a linear neural network whose input-output mapping is represented by a matrix, tensor factorization corresponds to a convolutional arithmetic circuit (certain type of *non-linear* neural network) whose input-output mapping is represented by a tensor. We thus obtain a second exemplar of a neural network architecture whose implicit regularization strives to lower a notion of rank for its input-output mapping. Theoretical investigation of the implicit regularization in tensor factorization is regarded as a promising direction for future research. More broadly, we believe that neural networks minimizing notions of rank for their input-output mappings may be a general phenomenon, and hypothesize that formalizing such notions in the context of contemporary models may be key to explaining generalization in deep learning.

## B  Related work

Theoretical analysis of implicit regularization in deep learning is a highly active area of research. Our work extends the bulk of literature concerning mathematical characterization of the implicit regularization induced by gradient-based optimization.[7] Existing characterizations focus on different aspects of learning, for example: dynamics of optimization ([2, 29, 53, 8, 32, 48, 30]); curvature ("flatness") of obtained minima ([60]); frequency spectrum of learned input-output mappings ([71]); invariant quantities throughout training ([27]); and statistical properties imported from data ([12]). A ubiquitous approach, arguably more prevalent than the aforementioned, is to demonstrate that learned input-output mappings minimize some notion of norm, or analogously, maximize some notion of margin. Works along this line have treated various models,[8] including: linear (single-layer) predictors ([79, 35, 46, 3]); normalized linear models ([85]); certain polynomially parameterized linear models ([84]); homogeneous (and sum of homogeneous) models ([61, 57, 47]); ultra-wide neural networks ([43, 59, 17, 67]); linear neural networks with a single output ([62, 36, 45]); and matrix factorization — the subject of our inquiry.

Matrix factorization is perhaps the most extensively studied model in the context of implicit regularization induced by non-convex gradient-based optimization. It corresponds to linear neural networks with multiple inputs and outputs, typically trained to recover low-rank linear mappings. The literature on matrix factorization for low-rank matrix recovery is far too broad to cover here — we refer to [16] for a recent survey, while mentioning that the technique is often attributed to [13]. Notable works proving successful recovery of a low-rank matrix via matrix factorization trained by gradient descent with no explicit regularization are [82, 58, 56]. Of these, [56] can be viewed as affirming Conjecture 1 (from [34]) for a certain special case.[9] [11] has affirmed Conjecture 1 under different assumptions, but nonetheless argued empirically that it does not hold true in general, resonating with Conjecture 2 (from [8]). To the best of our knowledge, no theoretical support for the latter was provided prior to its proof in this paper. We note that the proof relies on technical results derived in [6] and [8] (restated in Subappendix G.2.1 for completeness).

Extending the research on matrix factorization, the use of tensor factorization for recovering low-rank tensors is a frequent topic of investigation (*cf.* [14, 86, 89, 87, 49, 63, 44, 1, 4]). Nevertheless, the experiments reported in this paper provide the first evidence we are aware of for such use to be successful under gradient-based optimization with no explicit regularization (in particular without imposing low-rank on the tensor factorization).

## C  Robustness to perturbations theorem

**Theorem 2.** *Consider the setting of Theorem 1 subject to the following changes:* (i) *the observations from Equation* (6) *are generalized to:*

$$\Omega = \{(1,2),(2,1),(2,2)\} \quad , \quad b_{1,2} = z \in \mathbb{R}\backslash\{0\} \,, \; b_{2,1} = z' \in \mathbb{R}\backslash\{0\} \,, \; b_{2,2} = \epsilon \in \mathbb{R}, \quad (13)$$

*leading to the following solution set in place of that from Equation* (7):

$$\widetilde{\mathcal{S}} = \left\{ W \in \mathbb{R}^{2,2} : (W)_{1,2} = z, (W)_{2,1} = z', (W)_{2,2} = \epsilon \right\} ; \quad (14)$$

*and* (ii) *the assumption* $\det(W_{L:1}(0)) > 0$ *is generalized to* $\text{sign}(\det(W_{L:1}(0))) = \text{sign}(z \cdot z')$, *where* $W_{L:1}(t)$ *denotes the product matrix (Equation* (3)) *at time* $t \geq 0$ *of optimization. Under these conditions, for any norm or quasi-norm over matrices* $\|\cdot\|$:

$$\|W_{L:1}(t)\| \geq a_{\|\cdot\|} \cdot \frac{|z| \cdot |z'|}{|\epsilon| + \sqrt{2\ell(t)}} - b_{\|\cdot\|} \quad , \; t \geq 0, \quad (15)$$

*where* $b_{\|\cdot\|} := \max\left\{ a_{\|\cdot\|} \cdot |z| \cdot |z'| / (|\epsilon| + \min\{|z|,|z'|\}) , \; 8c_{\|\cdot\|}^2 \max\{|z|,|z'|,|\epsilon|\} \max_{i,j \in \{1,2\}} \left\| \mathbf{e}_i \mathbf{e}_j^\top \right\| \right\}$, $a_{\|\cdot\|} := \left\| \mathbf{e}_1 \mathbf{e}_1^\top \right\| / c_{\|\cdot\|}$, *the vectors* $\mathbf{e}_1, \mathbf{e}_2 \in \mathbb{R}^2$ *form the standard basis, and* $c_{\|\cdot\|} \geq 1$ *is a constant*

*with which $\|\cdot\|$ satisfies the weakened triangle inequality (see Footnote 2). On the other hand:*

$$\text{erank}(W_{L:1}(t)) \leq \inf_{W' \in \widetilde{S}} \text{erank}(W') + \tfrac{16}{\min\{|z|,|z'|\}} \left( |\epsilon| + \sqrt{2\ell(t)} \right) \qquad , t \geq 0, \quad (16)$$

$$D(W_{L:1}(t), \mathcal{M}_{\text{irank}(\widetilde{S})}) \leq 4\,|\epsilon| + \left( 4 + \tfrac{\sqrt{|z|\cdot|z'|}}{\min\{|z|,|z'|\}} \right) \sqrt{2\ell(t)} \qquad , t \geq 0, \quad (17)$$

*where* $\text{erank}(\cdot)$ *stands for effective rank (Definition 1), and* $D(\cdot, \mathcal{M}_{\text{irank}(\widetilde{S})})$ *represents distance from the infimal rank (Definition 2) of the solution set* $\widetilde{S}$. *Moreover, Equations* (15), (16) *and* (17) *hold even if the above setting is further generalized as follows:* (i) *the unobserved entry resides in location* $(i, j) \in \{1, 2\} \times \{1, 2\}$, *with* $z, z' \in \mathbb{R}\backslash\{0\}$ *observed in the adjacent locations and* $\epsilon \in \mathbb{R}$ *in the diagonally-opposite one; and* (ii) *the sign of* $\det(W_{L:1}(0))$ *is equal to that of* $z \cdot z'$ *if* $i = j$, *and opposite to it otherwise.*

*Proof sketch (for complete proof see Subappendix G.8).* The proof follows a line similar to that of Theorem 1, with slightly more involved derivations. $\qquad\square$

**Disqualifying implicit minimization of norms in finite settings** Theorem 1 (in Subsection 3.2), which applies to our original (unperturbed) matrix completion problem (Subsection 3.1), has shown that the implicit regularization of deep matrix factorization (Section 2) does not minimize norms or quasi-norms, by establishing lower bounds (Equation (8) with $\|\cdot\|$ ranging over all possible norms and quasi-norms) that tend to infinity as the training loss $\ell(t)$ converges to zero. The more general Theorem 2 allows disqualifying implicit minimization of norms or quasi-norms without requiring divergence to infinity. To see this, consider the lower bounds it establishes (Equation (15) with $\|\cdot\|$ ranging over all norms and quasi-norms), and in particular their limits as $\ell(t) \to 0$. If the observed value $\epsilon$ is different from zero, these limits are all finite. Moreover, given a particular norm or quasi-norm $\|\cdot\|$, we may choose $\epsilon$ different from zero, yet small enough such that the lower bound for $\|\cdot\|$ has limit arbitrarily larger than the infimum of $\|\cdot\|$ over the solution set.

# D    Extension to unbalanced initialization

In this appendix we present extensions of our theory to cases where initialization is unbalanced, *i.e.* in which Equation (5) only holds approximately. For simplicity, we limit the presentation to the square setting, where all dimensions of the deep matrix factorization $(d_0, d_1, \ldots, d_L$; see Section 2) are equal to $d \in \mathbb{N}$.

The following definition quantifies unbalancedness.

**Definition 3.** The *unbalancedness magnitude* of matrices $\{W_l \in \mathbb{R}^{d,d}\}_{l=1}^L$ is defined to be:

$$\max_{l \in \{1, 2, \ldots, L-1\}} \left\| W_{l+1}^\top W_{l+1} - W_l W_l^\top \right\|_{nuclear}. \quad (18)$$

We will present two approaches for showing that approximate versions of our main theoretical results (Theorems 1 and 2) hold if unbalancedness magnitude at initialization is small: *(i)* using continuity of optimizer trajectory with respect to its initialization (Subappendix D.1); and *(ii)* employing conservation of unbalancedness magnitude throughout optimization (Subappendix D.2). Both approaches rely on the fact that small unbalancedness magnitude implies proximity to perfect balancedness, as stated formally in the following lemma.

**Lemma 1.** *For any matrices* $\{W_l \in \mathbb{R}^{d,d}\}_{l=1}^L$ *with unbalancedness magnitude (Equation* (18)*) equal to* $\epsilon$, *there exist* $\{W'_l \in \mathbb{R}^{d,d}\}_{l=1}^L$ *that are balanced (i.e. have unbalancedness magnitude zero), such that* $\|W_l - W'_l\|_{Fro} \leq (l-1) \cdot \sqrt{\epsilon}$ *for all* $l = 1, 2, \ldots, L$.

*Proof sketch (for complete proof see Subappendix G.9).* Based on singular value decompositions of $W_1, W_2, \ldots, W_L$, the proof provides an explicit construction for $W'_1, W'_2, \ldots, W'_L$. Starting with $W'_1 := W_1$, for $l = 2, 3, \ldots, L$ the matrices $W'_l$ are defined such that: *(i)* $W'^\top_l W'_l = W'_{l-1} W'^\top_{l-1}$; and *(ii)* $\|W_l - W'_l\|_{Fro} \leq \|W_{l-1} - W'_{l-1}\|_{Fro} + \sqrt{\epsilon}$. The former ensures that $W'_1, W'_2, \ldots, W'_L$ are balanced, whereas the latter implies $\|W_l - W'_l\|_{Fro} \leq (l-1) \cdot \sqrt{\epsilon}$ for all $l = 1, 2, \ldots, L$. $\quad\square$

### D.1 First approach: continuity with respect to initialization

Trajectories of gradient flow over a smooth objective are Lipschitz continuous with respect to their initialization, in the sense that for any $T > 0$, the location at time $T$ of optimization is a Lipschitz continuous function of the initial point. This fact is established by Lemma 2 below.

**Lemma 2.** *Let $\mathcal{D}$ be an open domain in Euclidean space, and let $f : \mathcal{D} \to \mathbb{R}$ be a twice continuously differentiable function. Denote by $\|\cdot\|_2$ the Euclidean norm, and assume that $f(\cdot)$ is $\beta$-smooth with respect to $\|\cdot\|_2$ for some $\beta \geq 0$.[10] Let $\theta, \theta' : [0, T] \to \mathcal{D}$, where $T > 0$, be two curves born from gradient flow over $f(\cdot)$:*

$$\theta(0) = \theta_0 \in \mathcal{D} \quad , \quad \dot{\theta}(t) := \tfrac{d}{dt}\theta(t) = -\nabla f(\theta(t)) \; , \; t \in [0, T],$$

$$\theta'(0) = \theta'_0 \in \mathcal{D} \quad , \quad \dot{\theta}'(t) := \tfrac{d}{dt}\theta'(t) = -\nabla f(\theta'(t)) \; , \; t \in [0, T].$$

*Then, for any $\bar{t} \in [0, T]$ it holds that:*

$$\|\theta(\bar{t}) - \theta'(\bar{t})\|_2 \leq \|\theta(0) - \theta'(0)\|_2 \cdot \exp(\beta \cdot \bar{t}). \tag{19}$$

*Proof sketch (for complete proof see Subappendix G.10).* Define the function $g : [0, T] \to \mathbb{R}_{\geq 0}$ by $g(t) := \|\theta(t) - \theta'(t)\|_2^2$. Since $f(\cdot)$ is $\beta$-smooth, it holds that $\dot{g}(t) := \tfrac{d}{dt}g(t) \leq 2\beta \cdot g(t)$ for all $t \in [0, T]$. Dividing the latter inequality by $g(t)$ (with special treatment for the case where $g(t) = 0$) and integrating over time yields the desired result. $\square$

Specializing Lemma 2 to deep matrix factorization (Section 2) yields the following result.

**Proposition 5.** *Consider the overparameterized objective corresponding to a depth $L$ matrix factorization applied to an arbitrary matrix completion task (see Equations (2) and (3)). Let $\theta(t) := (W_1(t), W_2(t), \ldots, W_L(t))$ and $\theta'(t) := (W'_1(t), W'_2(t), \ldots, W'_L(t))$ be two (arbitrary) curves born from gradient flow over this objective (cf. Equation (4)). Given $T > 0$, denote $R := \sup_{t \in [0,T]} \max\{\|\theta(t)\|_{Fro}, \|\theta'(t)\|_{Fro}\}$.[11] Then, for any $\bar{t} \in [0, T]$ it holds that:*

$$\|\theta(\bar{t}) - \theta'(\bar{t})\|_{Fro} \leq \|\theta(0) - \theta'(0)\|_{Fro} \cdot \exp\left(L R^{L-2}\left(2R^L + B\right) \cdot \bar{t}\right), \tag{20}$$

*where $B := (\sum_{(i,j) \in \Omega} b_{i,j}^2)^{0.5}$, with $\{b_{i,j}\}_{(i,j) \in \Omega}$ standing for the observed matrix entries.*

*Proof sketch (for complete proof see Subappendix G.11).* The proof follows from Lemma 2, and the fact that for any $R' > 0$ the overparameterized objective is $L R'^{L-2}\left(2R'^L + B\right)$-smooth over $\mathcal{D}_{R'} := \{(W_1, W_2, \ldots, W_L) : \|(W_1, W_2, \ldots, W_L)\|_{Fro} < R'\}$. $\square$

Combining Proposition 5 with Lemma 1 makes it possible to derive extensions of Theorems 1 and 2 in which the assumption of initialization being perfectly balanced (Equation (5)) is relaxed to a requirement for small unbalancedness magnitude (Definition 3). The underlying idea is as follows. An initialization with small unbalancedness magnitude is close to one which is balanced (Lemma 1), and for the latter Theorems 1 and 2 may be applied. The distance between gradient flow trajectories emanating from the two initializations is controlled (Proposition 5), therefore results of Theorems 1 and 2 (bounds on norms, quasi-norms, effective rank and distance from infimal rank) carry over — with additional error terms — to the trajectory originating from the unbalanced initialization. A drawback of this approach is that the bounds on distance between trajectories, and accordingly the error terms incurred, grow exponentially with time (see Equation (20)). In the next subappendix we present a different approach that takes into account specific properties of gradient flow over deep matrix factorization, allowing one to overcome this exponential growth (for depth $L \geq 3$).

### D.2 Second approach: conservation of unbalancedness magnitude

Lemma 3 below shows that unbalancedness magnitude (Definition 3) is a conserved quantity of gradient flow over deep matrix factorization (Section 2).

**Lemma 3.** *Consider the overparameterized objective corresponding to a depth $L$ matrix factorization applied to an arbitrary matrix completion task (see Equations (2) and (3)). Let $(W_1(t), W_2(t), \ldots, W_L(t))$ be a curve born from gradient flow over this objective (cf. Equation (4)), and for any $t \geq 0$, denote by $\epsilon(t)$ the associated unbalancedness magnitude (Equation (18)). Then, $\epsilon(t)$ is constant through time, i.e. $\epsilon(t) = \epsilon(0)$ for all $t \geq 0$.*

*Proof sketch (for complete proof see Subappendix G.12).* For $l = 1, 2, \ldots, L-1$, using the dynamics of $W_l(t)$ and $W_{l+1}(t)$ under gradient flow, we show that:

$$\tfrac{d}{dt}(W_l(t)W_l(t)^\top) = \tfrac{d}{dt}(W_{l+1}(t)^\top W_{l+1}(t)) \quad , \forall t \geq 0 \, .$$

This implies $W_{l+1}(t)^\top W_{l+1}(t) - W_l(t)W_l(t)^\top = W_{l+1}(0)^\top W_{l+1}(0) - W_l(0)W_l(0)^\top$ for all $t \geq 0$. The proof concludes by taking nuclear norm of both sides of the latter equality, followed by maximization over $l \in \{1, 2, \ldots, L-1\}$. $\qquad\square$

Combining Lemma 3 with Lemma 1 implies that if unbalancedness magnitude is small at initialization, it remains that way throughout, and thus for every point along the optimization trajectory there exists some nearby point which is balanced (*i.e.* has unbalancedness magnitude zero). We may imagine a gradient flow trajectory emanating from such balanced point, and import certain characteristics from this imaginary trajectory to the original one. The idea of using imaginary balancedly-initialized trajectories for analyzing the unbalanced case also appears in the approach laid out in Subappendix D.1. However, whereas there only one such trajectory was employed, here there are infinitely many — one for each point in time. This allows us to maintain small distance from an imaginary trajectory (as opposed to a distance that grows exponentially with time — see Proposition 5), facilitating import of characteristics during which incurred error terms are small.

In the context of Theorems 1 and 2, the critical characteristic of trajectories originating from balanced initializations is that they do not allow the product matrix's (Equation (3)) determinant to change sign, or more specifically, its smallest singular value to cross zero. Using the aforementioned technique (proximity to imaginary balancedly-initialized trajectories), we may import an approximate version of this characteristic into trajectories whose initializations have small unbalancedness magnitude. This amounts to a bound on the rate at which the smallest singular value of the product matrix can approach zero, yielding a guaranteed time throughout which the results of Theorems 1 and 2 hold.

Theorem 3 below formalizes the logic outlined above, extending Theorem 1 to the case of unbalanced initialization (we omit here the formal extension of Theorem 2, as it is essentially the same).

**Theorem 3.** *Consider the setting of Theorem 1, with the assumption of balanced initialization (Equation (5)) removed, allowing initialization with unbalancedness magnitude $\epsilon > 0$ (Definition 3). Assume that:* (i) *the deep matrix factorization is square, i.e. its hidden dimensions are equal to* 2; (ii) *the loss at initialization is lower than that at zero, i.e. $\ell(W_{L:1}(0)) < \ell(0) = 1$, where $W_{L:1}(t)$ denotes the product matrix (Equation (3)) at time $t \geq 0$ of optimization; and* (iii)

$$\epsilon \leq \begin{cases} \exp\left( -\dfrac{2^{16}\left(\max\left\{32, \max_{l\in[L]}\|W_l(0)\|_{Fro}\right\}+1\right)^6}{(1-\sqrt{\ell_{init}})^4} \right) & \text{, if depth } L = 2 \\[2em] \dfrac{(1-\sqrt{\ell_{init}})^{128}}{2^{64L+256}L^{128}\left(\max\left\{32, \max_{l\in[L]}\|W_l(0)\|_{Fro}\right\}+1\right)^{128L-64}} & \text{, if depth } L \geq 3 \end{cases} ,$$

*where $\ell_{init} := \ell(W_{L:1}(0))$.*[12] *Then, the results of Theorem 1 — bounds on (quasi-)norms, effective rank and distance from infimal rank (Equations (8), (9) and (10) respectively) — all hold at least until one of the following takes place:*

- *Optimization time $t$ reaches:*

$$t = \begin{cases} \dfrac{1}{2^{2/3}(1-\sqrt{\ell_{init}})^{4/3}} \cdot \ln\left(\tfrac{1}{\epsilon}\right)^{2/3} - \ln\left(\dfrac{e}{(1-\sqrt{\ell_{init}})\sigma_{init}}\right) & \text{, if depth } L = 2 \\[1.5em] \dfrac{2^{4L/3}L}{(1-\sqrt{\ell_{init}})^2} \cdot \epsilon^{-\frac{3L-8}{32L-16}} - 2^{-(5L+5)}\sigma_{init}^{-\frac{L-2}{L}} & \text{, if depth } L \geq 3 \end{cases} , \qquad (21)$$

*where $\sigma_{init} := \min\{\sigma_{min}(W_{L:1}(0)), (1 - \sqrt{\ell_{init}})/2\}$, with $\sigma_{min}(W_{L:1}(0))$ standing for the minimal singular value of $W_{L:1}(0)$; or*

- *(Quasi-)norms, effective rank and distance from infimal rank are jointly bounded as follows:*

$$
\|W_{L:1}(t)\| \geq \begin{cases} \frac{\|\mathbf{e}_1 \mathbf{e}_1^\top\|(1-\sqrt{\ell_{init}})^{4/3}}{2^{11}c_{\|\cdot\|}} \cdot \ln\left(\frac{1}{\epsilon}\right)^{1/3} - 12c_{\|\cdot\|}^2 \max_{i,j\in\{1,2\}} \|\mathbf{e}_i \mathbf{e}_j^\top\| & \text{, if depth } L = 2 \\[2ex] \frac{\|\mathbf{e}_1 \mathbf{e}_1^\top\|(1-\sqrt{\ell_{init}})^{6/5}}{2^{4L}L^{6/5}c_{\|\cdot\|}} \cdot \epsilon^{-\frac{L}{128L-64}} - 12c_{\|\cdot\|}^2 \max_{i,j\in\{1,2\}} \|\mathbf{e}_i \mathbf{e}_j^\top\| & \text{, if depth } L \geq 3 \end{cases},
$$
(22)

$$
\mathrm{erank}(W_{L:1}(t)) \leq \begin{cases} \inf_{W'\in\mathcal{S}} \mathrm{erank}(W') + \frac{2^9}{(1-\sqrt{\ell_{init}})^{2/3}} \cdot \ln\left(\frac{1}{\epsilon}\right)^{-1/6} & \text{, if depth } L = 2 \\[2ex] \inf_{W'\in\mathcal{S}} \mathrm{erank}(W') + \frac{2^{2L+5}L}{1-\sqrt{\ell_{init}}} \cdot \epsilon^{\frac{L}{256L-128}} & \text{, if depth } L \geq 3 \end{cases},
$$
(23)

$$
D(W_{L:1}(t), \mathcal{M}_{\mathrm{irank}(\mathcal{S})}) \leq \begin{cases} \frac{2^{12}}{(1-\sqrt{\ell_{init}})^{4/3}} \cdot \ln\left(\frac{1}{\epsilon}\right)^{-1/3} + \sqrt{2\ell(W_{L:1}(t))} & \text{, if depth } L = 2 \\[2ex] \frac{2^{3L+4}L^{6/5}}{(1-\sqrt{\ell_{init}})^{6/5}} \cdot \epsilon^{\frac{L}{128L-64}} + \sqrt{2\ell(W_{L:1}(t))} & \text{, if depth } L \geq 3 \end{cases},
$$
(24)

*where $\|\cdot\|$ is any norm or quasi-norm over matrices, $c_{\|\cdot\|} \geq 1$ is a constant with which $\|\cdot\|$ satisfies the weakened triangle inequality (see Footnote 2), $\mathrm{erank}(\cdot)$ stands for effective rank (Definition 1), and $D(\cdot, \mathcal{M}_{\mathrm{irank}(\mathcal{S})})$ represents distance from the infimal rank (Definition 2) of the solution set $\mathcal{S}$ (Equation (7)).*

*Proof sketch (for complete proof see Subappendix G.13).* By the proof of Theorem 1 (given in Subappendix G.5), its results (Equations (8), (9) and (10)) hold for any $t \geq 0$ with $\det(W_{L:1}(t)) > 0$. Bearing in mind that by assumption $\det(W_{L:1}(0)) > 0$, we let $T \in (0, \infty)$ be the initial time at which $\det(W_{L:1}(T)) = 0$ (if no such $T$ exists, the proof concludes). Fixing an arbitrary time $\bar{t} \in [0, T]$, Lemmas 3 and 1 imply that there exists a point $(W_1', W_2', \ldots, W_L')$ which meets the balancedness condition (*i.e.* has unbalancedness magnitude zero), and is within (Frobenius) distance $\mathcal{O}(L^2 \cdot \sqrt{\epsilon})$ from $(W_1(\bar{t}), W_2(\bar{t}), \ldots, W_L(\bar{t}))$. Imagining a gradient flow path that emanates from $(W_1', W_2', \ldots, W_L')$, one may employ Lemma 5 from Subappendix G.2.1, to characterize the movement of the singular values of $W_{L:1}' := W_L'W_{L-1}' \cdots W_1'$. Continuity arguments then imply that the singular values of $W_{L:1}(\bar{t})$ move similarly (at time $\bar{t}$), allowing us to obtain an upper bound on the rate at which the minimal singular value of $W_{L:1}(\bar{t})$ can decay. Integrating this upper bound yields a lower bound on $T$, specified in Equation (21) as one of the possible outcomes. The continuity arguments employed require $\|(W_1(t), W_2(t), \ldots, W_L(t))\|_{Fro}$, $t \in [0, T]$, to be bounded by a certain constant. If this is not the case then necessarily at some time $t \leq T$ the unobserved entry of $W_{L:1}(t)$ is large, leading to the bounds on (quasi-)norms, effective rank and distance from infimal rank in the alternative outcome (Equations (22), (23) and (24) respectively). $\qquad\square$

Theorem 3 states that if initialization has unbalancedness magnitude $\epsilon > 0$ (Definition 3), then the results of Theorem 1 — bounds on (quasi-)norms, effective rank and distance from infimal rank (Equations (8), (9) and (10) respectively) — are guaranteed to hold for a certain period of time (Equation (21)), or until certain terminal bounds (Equations (22), (23) and (24)) are jointly satisfied. Taking $\epsilon \to 0^+$, the aforementioned period of time tends to infinity, and the terminal bounds tend to the limits (as loss goes to zero) of the bounds in Theorem 1, meaning we effectively converge to the latter. The rate of this convergence highly depends on the depth of the matrix factorization — roughly speaking, it is proportional to a fractional power of $\ln(1/\epsilon)$ for depth 2, and to a fractional power of $1/\epsilon$ for depth 3 or more. Disregarding constants (terms that do not depend on $\epsilon$),[13] this implies that in order to get comparable guarantees, the unbalancedness magnitude of initialization needs to be exponentially smaller with depth 2 than with depth 3 or more. We thus have a theoretical reasoning that resonates with the empirical phenomenon reported in Figure 1, by which in practical settings (gradient descent with small learning rate and near-zero initialization), the prediction of Theorem 1 — unobserved entry increasing (and therefore norms and quasi-norms increasing, with

effective rank and distance from infimal rank decreasing) as loss decreases — sustains for much longer with depth 3 or more than it does with depth 2.[14]

# E  Extension to different matrix dimensions

In this appendix we outline an extension of the construction and analysis given in Subsections 3.1 and 3.2 respectively, to completion of matrices with dimensions beyond 2-by-2. The extension presented here is not unique, but rather one simple option out of many. It is demonstrated empirically in Subappendix F.1 (Figure 4).

Beginning with square matrices, for $2 \leq d \in \mathbb{N}$, consider completion of a $d$-by-$d$ matrix based on the following observations:

$$
\begin{aligned}
\Omega &= \{1, \ldots, d\} \times \{1, \ldots, d\} \setminus \{(1,1)\}, \\
b_{i,j} &= \begin{cases} 1 & \text{, if } i = j \geq 3 \text{ or } (i,j) \in \{(1,2),(2,1)\} \\ 0 & \text{, otherwise} \end{cases} \quad \text{, for } (i,j) \in \Omega,
\end{aligned} \tag{25}
$$

where, as in Section 2, $\Omega$ represents the set of observed locations, and $\{b_{i,j} \in \mathbb{R}\}_{(i,j) \in \Omega}$ the corresponding set of observed values. The solution set for this problem (*i.e.* the set of matrices zeroing the loss in Equation (1)) is:

$$
\mathcal{S}_d := \left\{ \begin{pmatrix} w_{1,1} & 1 & 0 & 0 & \cdots & 0 \\ 1 & 0 & 0 & 0 & \cdots & 0 \\ 0 & 0 & 1 & 0 & \cdots & 0 \\ 0 & 0 & 0 & 1 & & 0 \\ \vdots & \vdots & \vdots & & \ddots & \\ 0 & 0 & 0 & 0 & & 1 \end{pmatrix} \in \mathbb{R}^{d,d} : w_{1,1} \in \mathbb{R} \right\}. \tag{26}
$$

Figure 4: Phenomenon of implicit regularization in matrix factorization driving *all* norms (and quasi-norms) *towards infinity* extends to arbitrary matrix dimensions. Appendix E outlines an extension of the construction and analysis given in Subsections 3.1 and 3.2 respectively, to completion of matrices with arbitrary dimensions. The extension implies that for any $2 \leq d, d' \in \mathbb{N}$, when applying matrix factorization to the specified $d$-by-$d'$ matrix completion problem, decreasing loss, *i.e.* fitting observations, can lead absolute value of unobserved entry to increase (which in turn means norms and quasi-norms increase). This is demonstrated in the plot above, which for representative runs corresponding to different choices of $d$ and $d'$, shows absolute value of unobserved entry as a function of the loss (Equation 1), with iteration number encoded by color. Runs were obtained with a depth 3 matrix factorization initialized randomly by an unbalanced (layer-wise independent) distribution, with the latter's standard deviation and the learning rate for gradient descent set to the smallest values used for depth 3 in Figure 1 (other settings we evaluated produced similar results). For further implementation details see Subappendix F.2.1.

Observing $\mathcal{S}_d$, while comparing to the solution set $\mathcal{S}$ in our original construction (Equation (7)), we see that the former has a 2-by-2 block diagonal structure, with the top-left block holding the latter, and the bottom-right block set to identity. This implies that $d - 2$ of the singular values along $\mathcal{S}_d$ are fixed to one, and the remaining two are identical to the singular values along $\mathcal{S}$. Results analogous to Propositions 1 and 2 can therefore easily be proven. Since the determinant along $\mathcal{S}_d$ is bounded below and away from zero (it is equal to $-1$), approaching $\mathcal{S}_d$ while having positive determinant necessarily means that absolute value of unobserved entry (*i.e.* of the entry in location $(1,1)$) grows towards infinity. Combining this with the fact that the product matrix (Equation (3)) of a depth $L \geq 2$ matrix factorization maintains the sign of its determinant (see Lemma 6 in Subappendix G.2.1), results analogous to Theorem 1 and Corollary 1 may readily be established. That is, one may show that, with probability $0.5$ or more over random near-zero initialization, gradient descent with small learning rate drives *all* norms (and quasi-norms) *towards infinity*, while essentially driving rank towards its minimum.

Moving on to the rectangular case, for $2 \leq d, d' \in \mathbb{N}$, consider completion of a $d$-by-$d'$ matrix based on the

Figure 5: Phenomenon of implicit regularization in matrix factorization driving *all* norms (and quasi-norms) *towards infinity* is robust to perturbations. Our analysis (Subsection 3.4) implies that, when applying matrix factorization to the matrix completion problem defined in Subsection 3.1, even if observations are perturbed and repositioned, decreasing loss, *i.e.* fitting them, leads absolute value of unobserved entry to increase (which in turn means norms and quasi-norms increase). Specifically, with $(i, j) \in \{1, 2\} \times \{1, 2\}$ representing the unobserved location and $\bar{i} := 3 - i$, $\bar{j} := 3 - j$, Theorem 2 implies that: *(i)* if the diagonally-opposite observation $b_{\bar{i}, \bar{j}}$ is unperturbed (stays at zero), the adjacent ones $b_{i, \bar{j}}, b_{\bar{i}, j}$ can take on *any* non-zero values, and as long as at initialization the sign of the product matrix's (Equation 3) determinant accords with that of $b_{i, \bar{j}} \cdot b_{\bar{i}, j}$, the absolute value of unobserved entry will grow to infinity; and *(ii)* the extent to which absolute value of unobserved entry grows gracefully recedes as $b_{\bar{i}, \bar{j}}$ is perturbed away from zero. This is demonstrated in the plots above, which for representative runs, show absolute value of unobserved entry as a function of the loss (Equation 1), with iteration number encoded by color. Each plot corresponds to a different choice of $(i, j)$ and a different assignment for $b_{i, \bar{j}}, b_{\bar{i}, j}$, presenting runs with varying values for $b_{\bar{i}, \bar{j}}$. Runs were obtained with a depth 3 matrix factorization initialized randomly by an unbalanced (layer-wise independent) distribution, with the latter's standard deviation and the learning rate for gradient descent set to the smallest values used for depth 3 in Figure 1 (other settings we evaluated produced similar results). For further implementation details see Subappendix F.2.1.

same observations as in Equation (25), but with additional zero observations such that only the entry in location $(1, 1)$ is unobserved. The singular values along the solution set for this problem are the same as those along $\mathcal{S}_d$ (Equation (26)). Moreover, assuming without loss of generality that $d \leq d'$, if a matrix factorization applied to this problem is initialized such that its product matrix holds zeros in columns $d + 1$ to $d'$, then a dynamical characterization from [6] (restated as Lemma 4 in Subappendix G.2.1), along with the structure of the loss (Equation (1)), ensure the leftmost $d$-by-$d$ submatrix of the product matrix evolves precisely as in the square case discussed above, while the remaining columns ($d + 1$ to $d'$) stay at zero. Results thus carry over from the square to the rectangular case.

# F  Further experiments and implementation details

## F.1  Further experiments

Figures 4 and 5 supplement Figure 1 from Subsection 4.1, by demonstrating empirically that the phenomenon of implicit regularization in matrix factorization driving all norms (and quasi-norms) towards infinity is, respectively: *(i)* applicable to arbitrary matrix dimensions, as outlined in Appendix E; and *(ii)* robust to perturbations, as proven in Subsection 3.4. Figure 6 supplements Figure 2 from Subsection 4.2, further demonstrating that gradient descent over tensor factorization exhibits an implicit regularization towards low (tensor) rank.

## F.2  Implementation details

Below we provide a full description of implementation details omitted from our experimental reports (Section 4 and Subappendix F.1). Source code for reproducing our results and figures, based on the PyTorch framework ([68]), can be found at `https://github.com/noamrazin/imp_reg_dl_not_norms`.

### F.2.1  Deep matrix factorization (Figures 1, 4, and 5)

In all experiments with deep matrix factorization, hidden dimensions were set to the minimal value ensuring unconstrained search space, *i.e.* to the minimum between the number of rows and the number of columns in the matrix to complete. Gradient descent was run with fixed learning rate until loss (Equation (1)) reached a value lower than $10^{-4}$ or $5 \cdot 10^6$ iterations elapsed. Both balanced (Equa-

Figure 6: Gradient descent over tensor factorization exhibits an implicit regularization towards low (tensor) rank. This figure is identical to Figure 2, except that the experiments it portrays had ground truth tensors of rank 3 (instead of 1). For further details see caption of Figure 2, as well as Subappendix F.2.2.

tion (5)) and unbalanced (layer-wise independent) random initializations were calibrated according to a desired standard deviation $\alpha > 0$ for the entries of the initial product matrix (Equation (3)). Namely: *(i)* under unbalanced initialization, entries of all weight matrices were sampled independently from a Gaussian distribution with zero mean and standard deviation $(\alpha^2/\bar{d}^{L-1})^{1/2L}$, where $L$ stands for the depth of the factorization, and $\bar{d}$ for the size of its hidden dimensions; and *(ii)* under balanced initialization, we used Procedure 1 from [7], based on a Gaussian distribution with independent entries, zero mean and standard deviation $\alpha$. In accordance with the description in Appendix E, if the matrix to complete was rectangular, we ensured that excess rows or columns of the initial product matrix held zeros, by clearing (setting to zero) corresponding rows or columns of the initial leftmost or rightmost (respectively) matrix in the factorization.[15] Random initializations were repeated until the determinant of the initial product matrix (or of its top-left $\min\{d, d'\}$-by-$\min\{d, d'\}$ submatrix if its size was $d$-by-$d'$ with $d \neq d'$) was of the necessary sign,[16] taking two attempts on average. In the experiment reported by Figure 1, runs with matrix factorization depths 2 and 3 were carried out with learning rates $\{6 \cdot 10^{-2}, 3 \cdot 10^{-2}, 9 \cdot 10^{-3}, 6 \cdot 10^{-3}, 3 \cdot 10^{-3}, 9 \cdot 10^{-4}\}$ and corresponding standard deviations for initialization $\{10^{-2}, 10^{-3}, 10^{-4}, 10^{-5}, 10^{-6}, 10^{-7}\}$. Factorizations of depth 4 were slightly more sensitive to changes in learning rate, thus we refined attempted values to $\{6 \cdot 10^{-3}, 4.5 \cdot 10^{-3}, 3 \cdot 10^{-3}, 1.5 \cdot 10^{-3}, 10^{-3}\}$, with corresponding standard deviations for initialization $\{10^{-1}, 10^{-2}, 10^{-3}, 10^{-4}, 10^{-5}\}$.

### F.2.2 Tensor factorization (Figures 2 and 6)

In all experiments with tensor factorization (Equation (12)), the number of terms $R$ was set to ensure an unconstrained search space, *i.e.* it was set to $8^2$ and $8^3$ for tensor sizes 8-by-8-by-8 and 8-by-8-by-8-by-8 respectively.[17] Horizontal axes in all plots begin from the smallest number of observations producing stable results, and end when all entries but one are observed. Specifically: *(i)* in the experiments with rank 1 ground truth tensors (Figure 2), the number of observations ranged over $\{50, 100, 150, \ldots, 400, 450, 511\}$ and $\{100, 500, 1000, 1500, \ldots, 3000, 3500, 4095\}$ for orders 3 and 4 respectively; and *(ii)* for experiments with rank 3 ground truth tensors (Figure 6), the minimal number of observations was increased threefold (*i.e.* ranges of $\{150, 200, 250, \ldots, 400, 450, 511\}$ and $\{300, 500, 1000, 1500, \ldots, 3000, 3500, 4095\}$ were used for orders 3 and 4 respectively). Gradient descent was run until the mean squared error over observations reached a value lower than $10^{-6}$ or $10^6$ iterations elapsed. For initialization, weights were sampled independently from a Gaussian distribution with zero mean and varying standard deviation. In particular, five trials (differing in random seed) were conducted for each standard deviation in the range $\{10^{-1}, 10^{-2}, 10^{-3}, 10^{-4}\}$. To facilitate more efficient experimentation, we employed an adaptive learning rate scheme, where at each iteration a base learning rate of $10^{-2}$ was divided by the square root of an exponential moving average of squared gradient norms. That is, with base learning rate $\eta = 10^{-2}$ and weighted average coefficient $\beta = 0.99$, at iteration $t$ the learning rate was set to $\eta_t = \eta/(\sqrt{\gamma_t/(1-\beta^t)} + 10^{-6})$, where $\gamma_t = \beta \cdot \gamma_{t-1} + (1-\beta) \cdot \sum_{r=1}^{R}\sum_{n=1}^{N}\|\partial/\partial\mathbf{w}_r^{(n)}\ell(\{\mathbf{w}_r^{(n)}(t)\}_{r,n})\|_F^2$, with $\gamma_0 = 0$ and $\ell(\cdot)$ standing for the mean squared error over observations. We emphasize that only the learning rate (step size) is affected by this scheme, not the direction of movement. Comparisons between the scheme and a fixed

(small) learning rate schedule have shown no noticeable impact on the end result, with significant difference in terms of run time.

When referring to tensor rank, we mean the classic *CP-rank* (see [51]). While exact inference of this rank is in the worst case computationally hard (*cf.* [39]), in practice, a standard way to estimate it is by the minimal number of terms ($R$ in Equation (12)) for which the Alternating Least Squares (ALS) algorithm achieves reconstruction (mean squared) error below a certain threshold (see [51] for further details). We follow this method with a threshold of $10^{-6}$. Generating a ground truth rank $R^*$ tensor $\mathcal{W}^* \in \mathbb{R}^{d_1, d_2, \ldots, d_N}$ was done by computing:

$$\mathcal{W}^* = \sum_{r=1}^{R^*} \mathbf{w}_r^{*(1)} \otimes \mathbf{w}_r^{*(2)} \otimes \cdots \otimes \mathbf{w}_r^{*(N)} \quad , \mathbf{w}_r^{*(n)} \in \mathbb{R}^{d_n} , r = 1, 2, \ldots, R^* , n = 1, 2, \ldots, N ,$$

with $\{\mathbf{w}_r^{*(n)}\}_{r=1}^{R^*}{}_{n=1}^{N}$ drawn independently from the standard normal distribution. After every such generation, we estimated the rank of the obtained tensor (its construction only ensures a rank of *at most $R^*$*), and repeated the process if it was smaller than $R^*$. For convenience, we subsequently normalized the ground truth tensor to be of unit Frobenius norm.

# G Deferred proofs

## G.1 Notations

We define a few notational conventions that will be used throughout our proofs. For $N \in \mathbb{N}$, let $[N]$ denote the set $\{1, 2, \ldots, N\}$. Let $\{\mathbf{e}_i\}_{i=1}^d \subset \mathbb{R}^d$ be the standard basis vectors, *i.e.* $\mathbf{e}_i$ holds 1 in its $i$'th coordinate and 0 elsewhere. The singular values of a matrix $W \in \mathbb{R}^{d,d'}$ are denoted by $\sigma_1(W) \geq \ldots \geq \sigma_{\min\{d,d'\}}(W) \geq 0$, where by convention $\sigma_i(W) := 0$ for $i > \min\{d, d'\}$. Similarly, the eigenvalues of a symmetric matrix $W \in \mathbb{R}^{d,d}$ are denoted by $\lambda_1(W) \geq \ldots \geq \lambda_d(W)$. We let $\|W\|_{S_p}$, with $p \in (0, \infty]$, stand for the Schatten-$p$ (quasi-)norm of a matrix $W \in \mathbb{R}^{d,d'}$, and denote by $\|W\|_F$ the special case $p = 2$, *i.e.* the Frobenius norm. The Euclidean norm of a vector $w \in \mathbb{R}^d$ is denoted by $\|w\|_2$. Since norms are a special case of quasi-norms, when providing results applicable to both, only the latter is explicitly treated. To admit a compact representation of matrix products, given $1 \leq a \leq b \leq L$ and matrices $W_1, W_2, \ldots, W_L$ for which the product $W_L W_{L-1} \cdots W_1$ is defined, we denote:

$$\prod_a^{r=b} W_r := W_b \cdots W_a ,$$
$$\prod_{r=a}^b W_r^\top := W_a^\top \cdots W_b^\top .$$

By definition, if $a > b$, then both $\prod_a^{r=b} W_r$ and $\prod_{r=a}^b W_r^\top$ are identity matrices, with size to be inferred by context. The $k$'th derivative of a function (from $\mathbb{R}$ to $\mathbb{R}$) $f(t)$ is denoted by $f^{(k)}(t)$, with $f^{(0)}(t) := f(t)$ by convention. For consistency with differential equations literature, when the variable $t$ is regarded as a time index, we also denote the first order derivative by $\dot{f}(t)$. Lastly, when clear from context, a time index $t$ will often be omitted.

## G.2 Useful lemmas

### G.2.1 Deep matrix factorization

For completeness, we include the following result from [6], which characterizes the evolution of the product matrix under gradient flow on a deep matrix factorization:

**Lemma 4** (adaptation of Theorem 1 in [6]). *Let $\ell : \mathbb{R}^{d,d'} \to \mathbb{R}_{\geq 0}$ be an analytic[18] loss, overparameterized by a depth $L$ matrix factorization:*

$$\phi(W_1, \ldots, W_L) = \ell(W_L W_{L-1} \cdots W_1).$$

*Suppose we run gradient flow over the factorization:*

$$\dot{W}_l(t) := \tfrac{d}{dt} W_l(t) = -\tfrac{\partial}{\partial W_l} \phi(W_1(t), \ldots, W_L(t)) \quad , t \geq 0 , l = 1, \ldots, L,$$

*with a balanced initialization, i.e.:*

$$W_{l+1}(0)^\top W_{l+1}(0) = W_l(0)W_l(0)^\top \quad , l = 1, \ldots, L-1.$$

*Then, the product matrix $W_{L:1}(t) = W_L(t) \cdots W_1(t)$ obeys the following dynamics:*

$$\dot{W}_{L:1}(t) = -\sum_{l=1}^{L} \left[ W_{L:1}(t)W_{L:1}(t)^\top \right]^{\frac{l-1}{L}} \cdot \nabla\ell\big(W_{L:1}(t)\big) \cdot \left[ W_{L:1}(t)^\top W_{L:1}(t) \right]^{\frac{L-l}{L}},$$

*where $[\cdot]^\beta$, $\beta \in \mathbb{R}_{\geq 0}$, stands for a power operator defined over positive semidefinite matrices (with $\beta = 0$ yielding identity by definition).*[19]

Additionally, recall from [8] the following characterization for the singular values of $W_{L:1}(t)$:

**Lemma 5** (adaptation of Lemma 1 and Theorem 3 in [8]). *Consider the setting of Lemma 4 for depth $2 \leq L \in \mathbb{N}$. Then, there exist analytical functions $\{\sigma_r : [0,\infty) \to \mathbb{R}\}_{r=1}^{\min\{d,d'\}}$, $\{\mathbf{u}_r : [0,\infty) \to \mathbb{R}^d\}_{r=1}^{\min\{d,d'\}}$ and $\{\mathbf{v}_r : [0,\infty) \to \mathbb{R}^{d'}\}_{r=1}^{\min\{d,d'\}}$ such that:*

$$\sigma_r(t) \geq 0 \;,\;\; \mathbf{u}_r(t)^\top \mathbf{u}_{r'}(t) = \mathbf{v}_r(t)^\top \mathbf{v}_{r'}(t) = \begin{cases} 1 & , r = r' \\ 0 & , r \neq r' \end{cases} \;,\; t \geq 0 \;,\; r,r' \in [\min\{d,d'\}]$$

$$W_{L:1}(t) = \sum_{r=1}^{\min\{d,d'\}} \sigma_r(t)\mathbf{u}_r(t)\mathbf{v}_r(t)^\top,$$

*i.e. $\sigma_r(t)$ are the singular values of $W_{L:1}(t)$, and $\mathbf{u}_r(t), \mathbf{v}_r(t)$ are corresponding left and right (respectively) singular vectors. Furthermore, the singular values $\sigma_r(t)$ evolve by:*

$$\dot{\sigma}_r(t) = -L \cdot \left(\sigma_r^2(t)\right)^{1-1/L} \cdot \left\langle \nabla\ell\left(W_{L:1}(t)\right), \mathbf{u}_r(t)\mathbf{v}_r(t)^\top \right\rangle \quad , r = 1, \ldots, \min\{d,d'\}. \quad (27)$$

We rely on this result to establish that for square product matrices the sign of $\det(W_{L:1}(t))$ does not change throughout time.

**Lemma 6.** *Consider the setting of Lemma 4 with depth $2 \leq L \in \mathbb{N}$ and $d = d'$. Then, the determinant of $W_{L:1}(t)$ has the same sign as its initial value $\det(W_{L:1}(0))$. That is, $\det(W_{L:1}(t))$ is identically zero if $\det(W_{L:1}(0)) = 0$, is positive if $\det(W_{L:1}(0)) > 0$, and is negative if $\det(W_{L:1}(0)) < 0$.*

*Proof.* We prove an analogous claim for the singular values of $W_{L:1}(t)$, from which the lemma readily follows. That is, for $r \in [d]$, the singular value $\sigma_r(t)$ is identically zero if $\sigma_r(0) = 0$, and is positive if $\sigma_r(0) > 0$.

For conciseness, define $g(t) := -L \cdot \left\langle \nabla\ell\left(W_{L:1}(t)\right), \mathbf{u}_r(t)\mathbf{v}_r(t)^\top \right\rangle$. Invoking Lemma 5, let us solve the differential equation for $\sigma_r(t)$. If $L = 2$, the solution to Equation (27) is $\sigma_r(t) = \sigma_r(0) \cdot \exp\left(\int_{t'=0}^{t} g(t')dt'\right)$. Clearly, $\sigma_r(t)$ is either identically zero or positive according to its initial value. If $L > 2$, Equation (27) is solved by:

$$\sigma_r(t) = \begin{cases} \left(\sigma_r(0)^{\frac{2}{L}-1} + \left(\frac{2}{L}-1\right)\int_{t'=0}^{t} g(t')dt'\right)^{\frac{1}{\frac{2}{L}-1}} & , \sigma_r(0) > 0 \\ 0 & , \sigma_r(0) = 0 \end{cases}.$$

As before, if $\sigma_r(0) = 0$, then $\sigma_r(t) = 0$ for all $t \geq 0$. If $\sigma_r(0) > 0$, divergence in finite time of $\sigma_r(t)$ is possible, however, its positivity is preserved until that occurs nonetheless.

Turning our attention to the determinant of $W_{L:1}(t)$, suppose $\det(W_{L:1}(0)) = 0$. Then, $W_{L:1}(0)$ has a singular value which is $0$, and for all $t$ that singular value and the determinant remain $0$. If $\det(W_{L:1}(0)) \neq 0$, the product matrix remains full rank for all $t$. The proof then immediately follows from the continuity of $\det(W_{L:1}(t))$. $\qquad\square$

We will also make use of the following lemmas:

**Lemma 7** (adapted from [8]). *Under the setting of Lemma 4, $W_1(t), W_2(t), \ldots, W_L(t), W_{L:1}(t)$ and $\nabla \ell(W_{L:1}(t))$ are analytic functions of $t$.*

*Proof.* Analytic functions are closed under summation, multiplication, and composition. The analyticity of $\ell(\cdot)$ therefore implies that $\phi(\cdot)$ (Equation (2)) is analytic as well. From Theorem 1.1 in [41], it then follows that under gradient flow (Equation (4)) $W_1(t), W_2(t), \ldots, W_L(t)$ are analytic functions of $t$. Lastly, the aforementioned closure properties imply that $W_{L:1}(t)$ and $\nabla \ell(W_{L:1}(t))$ are also analytic in $t$. □

**Lemma 8.** *For any matrices $\{W_l \in \mathbb{R}^{d_l, d_{l-1}}\}_{l=1}^L$, with $d_0, d_1, \ldots, d_L \in \mathbb{N}$, that are balanced, i.e. that meet $W_{l+1}^\top W_{l+1} = W_l W_l^\top$ for all $l \in [L-1]$, it holds that $\sigma_i(W_{L:1}) = \sigma_i(W_l)^L$ for all $l \in [L]$ and $i \in [\min\{d_L, d_0\}]$.*

*Proof.* We construct singular value decompositions for $W_1, W_2, \ldots, W_L$ in an iterative process as follows. First, let $W_1 = U_1 \Sigma_1 V_1^\top$ be an arbitrary singular value decomposition of $W_1$, i.e. $U_1 \in \mathbb{R}^{d_1, d_1}, V_1 \in \mathbb{R}^{d_0, d_0}$ are orthogonal matrices, and $\Sigma_1 \in \mathbb{R}_{\geq 0}^{d_1, d_0}$ is rectangular-diagonal holding the singular values of $W_1$ in non-increasing order. Then, for $l = 2, 3, \ldots, L$, balancedness of $W_1, W_2, \ldots, W_L$ and Lemma 13 imply that $W_l$ has a singular value decomposition $U_l \Sigma_l U_{l-1}^\top$, where: $U_l \in \mathbb{R}^{d_l d_l}$ is orthogonal; $\Sigma_l \in \mathbb{R}_{\geq 0}^{d_l, d_{l-1}}$ is rectangular-diagonal holding the singular values of $W_l$ in non-increasing order; and $\sigma_i(W_l) = (\Sigma_l)_{i,i} = (\Sigma_{l-1})_{i,i} = \sigma_i(W_{l-1})$ for $i \in [\min\{d_l, d_{l-1}, d_{l-2}\}]$, with the remaining diagonal entries of $\Sigma_l$ and $\Sigma_{l-1}$ being 0. With $\{U_l\}_{l=1}^L, \{\Sigma_l\}_{l=1}^L, V_1$ as described above, the product matrix can be written as:

$$W_{L:1} = \prod_1^{l=L} W_l = \left[ \prod_2^{l=L} U_l \Sigma_l U_{l-1}^\top \right] \cdot U_1 \Sigma_1 V_1^\top = U_L \cdot \prod_1^{l=L} \Sigma_l \cdot V_1^\top.$$

That is, $U_L \cdot \prod_1^{l=L} \Sigma_l \cdot V_1^\top$ is a singular value decomposition of $W_{L:1}$, and $\sigma_i(W_{L:1}) = (\prod_{l=1}^L \Sigma_l)_{i,i}$ for all $i \in [\min\{d_L, d_0\}]$.

Fix $l \in [L]$ and $i \in [\min\{d_L, d_0\}]$. If $i > \min\{d_{l'}\}_{l'=0}^L$, for any $l' \in [L]$ with $\min\{d_{l'}, d_{l'-1}\} \geq i$, by our construction it holds that $(\Sigma_{l'})_{i,i} = 0$. Hence, recalling that by convention $\sigma_i(W_l) = 0$ if $i > \min\{d_l, d_{l-1}\}$, we may conclude that $\sigma_i(W_{L:1}) = \sigma_i(W_l)^L = 0$. Otherwise, if $i \leq \min\{d_{l'}\}_{l'=0}^L$, the fact that $\sigma_i(W_{l'}) = (\Sigma_{l'})_{i,i} = (\Sigma_{l'-1})_{i,i} = \sigma_i(W_{l'-1})$ for all $l' = 2, 3, \ldots, L$ implies that $\sigma_i(W_{L:1}) = \sigma_i(W_l)^L$. □

### G.2.2 Technical

Included below are a few technical lemmas used in our analyses.

**Lemma 9.** *Let $h : [0, 1] \to \mathbb{R}$ be the binary entropy function $h(p) := -p \cdot \ln(p) - (1-p) \ln(1-p)$, where by convention $0 \cdot \ln(0) = 0$. Then, for all $p \in [0, 1]$:*

$$h(p) \leq 2\sqrt{p}.$$

*Proof.* We present a tighter inequality, $h(p) \leq 2\sqrt{p(1-p)}$, from which the proof immediately follows since $2\sqrt{p(1-p)} \leq 2\sqrt{p}$ for $p \in [0, 1]$.

Define the function $f(p) := \frac{h(p)^2}{p(1-p)}$ over the open interval $(0, 1)$. Differentiating it with respect to $p$ we have:

$$\frac{d}{dp} f(p) = \frac{(-p \cdot \ln(p))^2 - (-(1-p) \cdot \ln(1-p))^2}{p^2(1-p)^2}.$$

Introducing $g(p) := -p \cdot \ln(p)$, we show that $g(p)^2 > g(1-p)^2$ for all $p \in (0, \frac{1}{2})$. It is easily verified that $g(p) - g(1-p)$ is concave on the interval $(0, \frac{1}{2})$ (second derivative is negative). Since for $p = 0$ and $p = 1/2$ we have exactly $g(p) - g(1-p) = 0$, it holds that $g(p) - g(1-p) \geq 0$ and $g(p)^2 \geq g(1-p)^2$ for all $p \in (0, \frac{1}{2})$. Noticing $\frac{d}{dp} f(p) = \left( g(p)^2 - g(1-p)^2 \right) / p^2(1-p)^2$, it follows that $f(\cdot)$ is monotonically non-decreasing on $(0, \frac{1}{2})$. Due to the fact that $f(p) = f(1-p)$, it is non-increasing on $(\frac{1}{2}, 1)$, and attains its maximal value over $(0, 1)$ at $p = \frac{1}{2}$. Putting it all together, for $p \in (0, 1)$ we have:

$$h(p) \leq \sqrt{p(1-p)} \cdot \sqrt{f(1/2)} = 2\ln(2) \cdot \sqrt{p(1-p)} \leq 2\sqrt{p(1-p)},$$

and for $p = 0, 1$ there is exact equality, completing the proof. $\qquad\square$

**Lemma 10.** *Let $f, g : [0, \infty) \to \mathbb{R}$ be real analytic functions (see Footnote 18) such that $f^{(k)}(0) = g^{(k)}(0)$ for all $k \in \mathbb{N} \cup \{0\}$. Then, $f(t) = g(t)$ for all $t \geq 0$.*

*Proof.* Define the function $h(t) := f(t) - g(t)$. Since analytic functions are closed under subtraction, $h(\cdot)$ is analytic as well. An analytic function with all zero derivatives at a point is constant on the corresponding connected component. Noticing that $h^{(k)}(0) = 0$ for all $k \in \mathbb{N} \cup \{0\}$, we may conclude that $h(t) = 0$ and $f(t) = g(t)$ for all $t \geq 0$. $\qquad\square$

**Lemma 11.** *Let $A, B \in \mathbb{R}^{d,d}$, and suppose $B$ is positive semidefinite. Then,*
$$\mathrm{Tr}(A^\top B A) \geq \lambda_1(B) \cdot \sigma_d(A)^2 \,.$$

*Proof.* The matrix $A^\top B A$ is positive semidefinite since for all $\mathbf{y} \in \mathbb{R}^d$ we have:
$$\mathbf{y}^\top A^\top B A \mathbf{y} = (A\mathbf{y})^\top B (A\mathbf{y}) \geq 0 \,.$$

Therefore, $\mathrm{Tr}(A^\top B A) \geq \lambda_1(A^\top B A)$. Let $B = ODO^\top$ be an orthogonal eigenvalue decomposition of $B$, *i.e.* $O \in \mathbb{R}^{d,d}$ is an orthogonal matrix with columns $\{\mathbf{o}_i\}_{i=1}^d$ and $D \in \mathbb{R}^{d,d}$ is diagonal holding the non-negative eigenvalues of $B$. Additionally, let $A = U \Sigma V^\top$ be a singular value decomposition of $A$, where $U, V \in \mathbb{R}^{d,d}$ are orthogonal matrices, and $\Sigma \in \mathbb{R}_{\geq 0}^{d,d}$ is diagonal holding the singular values of $A$. For any unit vector (with respect to the $\ell_2$ norm) $\mathbf{y} \in \mathbb{R}^d$ it holds that:

$$\mathbf{y}^\top A^\top B A \mathbf{y} = \sum_{i=1}^d \lambda_i(B)(\mathbf{o}_i^\top A \mathbf{y})^2 \geq \lambda_1(B)(\mathbf{o}_1^\top A \mathbf{y})^2 \,.$$

Replacing $A$ with its singular value decomposition and choosing $\mathbf{y} = V U^\top \mathbf{o}_1$:

$$\lambda_1(B)(\mathbf{o}_1^\top A \mathbf{y})^2 = \lambda_1(B)(\mathbf{o}_1^\top U \Sigma U^\top \mathbf{o}_1)^2 \,.$$

Recalling that for any unit vector the quadratic form of a symmetric matrix is bounded by the maximal and minimal eigenvalues completes the proof:

$$\mathrm{Tr}(A^\top B A) \geq \lambda_1(A^\top B A) \geq \lambda_1(B)(\mathbf{o}_1^\top U \Sigma U^\top \mathbf{o}_1)^2 \geq \lambda_1(B) \cdot \sigma_d(A)^2 \,.$$

$\qquad\square$

**Lemma 12.** *For any $\{A_l \in \mathbb{R}^{d_l, d_{l-1}}\}_{l=1}^L$ and $\{B_l \in \mathbb{R}^{d_l, d_{l-1}}\}_{l=1}^L$, where $d_0, d_1, \ldots, d_L \in \mathbb{N}$, it holds that:*

$$\left\| \prod_1^{l=L} A_l - \prod_1^{l=L} B_l \right\|_F \leq \sum_{l=1}^L \|A_l - B_l\|_F \cdot \prod_{r \neq l} \max\{\|A_r\|_F, \|B_r\|_F\} \,.$$

*Proof.* The proof is by induction over $L \in \mathbb{N}$. For $L = 1$, the claim is trivial. Assuming it holds for $L - 1 \geq 1$, we prove that it holds for $L$ as well:

$$\left\| \prod_1^{l=L} A_l - \prod_1^{l=L} B_l \right\|_F = \left\| \prod_1^{l=L} A_l - B_L \cdot \prod_1^{l=L-1} A_l + B_L \cdot \prod_1^{l=L-1} A_l - \prod_1^{l=L} B_l \right\|_F$$

$$\leq \|A_L - B_L\|_F \cdot \left\| \prod_1^{l=L-1} A_l \right\|_F + \|B_L\|_F \cdot \left\| \prod_1^{l=L-1} A_l - \prod_1^{l=L-1} B_l \right\|_F$$

$$\leq \|A_L - B_L\|_F \cdot \prod_{r \neq L} \max\{\|A_r\|_F, \|B_r\|_F\}$$

$$+ \max\{\|A_L\|_F, \|B_L\|_F\} \cdot \left\| \prod_1^{l=L-1} A_l - \prod_1^{l=L-1} B_l \right\|_F \,,$$

The proof concludes by applying the inductive assumption for $L - 1$. $\qquad\square$

**Lemma 13.** *For $B \in \mathbb{R}^{d,k}, A \in \mathbb{R}^{k,d'}$ such that $B^\top B = AA^\top$, let $U_A \Sigma_A V_A^\top$ be a singular value decomposition of $A$, i.e. $U_A \in \mathbb{R}^{k,k}, V_A \in \mathbb{R}^{d',d'}$ are orthogonal matrices, and $\Sigma_A \in \mathbb{R}^{k,d'}_{\geq 0}$ is rectangular-diagonal holding the singular values of $A$ in non-increasing order. Then, there exist an orthogonal $U_B \in \mathbb{R}^{d,d}$ and a rectangular-diagonal $\Sigma_B \in \mathbb{R}^{d,k}_{\geq 0}$ such that:*

- *$(\Sigma_B)_{i,i} = (\Sigma_A)_{i,i}$ for any $i \in [\min\{d, k, d'\}]$, with the remaining diagonal entries of $\Sigma_A$ and $\Sigma_B$ being $0$; and*

- *$B = U_B \Sigma_B U_A^\top$, i.e. $U_B \Sigma_B U_A^\top$ is a singular value decomposition of $B$.*

*Proof.* For any matrix $W$ it holds that $\mathrm{rank}(W) = \mathrm{rank}(WW^\top) = \mathrm{rank}(W^\top W)$. Therefore, the fact that $B^\top B = AA^\top$ implies $\mathrm{rank}(A) = \mathrm{rank}(B)$. Denote $r := \mathrm{rank}(A)$. For $i \in [k]$, let $\mathbf{u}_A^{(i)} \in \mathbb{R}^k$ be the $i$'th column vector of $U_A$. Furthermore, for $i \in [r]$ define $\mathbf{u}_B^{(i)} := \frac{1}{(\Sigma_A)_{i,i}} B \mathbf{u}_A^{(i)} \in \mathbb{R}^d$. We claim that $\mathbf{u}_B^{(1)}, \mathbf{u}_B^{(2)}, \ldots, \mathbf{u}_B^{(r)}$ form an orthonormal set. Indeed, for any $i, j \in [r]$:

$$\left\langle \mathbf{u}_B^{(i)}, \mathbf{u}_B^{(j)} \right\rangle = \frac{1}{(\Sigma_A)_{i,i}(\Sigma_A)_{j,j}} \left(\mathbf{u}_A^{(i)}\right)^\top B^\top B \mathbf{u}_A^{(j)} = \frac{(\Sigma_A)_{j,j}}{(\Sigma_A)_{i,i}} \left(\mathbf{u}_A^{(i)}\right)^\top \mathbf{u}_A^{(j)} = \begin{cases} 1 & , \text{if } i = j \\ 0 & , \text{if } i \neq j \end{cases},$$

where the second equality follows from $B^\top B \mathbf{u}_A^{(j)} = AA^\top \mathbf{u}_A^{(j)} = U_A \Sigma_A \Sigma_A^\top U_A^\top \mathbf{u}_A^{(j)} = (\Sigma_A)^2_{j,j} \mathbf{u}_A^{(j)}$. If $r < d$, we let $\mathbf{u}_B^{(r+1)}, \mathbf{u}_B^{(r+2)}, \ldots, \mathbf{u}_B^{(d)}$ be an arbitrary completion of $\mathbf{u}_B^{(1)}, \mathbf{u}_B^{(2)}, \ldots, \mathbf{u}_B^{(r)}$ to an orthonormal basis for $\mathbb{R}^d$. Define $U_B \in \mathbb{R}^{d,d}$ such that its $i$'th column vector is $\mathbf{u}_B^{(i)}$, and let $\Sigma_B \in \mathbb{R}^{d,k}_{\geq 0}$ be rectangular-diagonal with $(\Sigma_B)_{i,i} = (\Sigma_A)_{i,i}$ for $i \in [r]$, and $(\Sigma_B)_{i,i} = 0$ for $i > r$. Since both $U_B$ and $U_A$ are orthogonal matrices, the proof concludes by showing that $B = U_B \Sigma_B U_A^\top$. By the definitions of $U_B$ and $\Sigma_B$, we may write:

$$U_B \Sigma_B U_A^\top = \sum_{i=1}^r (\Sigma_A)_{i,i} \mathbf{u}_B^{(i)} \left(\mathbf{u}_A^{(i)}\right)^\top = B \sum_{i=1}^r \mathbf{u}_A^{(i)} \left(\mathbf{u}_A^{(i)}\right)^\top.$$

Notice that $U_A \Sigma_A \Sigma_A^\top U_A^\top$ is an eigenvalue decomposition of $AA^\top$, and hence of $B^\top B$. Therefore, $\ker(B) = \ker(B^\top B) = \mathrm{span}\{\mathbf{u}_A^{(r+1)}, \mathbf{u}_A^{(r+2)}, \ldots, \mathbf{u}_A^{(k)}\}$. With this in hand, for any vector $\mathbf{x} \in \mathbb{R}^k$ we have that:

$$B\mathbf{x} = BU_A U_A^\top \mathbf{x} = B \sum_{i=1}^r \mathbf{u}_A^{(i)} \left(\mathbf{u}_A^{(i)}\right)^\top \mathbf{x}.$$

Therefore, we may conclude $B = B \sum_{i=1}^r \mathbf{u}_A^{(i)} \left(\mathbf{u}_A^{(i)}\right)^\top = U_B \Sigma_B U_A^\top$. $\qquad\square$

**Lemma 14.** *Let $g : [0, \infty) \to \mathbb{R}$ be a continuously differentiable function, and fix some $t > 0$. If $g(t) < g(0)$, then for any $a \in (g(t), g(0)]$ there exists $t_a \in [0, t)$ such that $g(t_a) = a$ and $\dot{g}(t_a) \leq 0$. Similarly, if $g(t) > g(0)$, then for any $a \in [g(0), g(t))$ there exists $t_a \in [0, t)$ such that $g(t_a) = a$ and $\dot{g}(t_a) \geq 0$.*

*Proof.* Let $t > 0$ be such that $g(t) < g(0)$, and fix some $a \in (g(t), g(0)]$. Define $t_a := \max\{t' : t' \leq t$ and $g(t') = a\}$. Continuity of $g(\cdot)$, along with the intermediate value theorem, imply that $t_a$ is well defined (maximum of a closed non-empty set bounded from above). Assume by contradiction that $\dot{g}(t_a) > 0$. Then, $g(\cdot)$ is monotonically increasing on some neighborhood of $t_a$. Thus, by the intermediate value theorem, there exists $t' \in (t_a, t)$ such that $g(t') = a$, in contradiction to the definition of $t_a$. An identical argument establishes the analogous result for the case $g(t) > g(0)$. $\quad\square$

**Lemma 15.** *Let $g : [0, \infty) \to \mathbb{R}$ be a non-negative differentiable function. Assume there exist constants $a, b > 0$ such that $\int_{t'=0}^t g(t')dt' \leq a$ and $\dot{g}(t) \leq b$ for all $t \geq 0$. Then, $\lim_{t \to \infty} g(t) = 0$.*

*Proof.* By way of contradiction let us assume that $g(t)$ does not converge to $0$. Let $\epsilon > 0$ be such that for all $M > 0$ there exists $t > M$ with $g(t) > \epsilon$.

We claim that for all $M, \epsilon' > 0$ there exists $t > M$ such that $g(t) < \epsilon'$. Otherwise, we have a contradiction to the bound on the integral of $g(\cdot)$. Combined with our assumption, this means that for

all $M > 0$ we can find an interval $[t_1, t_2]$, with $t_1 > M$, where $g(t)$ transitions from $\frac{\epsilon}{2}$ to $\epsilon$. We now examine one such interval. Formally, for $t_0$ with $g(t_0) < \frac{\epsilon}{2}$, we define:

$$t_2 := \min\{t | t \geq t_0 \text{ and } g(t) = \epsilon\} \quad, \quad t_1 := \max\{t | t \leq t_2 \text{ and } g(t) = \epsilon/2\}.$$

Due to the fact that $g(\cdot)$ is continuous, $t_2$ and $t_1$ are well defined as they are the minimum and maximum, respectively, of closed non-empty sets bounded from below and above, respectively. Furthermore, notice that $t_0 < t_1 < t_2$. From the mean value theorem and the bound on the derivative of $g(\cdot)$ we have $t_2 - t_1 \geq \epsilon/2b$. Since $g(t) \geq \epsilon/2$ over the interval $[t_1, t_2]$, this gives us $\int_{t'=t_1}^{t_2} g(t')dt' \geq \epsilon^2/4b$. Recall there are infinitely many such occurrences, implying that $\int_{t'=0}^{\infty} g(t')dt' = \infty$, in contradiction to the bound on the integral. $\qquad\square$

**Lemma 16.** *Let $t_0 > 0$ and $g : [t_0, \infty) \to \mathbb{R}$ be a continuously differentiable function such that there exists $T > t_0$ for which $g(t)$ is positive over $[t_0, T)$. Assume that $|\dot{g}(t)| \leq a + b \cdot g(t)^\gamma$ for all $t \in [t_0, T]$, with $a, b > 0$, and $\gamma \geq 1$. Then:*

- *If $\gamma = 1$:*

$$g(T) \geq \frac{a + b \cdot g(t_0)}{b} \cdot e^{-b(T-t_0)} - \frac{a}{b}.$$

- *Otherwise, if $\gamma > 1$:*

$$g(T) \geq \frac{1}{b^{1/\gamma}\left[b^{1/\gamma}(\gamma - 1)(T - t_0) + \left(a^{1/\gamma} + b^{1/\gamma} \cdot g(t_0)\right)^{1-\gamma}\right]^{1/(\gamma-1)}} - \left(\frac{a}{b}\right)^{1/\gamma}.$$

*Proof.* Since $g(\cdot)$ is continuous over $[t_0, \infty)$, and positive over $[t_0, T)$, it is non-negative at $T$. In the case where $\gamma = 1$, we have that $\dot{g}(t) \geq -a - b \cdot g(t)$. Dividing both sides by $a + b \cdot g(t)$ (positive since $a, b > 0$ and $g(t) \geq 0$), and integrating over $t \in [t_0, T]$, we have that:

$$\frac{1}{b}\left[\ln\left(a + b \cdot g(T)\right) - \ln\left(a + b \cdot g(t_0)\right)\right] = \int_{t=t_0}^{T} \frac{\dot{g}(t)}{a + b \cdot g(t)} dt \geq -(T - t_0).$$

The lower bound on $g(T)$ readily follows by rearranging the inequality above.

Suppose $\gamma > 1$. We begin by showing that $a + b \cdot g(t)^\gamma \leq (a^{1/\gamma} + b^{1/\gamma} \cdot g(t))^\gamma$ whenever $g(t) \geq 0$. To see it is so, notice that for any $\mathbf{x} := (x_1, \ldots, x_d) \in \mathbb{R}_{\geq 0}^d$ it holds that $\sum_{i=1}^{d} x_i^\gamma \leq (\sum_{i=1}^{d} x_i)^\gamma$. This is directly implied from the following norm inequality: $\|\mathbf{x}\|_\gamma := (\sum_{i=1}^{d} x_i^\gamma)^{1/\gamma} \leq \|\mathbf{x}\|_1 := \sum_{i=1}^{d} x_i$. Thus, for $t \in [t_0, T]$ it holds that $|\dot{g}(t)| \leq (a^{1/\gamma} + b^{1/\gamma} \cdot g(t))^\gamma$, and in particular:

$$\dot{g}(t) \geq -\left(a^{1/\gamma} + b^{1/\gamma} \cdot g(t)\right)^\gamma.$$

Dividing by $(a^{1/\gamma} + b^{1/\gamma} \cdot g(t))^\gamma$ (positive since $a, b > 0$ and $g(t) \geq 0$), and integrating over $t \in [t_0, T]$:

$$\int_{t=t_0}^{T} \frac{\dot{g}(t)}{\left(a^{1/\gamma} + b^{1/\gamma} \cdot g(t)\right)^\gamma} dt \geq -(T - t_0)$$

$$\implies \frac{1}{b^{1/\gamma}(1 - \gamma)}\left[\left(a^{1/\gamma} + b^{1/\gamma} \cdot g(T)\right)^{1-\gamma} - \left(a^{1/\gamma} + b^{1/\gamma} \cdot g(t_0)\right)^{1-\gamma}\right] \geq -(T - t_0).$$

Rearranging the inequality above establishes the desired result. $\qquad\square$

**Lemma 17.** *Let $\mathcal{D}$ be an open domain in Euclidean space, and let $f : \mathcal{D} \to \mathbb{R}$ be a continuously differentiable function. Let $\theta : [0, \infty) \to \mathcal{D}$ be a curve born from gradient flow over $f(\cdot)$:*

$$\theta(0) = \theta_0 \in \mathcal{D} \quad, \quad \dot{\theta}(t) := \frac{d}{dt}\theta(t) = -\nabla f(\theta(t)) \ , \ t \geq 0.$$

*Then, $f(\theta(t))$ is monotonically non-increasing with respect to $t$.*

*Proof.* The proof immediately follows from the fact that for any $t \in [0, \infty)$:

$$\frac{d}{dt}f(\theta(t)) = \left\langle \nabla f(\theta(t)), \dot{\theta}(t) \right\rangle = \langle \nabla f(\theta(t)), -\nabla f(\theta(t)) \rangle = -\|\nabla f(\theta(t))\|_2^2 \leq 0.$$

$\qquad\square$

### G.3 Proof of Proposition 1

For a quasi-norm $\|\cdot\|$, the weakened triangle inequality (see Footnote 2) implies that there exists a constant $c_{\|\cdot\|} \geq 1$ for which

$$\|W\| \geq \frac{1}{c_{\|\cdot\|}} \left\|(W)_{1,1}\mathbf{e}_1\mathbf{e}_1^\top\right\| - \left\|W - (W)_{1,1}\mathbf{e}_1\mathbf{e}_1^\top\right\|$$
$$= |(W)_{1,1}| \frac{\left\|\mathbf{e}_1\mathbf{e}_1^\top\right\|}{c_{\|\cdot\|}} - \left\|\mathbf{e}_2\mathbf{e}_1^\top + \mathbf{e}_1\mathbf{e}_3^\top\right\| , \tag{28}$$

for any $W \in \mathcal{S}$. Fix some $\epsilon > 0$ and define $M_{\|\cdot\|,\epsilon} := \{(W)_{1,1} \in \mathbb{R} : \|W\| \leq \inf_{W' \in \mathcal{S}} \|W'\| + \epsilon, \ W \in \mathcal{S}\}$, the set of $(W)_{1,1}$ values corresponding to $\epsilon$-minimizers of $\|\cdot\|$. The first part of the proposition thus boils down to showing $M_{\|\cdot\|,\epsilon}$ is bounded. By Equation (28), there exist a $C > 0$ such that $|(W)_{1,1}| > C$ means $\|W\| > \inf_{W' \in \mathcal{S}} \|W'\| + \epsilon$. Hence, $M_{\|\cdot\|,\epsilon} \subset I_{\|\cdot\|,\epsilon} := [-C, C]$.

If in addition $\|\cdot\|$ is a Schatten-$p$ quasi-norm for $p \in (0, \infty]$, we now show that $W$ is its minimizer over $\mathcal{S}$ if and only if $(W)_{1,1} = 0$. Let $W_x \in \mathcal{S}$ denote the solution matrix with $(W_x)_{1,1} = x$ for $x \in \mathbb{R}$. The singular values of an arbitrary such $W_x$ are:

$$\{\sigma_1(W_x), \sigma_2(W_x)\} = \left\{ \left|\left(x + \sqrt{x^2 + 4}\right)/2\right|, \left|\left(x - \sqrt{x^2 + 4}\right)/2\right| \right\}. \tag{29}$$

Starting with $p = \infty$, the corresponding norm is the spectral norm $\|W_x\|_{S_\infty} := \sigma_1(W_x)$. When $x = 0$, we have that $\sigma_1(W_0) = 1$. If $x > 0$, then $\sigma_1(W_x) = (x + \sqrt{x^2 + 4})/2 > 1$. Similarly, if $x < 0$, then $\sigma_1(W_x) = (-x + \sqrt{x^2 + 4})/2 > 1$. Therefore, $\|W_x\|_{S_\infty}$ attains its minimal value of 1 if and only if $x = 0$.

Moving to the case of $p \in (0, \infty)$, the corresponding quasi-norm is $\|W_x\|_{S_p} := (\sigma_1(W_x)^p + \sigma_2(W_x)^p)^{\frac{1}{p}}$. We now examine $\|W_x\|_{S_p}^p$ for $x > 0$:

$$\|W_x\|_{S_p}^p = \left(\frac{x + \sqrt{x^2 + 4}}{2}\right)^p + \left(\frac{-x + \sqrt{x^2 + 4}}{2}\right)^p.$$

Differentiating with respect to $x$, we arrive at:

$$\frac{p}{2^p}\left(\left(x + \sqrt{x^2+4}\right)^{p-1}\left(1 + \frac{x}{\sqrt{x^2+4}}\right) + \left(-x + \sqrt{x^2+4}\right)^{p-1}\left(-1 + \frac{x}{\sqrt{x^2+4}}\right)\right)$$
$$> \frac{p}{2^p}\left(\left(x + \sqrt{x^2+4}\right)^{p-1} - \left(-x + \sqrt{x^2+4}\right)^{p-1}\right)$$
$$> 0,$$

where in the first transition we used the fact that both $\left(x + \sqrt{x^2+4}\right)^{p-1}$ and $\left(-x + \sqrt{x^2+4}\right)^{p-1}$ are positive (as well as $x$). It then directly follows that $\|W_x\|_{S_p}^p$ and thus $\|W_x\|_{S_p}$ are monotonically increasing with respect to $x$ on $(0, \infty)$.

Similar arguments show that when $x < 0$ the Schatten-$p$ quasi-norm of $W_x$ is monotonically decreasing with respect to $x$, implying that $\|W_x\|_{S_p}$ is minimized if and only if $x = 0$.[20] $\qquad \square$

### G.4 Proof of Proposition 2

As in the proof of Proposition 1 (Subappendix G.3), we denote by $W_x \in \mathcal{S}$ the solution matrix with $(W_x)_{1,1} = x$. We begin by analyzing the behavior of $\sigma_1(W_x)$ and $\sigma_2(W_x)$ with respect to $x$. When $x = 0$ the singular values are simply $\sigma_1(W_0) = \sigma_2(W_0) = 1$. When $x$ is positive, the singular values may be written as:

$$\sigma_1(W_x) = \frac{x + \sqrt{x^2 + 4}}{2} \quad , \quad \sigma_2(W_x) = \frac{-x + \sqrt{x^2 + 4}}{2}.$$

Taking the derivative with respect to $x$, we arrive at:

$$\frac{d}{dx}\sigma_1(W_x) = \frac{1}{2} + \frac{x}{2\sqrt{x^2+4}} \quad , \quad \frac{d}{dx}\sigma_2(W_x) = -\frac{1}{2} + \frac{x}{2\sqrt{x^2+4}}\,.$$

Since $x > 0$, we have that $d/dx\,\sigma_1(W_x) > 0$ and $d/dx\,\sigma_2(W_x) < 0$. In other words, $\sigma_1(W_x)$ is monotonically increasing, while $\sigma_2(W_x)$ is monotonically decreasing, when $x > 0$. It can easily be verified that $\sigma_1(W_x)$ and $\sigma_2(W_x)$ are even functions of $x$, $i.e.$ $\sigma_1(W_x) = \sigma_1(W_{-x})$ and $\sigma_2(W_x) = \sigma_2(W_{-x})$. It then follows that $\sigma_1(W_x)$ is monotonically decreasing (conversely $\sigma_2(W_x)$ is monotonically increasing) when $x < 0$. Noticing that $\lim_{x\to\infty}\sigma_1(W_x) = \infty$ and $\lim_{x\to\infty}\sigma_2(W_x) = 0$ (accordingly $\lim_{x\to-\infty}\sigma_1(W_x) = \infty$ and $\lim_{x\to-\infty}\sigma_2(W_x) = 0$ ), we have a characterization of the behavior of $\sigma_1(W_x)$ and $\sigma_2(W_x)$.

We are now in a position to obtain the desired results for effective and infimal ranks. The effective rank (Definition 1) of $W_x$ can be written as

$$\mathrm{erank}(W_x) = \exp\left\{H\left(\frac{\sigma_1(W_x)}{\sigma_1(W_x)+\sigma_2(W_x)}, \frac{\sigma_2(W_x)}{\sigma_1(W_x)+\sigma_2(W_x)}\right)\right\}\,.$$

The binary entropy function is bounded by $\ln(2)$, hence, the effective rank over $\mathcal{S}$ is bounded by 2. This upper bound is attained at $x = 0$. According to the singular values analysis, when $|x| \to \infty$ we have that $\rho_1(W_x)$ monotonically increases towards 1, starting from the value $\rho_1(W_0) = \frac{1}{2}$. Noticing that this implies the entropy function and effective rank monotonically decrease towards 0 and 1, respectively, completes the effective rank analysis.

Next, we analyze the infimal rank of $\mathcal{S}$ and the distance of $W_x$ from that infimal rank. The distance of $W_x$ from $\mathcal{M}_1$ is $D(W_x, \mathcal{M}_1) = \sigma_2(W_x)$. Since $\lim_{x\to\infty}\sigma_2(W_x) = 0$, we have $D(\mathcal{S}, \mathcal{M}_1) = 0$. Clearly $D(\mathcal{S}, \mathcal{M}_0) > 0$, leading to the conclusion that the infimal rank of $\mathcal{S}$ is 1. Finally, the analysis of $\sigma_2(W_x)$ directly implies that the distance of $W_x$ from the infimal rank of $\mathcal{S}$ is maximized when $x = 0$, monotonically tending to 0 as $|x| \to \infty$. $\qquad\square$

### G.5 Proof of Theorem 1

In the following, as stated in Subappendix G.1, for results that hold for all $t \geq 0$ or when clear from the context, we omit the time index $t$. Furthermore, we denote the entries of the product matrix $W_{L:1}$ by $\{w_{i,j}\}_{i,j\in[2]}$.

We begin by deriving loss-dependent bounds for $|w_{1,1}|, \sigma_1(W_{L:1})$ and $\sigma_2(W_{L:1})$. Writing the loss explicitly:

$$\ell(W_{L:1}) = \frac{1}{2}\left[(w_{1,2}-1)^2 + (w_{2,1}-1)^2 + w_{2,2}^2\right]\,,$$

we can upper bound each of the non-negative terms separately. Multiplying by 2 and taking the square root of both sides yields:

$$|w_{2,2}| \leq \sqrt{2\ell(W_{L:1})} \quad , \quad |w_{1,2}-1| \leq \sqrt{2\ell(W_{L:1})} \quad , \quad |w_{2,1}-1| \leq \sqrt{2\ell(W_{L:1})}\,. \tag{30}$$

The following lemma characterizes the relation between $|w_{1,1}|$ and the loss.

**Lemma 18.** *Suppose $\ell(W_{L:1}) < \frac{1}{2}$. Then:*

$$|w_{1,1}| > \frac{(1-\sqrt{2\ell(W_{L:1})})^2}{\sqrt{2\ell(W_{L:1})}} = \frac{1}{\sqrt{2\ell(W_{L:1})}} - 2 + \sqrt{2\ell(W_{L:1})}\,.$$

*Proof.* From Lemma 6, the determinant of $W_{L:1}$ does not change signs and remains positive, $i.e.$:

$$\det(W_{L:1}) = w_{1,1}w_{2,2} - w_{1,2}w_{2,1} > 0\,. \tag{31}$$

Under the assumption that $\ell(W_{L:1}) < \frac{1}{2}$, both $w_{1,2}$ and $w_{2,1}$ are positive and lie inside the open interval $(0,2)$. Since the determinant is positive, $w_{2,2} \neq 0$ and $w_{1,1}w_{2,2} > 0$ must hold. Rearranging Equation (31), we may therefore write $|w_{1,1}w_{2,2}| > w_{1,2}w_{2,1}$. Dividing both sides by $|w_{2,2}|$ and applying the bounds from Equation (30) completes the proof:

$$|w_{1,1}| > \frac{(1-\sqrt{2\ell(W_{L:1})})^2}{\sqrt{2\ell(W_{L:1})}} = \frac{1}{\sqrt{2\ell(W_{L:1})}} - 2 + \sqrt{2\ell(W_{L:1})}\,.$$

$\qquad\square$

An immediate consequence of the lemma above is that decreasing the loss towards zero drives $|w_{1,1}|$ towards infinity.

With this bound in hand, Lemma 19 below establishes bounds on the singular values of $W_{L:1}$. In turn, they will allow us to obtain the necessary results for effective rank (Definition 1) and distance from infimal rank of $\mathcal{S}$ (Definition 2).

**Lemma 19.** *The singular values of $W_{L:1}$ fulfill:*

$$\sigma_1(W_{L:1}) \geq |w_{1,1}| - \sqrt{2\ell(W_{L:1})} \quad , \quad \sigma_2(W_{L:1}) \leq 3\sqrt{2\ell(W_{L:1})}. \tag{32}$$

*Furthermore, if $\ell(W_{L:1}) < \frac{1}{2}$, then:*

$$\sigma_1(W_{L:1}) \geq \frac{1}{\sqrt{2\ell(W_{L:1})}} - 2. \tag{33}$$

*Proof.* Define $W_{\mathcal{S}} := \begin{pmatrix} w_{1,1} & 1 \\ 1 & 0 \end{pmatrix}$, the orthogonal projection of $W_{L:1}$ onto the solution set $\mathcal{S}$. By Corollary 8.6.2 in [33] we have that:

$$|\sigma_i(W_{L:1}) - \sigma_i(W_{\mathcal{S}})| \leq \|W_{L:1} - W_{\mathcal{S}}\|_F = \sqrt{2\ell(W_{L:1})} \quad , \ i = 1, 2. \tag{34}$$

One can easily verify that $W_{\mathcal{S}}$ is a symmetric indefinite matrix with eigenvalues

$$\{\lambda_1(W_{\mathcal{S}}), \lambda_2(W_{\mathcal{S}})\} = \left\{ \left( w_{1,1} + \sqrt{w_{1,1}^2 + 4} \right) / 2 \,, \ \left( w_{1,1} - \sqrt{w_{1,1}^2 + 4} \right) / 2 \right\}.$$

Suppose that $w_{1,1} \geq 0$. We thus have:

$$\sigma_1(W_{\mathcal{S}}) = \max_{i=1,2} |\lambda_i(W_{\mathcal{S}})| = \frac{w_{1,1} + \sqrt{w_{1,1}^2 + 4}}{2} \geq |w_{1,1}|,$$

and

$$\begin{aligned}
\sigma_2(W_{\mathcal{S}}) &= \min_{i=1,2} |\lambda_i(W_{\mathcal{S}})| \\
&= \frac{\sqrt{w_{1,1}^2 + 4} - w_{1,1}}{2} \\
&= \frac{2}{\sqrt{w_{1,1}^2 + 4} + w_{1,1}} \\
&\leq \frac{2}{2 + w_{1,1}},
\end{aligned}$$

where in the third transition we made use of the identity $a - b = \frac{a^2 - b^2}{a+b}$ for $a, b \in \mathbb{R}$ such that $a + b \neq 0$. If $\ell(W_{L:1}) \geq \frac{1}{2}$, it holds that $\sigma_2(W_{\mathcal{S}}) \leq 2/(2 + w_{1,1}) \leq 1 \leq 2\sqrt{2\ell(W_{L:1})}$. Otherwise, we may apply the lower bound on $w_{1,1}$ (Lemma 18) and conclude that $\sigma_2(W_{\mathcal{S}}) \leq 2\sqrt{2\ell(W_{L:1})}$ for any loss value. Having established that $\sigma_1(W_{\mathcal{S}}) \geq |w_{1,1}|$ and $\sigma_2(W_{\mathcal{S}}) \leq 2\sqrt{2\ell(W_{L:1})}$, Equation (34) completes the proof of Equation (32). It remains to see that if $\ell(W_{L:1}) < \frac{1}{2}$, from the lower bound on $w_{1,1}$ (Lemma 18), Equation (33) immediately follows.

By similar arguments, Equations (32) and (33) hold for $w_{1,1} < 0$ as well. $\square$

### G.5.1 Proof of Equation (8) (lower bound for quasi-norm)

We turn to lower bound the quasi-norm of the product matrix. It holds that:

$$\|W_{L:1}\| \geq \frac{1}{c_{\|\cdot\|}} \left\| w_{1,1} \mathbf{e}_1 \mathbf{e}_1^\top \right\| - \left\| W_{L:1} - w_{1,1} \mathbf{e}_1 \mathbf{e}_1^\top \right\|, \tag{35}$$

where $c_{\|\cdot\|} \geq 1$ is a constant for which $\|\cdot\|$ satisfies the weakened triangle inequality (see Footnote 2). We now assume that $\ell(W_{L:1}) < \frac{1}{2}$. Later this assumption will be lifted, providing a bound that

holds for all loss values. Subsequent applications of the weakened triangle inequality, together with homogeneity of $\|\cdot\|$ and the bounds on the entries of $W_{L:1}$ (Equation (30)), give:

$$
\begin{aligned}
\left\|W_{L:1} - w_{1,1}\mathbf{e}_1\mathbf{e}_1^\top\right\| &\leq c_{\|\cdot\|}|w_{2,2}|\left\|\mathbf{e}_2\mathbf{e}_2^\top\right\| + c_{\|\cdot\|}^2\left(|w_{2,1}|\left\|\mathbf{e}_2\mathbf{e}_1^\top\right\| + |w_{1,2}|\left\|\mathbf{e}_1\mathbf{e}_2^\top\right\|\right) \\
&\leq c_{\|\cdot\|}\sqrt{2\ell(W_{L:1})}\left\|\mathbf{e}_2\mathbf{e}_2^\top\right\| + c_{\|\cdot\|}^2\left(1 + \sqrt{2\ell(W_{L:1})}\right)\left(\left\|\mathbf{e}_2\mathbf{e}_1^\top\right\| + \left\|\mathbf{e}_1\mathbf{e}_2^\top\right\|\right) \\
&\leq c_{\|\cdot\|}\left\|\mathbf{e}_2\mathbf{e}_2^\top\right\| + 2c_{\|\cdot\|}^2\left(\left\|\mathbf{e}_2\mathbf{e}_1^\top\right\| + \left\|\mathbf{e}_1\mathbf{e}_2^\top\right\|\right) \\
&\leq 2c_{\|\cdot\|}^2\left(\left\|\mathbf{e}_2\mathbf{e}_2^\top\right\| + \left\|\mathbf{e}_2\mathbf{e}_1^\top\right\| + \left\|\mathbf{e}_1\mathbf{e}_2^\top\right\|\right).
\end{aligned}
$$

Plugging the inequality above and the lower bound on $|w_{1,1}|$ (Lemma 18) into Equation (35), we have:

$$
\begin{aligned}
\|W_{L:1}\| &\geq \frac{\left\|\mathbf{e}_1\mathbf{e}_1^\top\right\|}{c_{\|\cdot\|}}\left(\frac{1}{\sqrt{2\ell(W_{L:1})}} - 2 + \sqrt{2\ell(W_{L:1})}\right) - 2c_{\|\cdot\|}^2\left(\left\|\mathbf{e}_2\mathbf{e}_2^\top\right\| + \left\|\mathbf{e}_2\mathbf{e}_1^\top\right\| + \left\|\mathbf{e}_1\mathbf{e}_2^\top\right\|\right) \\
&\geq \frac{\left\|\mathbf{e}_1\mathbf{e}_1^\top\right\|}{c_{\|\cdot\|}}\frac{1}{\sqrt{2\ell(W_{L:1})}} - 2\frac{\left\|\mathbf{e}_1\mathbf{e}_1^\top\right\|}{c_{\|\cdot\|}} - 2c_{\|\cdot\|}^2\left(\left\|\mathbf{e}_2\mathbf{e}_2^\top\right\| + \left\|\mathbf{e}_2\mathbf{e}_1^\top\right\| + \left\|\mathbf{e}_1\mathbf{e}_2^\top\right\|\right) \\
&\geq \frac{\left\|\mathbf{e}_1\mathbf{e}_1^\top\right\|}{c_{\|\cdot\|}}\frac{1}{\sqrt{2\ell(W_{L:1})}} - 2c_{\|\cdot\|}^2\left(\left\|\mathbf{e}_1\mathbf{e}_1^\top\right\| + \left\|\mathbf{e}_2\mathbf{e}_2^\top\right\| + \left\|\mathbf{e}_2\mathbf{e}_1^\top\right\| + \left\|\mathbf{e}_1\mathbf{e}_2^\top\right\|\right).
\end{aligned}
$$

Since $\|W_{L:1}\|$ is trivially lower bounded by zero, defining the constants

$$
a_{\|\cdot\|} := \frac{\left\|\mathbf{e}_1\mathbf{e}_1^\top\right\|}{\sqrt{2}c_{\|\cdot\|}} \ , \ b_{\|\cdot\|} := \max\left\{\sqrt{2}a_{\|\cdot\|}, 8c_{\|\cdot\|}^2\max_{i,j\in\{1,2\}}\left\|\mathbf{e}_i\mathbf{e}_j^\top\right\|\right\},
$$

allows us, on the one hand, to arrive at a bound of the form:

$$
\|W_{L:1}\| \geq a_{\|\cdot\|}\cdot\frac{1}{\sqrt{\ell(W_{L:1})}} - b_{\|\cdot\|},
$$

and on the other hand, to lift our previous assumption on the loss: when $\ell(W_{L:1}) \geq \frac{1}{2}$ the bound is vacuous, *i.e.* non-positive and trivially holds. Noticing this is exactly Equation (8) (recall we omitted the time index $t$), concludes the first part of the proof.

### G.5.2 Proof of Equation (9) (upper bound for effective rank)

During the following effective rank (Definition 1) analysis we operate under the assumption of $\ell(W_{L:1}) < \frac{1}{32}$. We later remove this assumption, delivering a bound that holds for all loss values. Making use of the obtained bounds on $\sigma_1(W_{L:1})$ and $\sigma_2(W_{L:1})$ (Lemma 19) we arrive at:

$$
\begin{aligned}
\rho_1(W_{L:1}) &= \frac{\sigma_1(W_{L:1})}{\sigma_1(W_{L:1}) + \sigma_2(W_{L:1})} \\
&\geq \frac{\sigma_1(W_{L:1})}{\sigma_1(W_{L:1}) + 3\sqrt{2\ell(W_{L:1})}} \\
&= 1 - \frac{3\sqrt{2\ell(W_{L:1})}}{\sigma_1(W_{L:1}) + 3\sqrt{2\ell(W_{L:1})}} \\
&\geq 1 - \frac{3\sqrt{2\ell(W_{L:1})}}{\frac{1}{\sqrt{2\ell(W_{L:1})}} - 2 + 3\sqrt{2\ell(W_{L:1})}} \\
&= 1 - \frac{6\ell(W_{L:1})}{6\ell(W_{L:1}) - 2\sqrt{2\ell(W_{L:1})} + 1}.
\end{aligned}
$$

Given our assumption on the loss, we have $1 - 2\sqrt{2\ell((W_{L:1}))} \geq \frac{1}{2}$ and thus

$$
\rho_2(W_{L:1}) = 1 - \rho_1(W_{L:1}) \leq \frac{6\ell(W_{L:1})}{6\ell(W_{L:1}) + \frac{1}{2}} \leq 12\ell(W_{L:1}). \tag{36}
$$

Let $h\left(\rho_2(W_{L:1})\right) := -\rho_2(W_{L:1})\cdot\ln\left(\rho_2(W_{L:1})\right) - (1 - \rho_2(W_{L:1}))\cdot\ln\left(1 - \rho_2(W_{L:1})\right)$ denote the binary entropy function, and recall that the effective rank of $W_{L:1}$ is defined to be $\mathrm{erank}(W_{L:1}) :=$

$\exp\{h\left(\rho_2(W_{L:1})\right)\}$. The exponent function is convex and therefore upper bounded on the interval $[0, \ln(2)]$ by the linear function that intersects it at these points. Formally, for $x \in [0, \ln(2)]$ it holds that $\exp(x) \leq 1 + \frac{1}{\ln(2)}x$, yielding the following bound:

$$\mathrm{erank}(W_{L:1}) \leq 1 + \frac{1}{\ln(2)} \cdot h\left(\rho_2(W_{L:1})\right).$$

By Lemma 9 we have that $h\left(\rho_2(W_{L:1})\right) \leq 2\sqrt{\rho_2(W_{L:1})}$. Combined with Equation (36), since $\inf_{W' \in \mathcal{S}} \mathrm{erank}(W') = 1$ (Proposition 2), this leads to:

$$\mathrm{erank}(W_{L:1}) \leq \inf_{W' \in \mathcal{S}} \mathrm{erank}(W') + \frac{2\sqrt{12}}{\ln(2)} \cdot \sqrt{\ell(W_{L:1})}.$$

Recall that the time index $t$ is omitted, and the result holds for all $t \geq 0$, *i.e.* this is exactly Equation (9). To remove our assumption on the loss, notice that when $\ell(W_{L:1}) \geq \frac{1}{32}$ the bound is trivial, as the right-hand side is greater than 2, which is the maximal effective rank (for a $2 \times 2$ matrix).

### G.5.3 Proof of Equation (10) (upper bound for distance from infimal rank)

According to Proposition 2, the infimal rank of $\mathcal{S}$ is 1. The quantity we seek to upper bound is therefore $D(W_{L:1}(t), \mathcal{M}_1) = \sigma_2(W_{L:1}(t))$. By Equation (32) in Lemma 19, for all $t \geq 0$ we have

$$D(W_{L:1}(t), \mathcal{M}_1) \leq 3\sqrt{2} \cdot \sqrt{\ell(t)},$$

completing the proof. □

### G.6 Proof of Proposition 3

Define $W_{-1}$ to be the matrix obtained from $W$ by multiplying its first row by $-1$. On the one hand, symmetry around the origin implies that $W_{-1}$ and $W$ follow the same distribution. On the other hand, $\det(W_{-1}) = -\det(W)$. Due to the fact that the set of matrices with zero determinant has probability 0 under continuous distributions (see, *e.g.*, Remark 2.5 in [37]), we may conclude $\Pr(\det(W) > 0) = \Pr(\det(W) < 0) = 0.5$.

For $W_1, W_2, \ldots, W_L$ random matrices drawn independently, let $l \in [L]$ be the index such that $\Pr(\det(W_l) > 0) = 0.5$. Since $Pr(\det(W_{l'}) = 0) = 0$ for any $l' \in [L]$, the proof readily follows from determinant multiplicativity and the law of total probability:

$$\begin{aligned}
\Pr\left(\det(W_L W_{L-1} \cdots W_1) > 0\right) &= \Pr(\det(W_l) > 0) \cdot \Pr\left(\Pi_{i \neq l} \det(W_i) > 0\right) \\
&\quad + \Pr(\det(W_l) < 0) \cdot \Pr\left(\Pi_{i \neq l} \det(W_i) < 0\right) \\
&= \frac{1}{2}\left[\Pr\left(\Pi_{i \neq l} \det(W_i) > 0\right) + \Pr\left(\Pi_{i \neq l} \det(W_i) < 0\right)\right] \\
&= 0.5.
\end{aligned}$$

An identical computation yields $\Pr\left(\det(W_L W_{L-1} \cdots W_1) < 0\right) = 0.5$. □

### G.7 Proof of Proposition 4

The proof makes use of the following lemma, proven in Subappendix G.7.1.

**Lemma 20.** *Consider the setting of Theorem 1 (arbitrary depth $L \in \mathbb{N}$) in the special case of an initial product matrix $W_{L:1}(0) = \alpha \cdot I$, where $I$ stands for identity matrix and $\alpha \in (0, 1]$. Then, $W_{L:1}(t)$ is positive definite for all $t \geq 0$.*

With Lemma 20 in place, we may derive the exact differential equations governing the product matrix in our setting of depth $L = 2$. Then, a detailed analysis of the dynamics will yield convergence of the loss to global minimum, *i.e.* $\lim_{t \to \infty} \ell(t) = 0$. As usual, we omit the time index $t$ when stating results for all $t$ or when clear from the context.

According to Lemma 20, the product matrix $W_{L:1}$ is symmetric and positive definite. Thus, we may write the loss and its gradient with respect to $W_{L:1}$ as:

$$\ell(W_{L:1}) = \frac{1}{2}\left[w_{2,2}^2 + 2(w_{1,2} - 1)^2\right] \quad , \quad \nabla\ell(W_{L:1}) = \begin{pmatrix} 0 & w_{1,2} - 1 \\ w_{1,2} - 1 & w_{2,2} \end{pmatrix}, \quad (37)$$

where $\{w_{i,j}\}_{i,j \in [2]}$ are the entries of $W_{L:1}$. Since the factors $W_1$ and $W_2$ are balanced at initialization (Equation (5)), the differential equation governing the product matrix (Lemma 4) for depth $L = 2$

gives:

$$
\begin{aligned}
\dot{W}_{L:1} &= -\left[W_{L:1}W_{L:1}^\top\right]^{\frac{1}{2}} \cdot \nabla\ell(W_{L:1}) - \nabla\ell(W_{L:1}) \cdot \left[W_{L:1}^\top W_{L:1}\right]^{\frac{1}{2}} \\
&= -W_{L:1}\nabla\ell(W_{L:1}) - \nabla\ell(W_{L:1})W_{L:1}\,,
\end{aligned}
\tag{38}
$$

where the transition is by positive definiteness of $W_{L:1}$. Writing the differential equation of each entry separately, we have:

$$
\begin{aligned}
\dot{w}_{1,1} &= 2w_{1,2}(1 - w_{1,2})\,, \\
\dot{w}_{2,2} &= 2w_{1,2}(1 - w_{1,2}) - 2w_{2,2}^2\,, \\
\dot{w}_{1,2} &= w_{2,2}(1 - 2w_{1,2}) + w_{1,1}(1 - w_{1,2})\,.
\end{aligned}
\tag{39}
$$

Let us characterize the behavior of these entries throughout time.

**Lemma 21.** *The following holds for all $t \geq 0$:*

1. *$w_{1,1} > 0$ and is monotonically non-decreasing.*

2. *$0 \leq w_{1,2} \leq 1$.*

3. *$0 < w_{2,2} \leq 1$.*

*Proof.* Since $W_{L:1}$ is positive definite, it follows that $w_{1,1}$ and $w_{2,2}$ are positive. Examining the behavior of $w_{1,2}$ (Equation (39)): on the one hand, when $w_{1,2} = 0$ then $\dot{w}_{1,2} = w_{2,2} + w_{1,1} > 0$, and on the other hand, when $w_{1,2} = 1$ then $\dot{w}_{1,2} = -w_{2,2} < 0$. Because $w_{1,2}$ is initialized at $0$, it stays in the interval $[0, 1]$. Otherwise, by Lemma 14, we have a contradiction to the positivity of $\dot{w}_{1,2}$ when $w_{1,2} = 0$ or its negativity when $w_{1,2} = 1$. Similarly, if $w_{2,2} > \frac{1}{2}$ we have $\dot{w}_{2,2} < 2w_{1,2}(1 - w_{1,2}) - \frac{1}{2} \leq 0$. Since at initialization $w_{2,2}(0) = \alpha \leq 1$, by Lemma 14, it will not go above $1$. Lastly, since $w_{1,2}$ is in the interval $[0, 1]$, it holds that $\dot{w}_{1,1} \geq 0$, *i.e.* $w_{1,1}$ is monotonically non-decreasing. $\qquad\square$

We turn our focus to the derivative of the loss with respect to $t$:

$$
\frac{d}{dt}\ell(W_{L:1}) = \langle \nabla\ell(W_{L:1}), \dot{W}_{L:1} \rangle\,.
$$

Plugging in Equation (38) and recalling the fact that $\langle A, B \rangle = \mathrm{Tr}(A^\top B)$ for matrices $A, B$ of the same size:

$$
\frac{d}{dt}\ell(W_{L:1}) = -\mathrm{Tr}(\nabla\ell(W_{L:1})^\top W_{L:1}\nabla\ell(W_{L:1})) - \mathrm{Tr}(\nabla\ell(W_{L:1})^\top \nabla\ell(W_{L:1})W_{L:1})\,.
$$

From the cyclic property of the trace operator and symmetry of $\nabla\ell(W_{L:1})$ (Equation (37)), we arrive at the following expression:

$$
\frac{d}{dt}\ell(W_{L:1}) = -2\,\mathrm{Tr}(\nabla\ell(W_{L:1})W_{L:1}\nabla\ell(W_{L:1}))\,.
$$

Notice that since $\nabla\ell(W_{L:1})W_{L:1}\nabla\ell(W_{L:1})$ is positive semidefinite the trace is non-negative and $\frac{d}{dt}\ell(W_{L:1}) \leq 0$. That is, the loss is monotonically non-increasing throughout time. Invoking Lemma 11, we can upper bound the derivative by:

$$
\frac{d}{dt}\ell(W_{L:1}) \leq -2\lambda_1(W_{L:1}) \cdot \sigma_2(\nabla\ell(W_{L:1}))^2\,,
\tag{40}
$$

where $\lambda_1(W_{L:1})$ is the maximal eigenvalue of $W_{L:1}$ and $\sigma_2(\nabla\ell(W_{L:1}))$ is the minimal singular value of $\nabla\ell(W_{L:1})$. The maximal eigenvalue of a symmetric matrix is greater than its diagonal entries. Therefore, $\lambda_1(W_{L:1}) \geq w_{1,1}$. Since $w_{1,1}$ is initialized at $\alpha > 0$, and by Lemma 21 is monotonically non-decreasing, we have $\lambda_1(W_{L:1}) \geq \alpha$. Writing the eigenvalues of $\nabla\ell(W_{L:1})$ explicitly:

$$
\lambda_1(\nabla\ell(W_{L:1})) = \frac{w_{2,2} + \sqrt{w_{2,2}^2 + 4(1 - w_{1,2})^2}}{2}\,,
$$

$$
\lambda_2(\nabla\ell(W_{L:1})) = \frac{w_{2,2} - \sqrt{w_{2,2}^2 + 4(1 - w_{1,2})^2}}{2}\,,
$$

we can see that, since $w_{2,2}$ is positive (Lemma 21), $\sigma_2(\nabla \ell(W_{L:1})) = \min_{i=1,2} |\lambda_i(\nabla \ell(W_{L:1}))| = (\sqrt{w_{2,2}^2 + 4(1 - w_{1,2})^2} - w_{2,2})/2$. Applying the identity $a - b = \frac{a^2 - b^2}{a+b}$ for $a, b \in \mathbb{R}$ such that $a + b \neq 0$, and the bounds on $w_{2,2}$ and $w_{1,2}$ (Lemma 21):

$$
\begin{aligned}
\sigma_2(\nabla \ell(W_{L:1})) &= \frac{2(1 - w_{1,2})^2}{\sqrt{w_{2,2}^2 + 4(1 - w_{1,2})^2} + w_{2,2}} \\
&\geq \frac{2(1 - w_{1,2})^2}{\sqrt{1 + 4(1 - w_{1,2})^2} + 1} \\
&\geq \frac{2(1 - w_{1,2})^2}{2\,|(1 - w_{1,2})| + 2} \\
&\geq \frac{1}{2}(1 - w_{1,2})^2,
\end{aligned}
$$

where in the penultimate transition we bounded the square root of a sum by the sum of square roots. Returning to Equation (40) we have:

$$
\frac{d}{dt} \ell(W_{L:1}) \leq -b(1 - w_{1,2})^4,
$$

for $b = \frac{1}{2}\alpha$. We are now in a position to prove that $w_{1,2} \to 1$ as $t$ tends to infinity. Integrating both sides with respect to time:

$$
\ell(W_{L:1}(t)) - \ell(W_{L:1}(0)) \leq -b \int_{t'=0}^{t} (1 - w_{1,2}(t'))^4 dt'.
$$

Since $\ell(W_{L:1}(t)) \geq 0$, by rearranging the inequality we may write:

$$
\int_{t'=0}^{t} (1 - w_{1,2}(t'))^4 dt' \leq \frac{\ell(W_{L:1}(0))}{b}.
$$

Going back to the differential equation of $\dot{w}_{1,2}$ (Equation (39)), by applying the bounds on $w_{1,2}$ and $w_{2,2}$ (Lemma 21) we have that $\dot{w}_{1,2} \geq -1$. Defining $g(t) := (1 - w_{1,2}(t))^4$, it then holds that $\dot{g}(t) \leq 4$. Since $g(\cdot)$ is non-negative and has an upper bounded integral and derivative, from Lemma 15, we can conclude that $\lim_{t \to \infty} g(t) = 0$ and $\lim_{t \to \infty} w_{1,2}(t) = 1$.

Because $\ell(W_{L:1}(t))$ is monotonically non-increasing, we need only show that for each $\epsilon > 0$ there exists a $t_\epsilon > 0$ such that $\ell(W_{L:1}(t_\epsilon)) < \epsilon$. Having already established that $w_{1,2}(t)$ converges to 1, this amounts to finding a large enough $t_\epsilon$ for which $w_{2,2}(t_\epsilon)$ is sufficiently close to 0. Fix some $\epsilon > 0$ and let $\hat{t} > 0$ be such that for all $t \geq \hat{t}$ the following holds:

$$
2(1 - w_{1,2}(t))^2 < \epsilon \quad , \quad 2w_{1,2}(t)(1 - w_{1,2}(t)) < \epsilon. \tag{41}
$$

Such $\hat{t}$ exists since all terms above converge to 0. Returning to the differential equation of $\dot{w}_{2,2}$ (Equation (39)):

$$
\dot{w}_{2,2}(t) < \epsilon - 2w_{2,2}(t)^2. \tag{42}
$$

Recalling that $w_{2,2}(t) > 0$ (Lemma 21), it follows that there exists $t_\epsilon \geq \hat{t}$ with $\dot{w}_{2,2}(t_\epsilon) > -\epsilon$ (otherwise $w_{2,2}(t)$ goes to $-\infty$ as $t \to \infty$, in contradiction to the positivity of $w_{2,2}(t)$). For the above $t_\epsilon$, by rearranging the terms in Equation (42) we achieve $w_{2,2}(t_\epsilon) < \sqrt{\epsilon}$. Finally, combined with Equation (41), the result readily follows:

$$
\ell(W_{L:1}(t_\epsilon)) = \frac{1}{2} \left[ w_{2,2}(t_\epsilon)^2 + 2(w_{1,2}(t_\epsilon) - 1)^2 \right] < \epsilon,
$$

concluding the proof. $\qquad \square$

### G.7.1 Proof of Lemma 20

The proof proceeds as follows. We initially consider initializations where $W_1(0), \ldots, W_L(0)$ form a *symmetric factorization* of $W_{L:1}(0)$ (Definition 4), and show that this ensures the product matrix stays symmetric. Then, we establish that for every balanced initial factors (Equation (5)) with a positive definite product matrix there exist alternative balanced factors such that: *(i)* the initial product matrix is the same; and *(ii)* the factors form a symmetric factorization of the product matrix. Since the product matrices for the original and the constructed initializations obey the exact same dynamics (Lemma 4), the proof concludes.

**Definition 4.** We say that the matrices $W_1, W_2, \ldots, W_L \in \mathbb{R}^{d,d}$ form a *symmetric factorization* of $W \in \mathbb{R}^{d,d}$ if $W = W_L W_{L-1} \cdots W_1$ and

$$W_l = W_{L-l+1}^\top \quad , l \in \{1, \ldots, \lfloor L/2 \rfloor + 1\}.$$

A straightforward result is that matrices with a symmetric factorization are symmetric themselves.

**Lemma 22.** *If a matrix $W \in \mathbb{R}^{d,d}$ has a symmetric factorization, then it is symmetric.*

*Proof.* Let $W_1, W_2, \ldots, W_L \in \mathbb{R}^{d,d}$ form a symmetric factorization of $W$. It directly follows that

$$W = W_L W_{L-1} \cdots W_1 = W_1^\top \cdots W_{L-1}^\top W_L^\top = W^\top.$$

$\square$

By Lemma 7, $W_1(t), \ldots, W_L(t), W_{L:1}(t)$ and $\nabla \ell(W_{L:1}(t))$ are analytic, and hence infinitely differentiable, with respect to $t$. Lemmas 23 and 24 below thus establish that if $W_1(0), \ldots, W_L(0)$ form a symmetric factorization of $W_{L:1}(0)$, then the product matrix stays symmetric for all $t$.

**Lemma 23.** *Under the setting of Lemma 20, assume that the matrices $W_1(0), \ldots, W_L(0)$ form a symmetric factorization of $W_{L:1}(0)$ (Definition 4). Then, for all $k \in \mathbb{N} \cup \{0\}$:*

$$W_{L:1}^{(k)}(0) = W_{L:1}^{(k)}(0)^\top, \tag{43}$$

*and*

$$W_l^{(k)}(0) = W_{L-l+1}^{(k)}(0)^\top \quad , l \in \{1, \ldots, \lfloor L/2 \rfloor + 1\}. \tag{44}$$

*Proof.* The proof is by induction over $k$. For $k = 0$, the claim holds directly from the initialization assumption and Lemma 22. For $k \in \mathbb{N}$, suppose the claim is true for all $m \in \mathbb{N} \cup \{0\}$ with $m < k$. We begin by showing Equation (44) holds for $k$. In turn, this will lead to Equation (43) holding as well. For $l \in [L]$, the dynamics of $W_l(t)$ under gradient flow are

$$W_l^{(1)}(t) = -\frac{\partial}{\partial W_l} \phi(W_1(t), W_2(t), \ldots, W_L(t)) = -\prod_{r=l+1}^{L} W_r(t)^\top \cdot G(t) \cdot \prod_{r=1}^{l-1} W_r(t)^\top,$$

where $G(t) := \nabla \ell(W_{L:1}(t))$ denotes the loss gradient with respect to $W_{L:1}$ at time $t$. We can explicitly write the $k$'th ($k \geq 1$) derivative with respect to $t$ of each $W_l(t)$ using the product rule for higher order derivatives:

$$W_l^{(k)}(t) = -\sum_{i_1, \ldots, i_L} \binom{k-1}{i_1, \ldots, i_L} \prod_{r=l+1}^{L} W_r^{(i_r)}(t)^\top \cdot G^{(i_l)}(t) \cdot \prod_{r=1}^{l-1} W_r^{(i_r)}(t)^\top,$$

where $\sum_{l=1}^{L} i_l = k - 1$ and $\binom{k-1}{i_1, \ldots, i_L} = (k-1)! / (i_1! \cdots i_L!)$ for $i_1, \ldots, i_L \in \{0, 1, \ldots, k-1\}$. Taking the transpose of both sides we have:

$$W_l^{(k)}(t)^\top = -\sum_{i_1, \ldots, i_L} \binom{k-1}{i_1, \ldots, i_L} \prod_{1}^{r=l-1} W_r^{(i_r)}(t) \cdot G^{(i_l)}(t)^\top \cdot \prod_{l+1}^{r=L} W_r^{(i_r)}(t). \tag{45}$$

Turning our attention to $G(t)$, we may write it explicitly as:

$$G(t) = \nabla \ell(W_{L:1}(t)) = \begin{pmatrix} 0 & w_{1,2}(t) - 1 \\ w_{2,1}(t) - 1 & w_{2,2}(t) \end{pmatrix},$$

where $\{w_{i,j}(t)\}_{i,j \in [2]}$ are the entries of $W_{L:1}(t)$. For $m < k$, note that when $W_{L:1}^{(m)}(t)$ is symmetric so is $G^{(m)}(t)$. With this in hand, the inductive assumption (Equation (43)) implies that $G^{(m)}(0)$ is symmetric (for all $m < k$). Combined with Equation (44) (for $m < k$, from the inductive assumption), we may write Equation (45) for $t = 0$ as:

$$W_l^{(k)}(0)^\top = -\sum_{i_1, \ldots, i_L} \binom{k-1}{i_1, \ldots, i_L} \prod_{r=L-l+2}^{L} W_r^{(i_{L-r+1})}(0)^\top \cdot G^{(i_l)}(0) \cdot \prod_{r=1}^{L-l} W_r^{(i_{L-r+1})}(0)^\top.$$

Reordering the sum according to $h_r := i_{L-r+1}$ and noticing that $\binom{k-1}{h_1,\ldots,h_L} = \binom{k-1}{i_1,\ldots,i_L}$, we conclude:

$$W_l^{(k)}(0)^\top = -\sum_{h_1,\ldots,h_L} \binom{k-1}{h_1,\ldots,h_L} \prod_{r=L-l+2}^{L} W_r^{(h_r)}(0)^\top \cdot G^{(h_{L-l+1})}(0) \cdot \prod_{r=1}^{L-l} W_r^{(h_r)}(0)^\top .$$

That is,

$$W_l^{(k)}(0)^\top = W_{L-l+1}^{(k)}(0) ,$$

proving Equation (44).

It remains to show that $W_{L:1}^{(k)}(0)$ is symmetric. Similarly to before, we take the $k$'th derivative of $W_{L:1}(t) := W_L(t)\cdots W_1(t)$ using the product rule:

$$W_{L:1}^{(k)}(t) = \sum_{i_1,\ldots,i_L} \binom{k}{i_1,\ldots,i_L} \prod_{1}^{l=L} W_l^{(i_l)}(t) ,$$

where $\sum_{l=1}^{L} i_l = k$ and $\binom{k}{i_1,\ldots,i_L} = k!/(i_1!\cdots i_L!)$ for $i_1,\ldots,i_L \in \{0,1,\ldots,k\}$. For convenience, we denote $B_{i_1,\ldots,i_L}(t) := \binom{k}{i_1,\ldots,i_L} \prod_{1}^{l=L} W_l^{(i_l)}(t)$. Pairing up elements in the sum with indices $(i_1,\ldots,i_L)$ that are a reverse order of each other, i.e. $(i_1,\ldots,i_L)$ is paired with $(i_L,\ldots,i_1)$:

$$W_{L:1}^{(k)}(t) = \sum_{i_1,\ldots,i_L} \frac{1}{2}\left[ B_{i_1,\ldots,i_L}(t)) + B_{i_L,\ldots,i_1}(t) \right] . \tag{46}$$

With Equation (46) in place, we can conclude the proof by showing $W_{L:1}^{(k)}(0)$ is a sum of symmetric matrices. By the inductive assumption for Equation (44), which was established in the first part of the proof for $k$ as well, we have:

$$B_{i_1,\ldots,i_L}(0) = B_{i_L,\ldots,i_1}(0)^\top , \tag{47}$$

for each $(i_1,\ldots,i_L)$. Therefore, the matrix $B_{i_1,\ldots,i_L}(0) + B_{i_L,\ldots,i_1}(0)$ is symmetric. Plugging Equation (47) into Equation (46) with $t = 0$, we arrive at a representation of $W_{L:1}^{(k)}(0)$ as a sum of symmetric matrices. Thus, $W_{L:1}^{(k)}(0)$ is symmetric, completing the proof. $\qquad\square$

**Lemma 24.** *Under the setting of Lemma 20, assume that the matrices $W_1(0),\ldots,W_L(0)$ form a symmetric factorization of $W_{L:1}(0)$ (Definition 4). Then, $W_{L:1}(t)$ is symmetric for all $t \geq 0$.*

*Proof.* By Lemmas 23 and 10, we may conclude that for all $t \geq 0$:

$$W_l(t) = W_{L-l+1}(t)^\top \quad ,l \in \{1,\ldots,\lfloor L/2\rfloor + 1\} .$$

In words, $W_1(t),\ldots,W_L(t)$ form a symmetric factorization of $W_{L:1}(t)$, and therefore $W_{L:1}(t)$ is symmetric (Lemma 22). $\qquad\square$

Going back to the setting of Lemma 20 — initialization is balanced (Equation (5)), but does not necessarily comprise a symmetric factorization — we show that here too the product matrix remains symmetric throughout optimization. To do so, we first construct a factorization of $W_{L:1}(0)$ that is both balanced and symmetric, for which Lemma 24 ensures the product matrix stays symmetric throughout optimization. We then prove that the trajectories of the product matrix for the original and the modified initializations coincide.

Recall that $W_{L:1}(0) = \alpha \cdot I$ and define $\bar{W}_l(0) := \alpha^{\frac{1}{L}} \cdot I$ for $l \in [L]$. It is easily verified that:

- $W_{L:1}(0) = \bar{W}_L(0)\cdots\bar{W}_1(0)$.
- $\bar{W}_l(0) = \bar{W}_{L-l+1}(0)^\top$ for $l \in [L]$.
- $\bar{W}_{l+1}(0)^\top \bar{W}_{l+1}(0) = \bar{W}_l(0)\bar{W}_l(0)^\top$ for $l \in [L-1]$.

Meaning, $\bar{W}_1(0), \ldots, \bar{W}_L(0)$ are balanced, and form a symmetric factorization of $W_{L:1}(0)$. Suppose the factors $\bar{W}_1(t), \ldots, \bar{W}_L(t)$ follow the gradient flow dynamics, with initial values $\bar{W}_1(0), \ldots, \bar{W}_L(0)$, and let $\bar{W}_{L:1}(t) := \bar{W}_L(t) \cdots \bar{W}_1(t)$ be the induced product matrix. From Lemma 24, it follows that $\bar{W}_{L:1}(t)$ is symmetric for all $t \geq 0$.

As characterized in [6] (restated as Lemma 4), if the initial factors are balanced, the product matrix trajectory depends only on its initial value $W_{L:1}(0)$. Since both the original and modified initializations are balanced and have the same product matrix, they lead to the exact same trajectory. Thus, $W_{L:1}(t) = \bar{W}_{L:1}(t)$ for all $t \geq 0$, and specifically, $W_{L:1}(t)$ is symmetric.

The last step is to see that $W_{L:1}(t)$ is not only symmetric, but positive definite as well. Since its initial value $W_{L:1}(0)$ is positive definite, it suffices to show that its eigenvalues do not change sign. By Lemma 6, the determinant of $W_{L:1}(t)$ is positive for all $t$. Specifically, the product matrix does not have zero eigenvalues. Recalling that $W_{L:1}(t)$ is an analytic function of $t$ (Lemma 7), Theorem 6.1 in [50] implies that its eigenvalues are continuous in $t$. Therefore, they can not change sign, as that would require them to pass through zero, concluding the proof. □

### G.8 Proof of Theorem 2

The proof follows a similar line to that of Theorem 1 (Subappendix G.5), where the differences mostly stem from the fact that the solution set $\widetilde{S}$ (Equation (14)) is not confined to symmetric matrices, as opposed to the original $S$ (Equation (7)), slightly complicating the computation of singular values. For the sake of the proof, as mentioned in Subappendix G.1, we omit the time index $t$ when stating results for all $t \geq 0$ or when clear from context. We also let $\{w_{i,j}\}_{i,j \in [2]}$ denote the entries of the product matrix $W_{L:1}$.

We begin by deriving loss-dependent bounds for $|w_{1,1}|$, $\sigma_1(W_{L:1})$ and $\sigma_2(W_{L:1})$. The entries of $W_{L:1}$ can be trivially bounded by the loss as follows:

$$|w_{2,2} - \epsilon| \leq \sqrt{2\ell(W_{L:1})} \quad , \quad |w_{1,2} - z| \leq \sqrt{2\ell(W_{L:1})} \quad , \quad |w_{2,1} - z'| \leq \sqrt{2\ell(W_{L:1})}. \quad (48)$$

Lemma 25 below, analogous to Lemma 18 from the proof of Theorem 1, characterizes the relation between $|w_{1,1}|$ and the loss.

**Lemma 25.** *Suppose $\ell(W_{L:1}) < \min\{z^2/2, z'^2/2\}$. Then:*

$$|w_{1,1}| > \frac{(|z| - \sqrt{2\ell(W_{L:1})})(|z'| - \sqrt{2\ell(W_{L:1})})}{|\epsilon| + \sqrt{2\ell(W_{L:1})}} \geq \frac{|z| \cdot |z'|}{|\epsilon| + \sqrt{2\ell(W_{L:1})}} - (|z| + |z'|).$$

*Proof.* According to Lemma 6, the determinant of $W_{L:1}$ does not change sign, *i.e.* it remains equal to $\text{sign}(z \cdot z')$ (the initial sign assumed). Under the assumption that $\ell(W_{L:1}) < \min\{z^2/2, z'^2/2\}$, both $w_{1,2}$ and $w_{2,1}$ have the same signs as $z$ and $z'$, respectively, implying that $w_{2,2} \neq 0$ (otherwise we have a contradiction to the sign of the product matrix determinant). If $z \cdot z' > 0$, the determinant is positive as well, and it holds that $w_{1,1}w_{2,2} > w_{1,2}w_{2,1} > 0$. Otherwise, if $z \cdot z' < 0$ we have $w_{1,1}w_{2,2} < w_{1,2}w_{2,1} < 0$. Putting it together we may write $|w_{1,1}w_{2,2}| > |w_{1,2}w_{2,1}|$. Dividing by $|w_{2,2}|$ and applying the bounds from Equation (48) then completes the proof:

$$|w_{1,1}| > \frac{(|z| - \sqrt{2\ell(W_{L:1})})(|z'| - \sqrt{2\ell(W_{L:1})})}{|\epsilon| + \sqrt{2\ell(W_{L:1})}} \geq \frac{|z| \cdot |z'|}{|\epsilon| + \sqrt{2\ell(W_{L:1})}} - (|z| + |z'|).$$

□

We are now able to see that, indeed, the smaller $|\epsilon|$ is compared to $|z \cdot z'|$, the higher $|w_{1,1}|$ will be driven when the loss is minimized. With Lemma 25 in place, we are now able to bound the singular values of $W_{L:1}$.

**Lemma 26.** *The singular values of $W_{L:1}$ fulfill:*

$$\sigma_1(W_{L:1}) \geq \frac{1}{\sqrt{2}} \cdot |w_{1,1}| - \sqrt{2\ell(W_{L:1})} \,,$$

$$\sigma_2(W_{L:1}) \leq 4|\epsilon| + \left(4 + \frac{\sqrt{|z| \cdot |z'|}}{\min\{|z|, |z'|\}}\right) \sqrt{2\ell(W_{L:1})}. \quad (49)$$

*Furthermore, if $\ell(W_{L:1}) < \min\left\{z^2/2 \,,\; z'^2/2\right\}$, the bound on $\sigma_2(W_{L:1})$ may be simplified:*

$$\sigma_2(W_{L:1}) \leq 4|\epsilon| + 4\sqrt{2\ell(W_{L:1})}. \quad (50)$$

*Proof.* Define $W_{\widetilde{\mathcal{S}}} := \begin{pmatrix} w_{1,1} & z' \\ z & \epsilon \end{pmatrix}$, the orthogonal projection of $W_{L:1}$ onto the solution set $\widetilde{\mathcal{S}}$. From Corollary 8.6.2 in [33] we know that:

$$|\sigma_i(W_{L:1}) - \sigma_i(W_{\widetilde{\mathcal{S}}})| \leq \left\|W_{L:1} - W_{\widetilde{\mathcal{S}}}\right\|_F = \sqrt{2\ell(W_{L:1})} \quad , \ i = 1, 2. \tag{51}$$

This means that any bound on the singular values of $W_{\widetilde{\mathcal{S}}}$ can be transferred to those of $W_{L:1}$ (up to an additive loss-dependent term). It is straightforwardly verified that the squared singular values of $W_{\widetilde{\mathcal{S}}}$ are

$$\sigma_1^2(W_{\widetilde{\mathcal{S}}}) = \frac{1}{2}\left( w_{1,1}^2 + z^2 + z'^2 + \epsilon^2 + \sqrt{\left(w_{1,1}^2 + z^2 + z'^2 + \epsilon^2\right)^2 - 4\left(w_{1,1}\epsilon - zz'\right)^2} \right),$$
$$\sigma_2^2(W_{\widetilde{\mathcal{S}}}) = \frac{1}{2}\left( w_{1,1}^2 + z^2 + z'^2 + \epsilon^2 - \sqrt{\left(w_{1,1}^2 + z^2 + z'^2 + \epsilon^2\right)^2 - 4\left(w_{1,1}\epsilon - zz'\right)^2} \right). \tag{52}$$

Note that the term inside the square roots is non-negative for all $w_{1,1}, z, z', \epsilon$. Since all elements in the expression for $\sigma_1^2(W_{\widetilde{\mathcal{S}}})$ are non-negative, we have $\sigma_1(W_{\widetilde{\mathcal{S}}}) \geq (1/\sqrt{2}) \cdot |w_{1,1}|$. Combining this with Equation (51) completes the lower bound for $\sigma_1(W_{L:1})$.

Next, let $W_{\widetilde{\mathcal{S}}_0} := \begin{pmatrix} w_{1,1} & z' \\ z & 0 \end{pmatrix}$ be the matrix obtained by replacing the bottom-right entry of $W_{\widetilde{\mathcal{S}}}$ by 0. Replacing $\epsilon$ with 0 in Equation (52), and applying the identity $a - b = \frac{a^2 - b^2}{a+b}$ for $a, b \in \mathbb{R}$ such that $a + b \neq 0$, we get:

$$\sigma_2^2(W_{\widetilde{\mathcal{S}}_0}) = \frac{2z^2 z'^2}{w_{1,1}^2 + z^2 + z'^2 + \sqrt{\left(w_{1,1}^2 + z^2 + z'^2\right)^2 - 4z^2 z'^2}}$$
$$\leq \frac{2z^2 z'^2}{w_{1,1}^2 + z^2 + z'^2}. \tag{53}$$

We initially prove Equation (50) holds in the case where $\ell(W_{L:1}) < \min\left\{z^2/2 \,,\, z'^2/2\right\}$. By lifting said assumption we then show that the bound on $\sigma_2(W_{L:1})$ in Equation (49) holds for any loss value. Under the assumption that $\ell(W_{L:1}) < \min\left\{z^2/2 \,,\, z'^2/2\right\}$, taking the square root of both sides in Equation (53), we arrive at the following bound:

$$\sigma_2(W_{\widetilde{\mathcal{S}}_0}) \leq \sqrt{2} \cdot \frac{|z| \cdot |z'|}{\sqrt{w_{1,1}^2 + z^2 + z'^2}}$$
$$\leq \sqrt{6} \cdot \frac{|z| \cdot |z'|}{|w_{1,1}| + |z| + |z'|}$$
$$\leq \sqrt{6} \cdot \frac{|z| \cdot |z'|}{\frac{|z| \cdot |z'|}{|\epsilon| + \sqrt{2\ell(W_{L:1})}}}$$
$$\leq 3\left(|\epsilon| + \sqrt{2\ell(W_{L:1})}\right),$$

where in the second transition we applied the inequality $\sqrt{w_{1,1}^2 + z^2 + z'^2} \geq (|w_{1,1}| + |z| + |z'|)/\sqrt{3}$, and in the third made use of the bound on $|w_{1,1}|$ (Lemma 25). Applying Corollary 8.6.2 from [33] twice, once for the matrices $W_{L:1}$ and $W_{\widetilde{\mathcal{S}}}$, and another for $W_{\widetilde{\mathcal{S}}}$ and $W_{\widetilde{\mathcal{S}}_0}$, we have:

$$\sigma_2(W_{L:1}) \leq 3\left(|\epsilon| + \sqrt{2\ell(W_{L:1})}\right) + |\epsilon| + \sqrt{2\ell(W_{L:1})} = 4\left(|\epsilon| + \sqrt{2\ell(W_{L:1})}\right),$$

achieving the desired result from Equation (50). It remains to see that the bound on $\sigma_2(W_{L:1})$ in Equation (49) holds regardless of the loss value. When $\ell(W_{L:1}) < \min\left\{z^2/2 \,,\, z'^2/2\right\}$ it obviously holds since it is only looser than the bound already obtained under this assumption. Otherwise, going back to Equation (53), it can be seen that

$$\sigma_2^2(W_{\widetilde{\mathcal{S}}_0}) \leq \frac{2z^2 z'^2}{(z - z')^2 + 2|z| \cdot |z'|} \leq |z| \cdot |z'|.$$

Thus, $\sigma_2(W_{\widetilde{\mathcal{S}}_0}) \leq \sqrt{|z| \cdot |z'|}$. Following the same procedure as before (applying Corollary 8.6.2 from [33]), combined with the fact that $\ell(W_{L:1}) \geq \min\left\{z^2/2 \,,\, z'^2/2\right\}$ concludes the proof:

$$
\begin{aligned}
\sigma_2(W_{L:1}) &\leq \sqrt{|z| \cdot |z'|} + |\epsilon| + \sqrt{2\ell(W_{L:1})} \\
&\leq \frac{\sqrt{|z| \cdot |z'|}}{\min\{|z|\,,\,|z'|\}} \cdot \sqrt{2\ell(W_{L:1})} + |\epsilon| + \sqrt{2\ell(W_{L:1})} \\
&\leq 4\,|\epsilon| + \left(4 + \frac{\sqrt{|z| \cdot |z'|}}{\min\{|z|\,,\,|z'|\}}\right)\sqrt{2\ell(W_{L:1})}\,.
\end{aligned}
$$

$\square$

### G.8.1 Proof of Equation (15) (lower bound for quasi-norm)

Turning our attention to $\|W_{L:1}\|$, following the same steps as in the proof of Theorem 1 (Subappendix G.5.1) will lead to a generalized bound. By the triangle inequality:

$$
\|W_{L:1}\| \geq \frac{1}{c_{\|\cdot\|}}\left\|w_{1,1}\mathbf{e}_1\mathbf{e}_1^\top\right\| - \left\|W_{L:1} - w_{1,1}\mathbf{e}_1\mathbf{e}_1^\top\right\|\,, \tag{54}
$$

where $c_{\|\cdot\|} \geq 1$ is a constant with which $\|\cdot\|$ satisfies the weakened triangle inequality (see Footnote 2). Let us initially assume that $\ell(W_{L:1}) < \min\{z^2/2, z'^2/2\}$. We later lift this assumption, delivering a bound that holds for all loss values. Invoking Equation (48) we may bound the negative term in Equation (54) as follows:

$$
\begin{aligned}
\left\|W_{L:1} - w_{1,1}\mathbf{e}_1\mathbf{e}_1^\top\right\| &\leq c_{\|\cdot\|}|w_{2,2}|\left\|\mathbf{e}_2\mathbf{e}_2^\top\right\| + c_{\|\cdot\|}^2\left(|w_{2,1}|\left\|\mathbf{e}_2\mathbf{e}_1^\top\right\| + |w_{1,2}|\left\|\mathbf{e}_1\mathbf{e}_2^\top\right\|\right) \\
&\leq 3c_{\|\cdot\|}^2\left(\max\{|z|\,,|z'|,|\epsilon|\} + \sqrt{2\ell(W_{L:1})}\right)\max_{\substack{i,j\in\{1,2\}\\(i,j)\neq(1,1)}}\left\|\mathbf{e}_i\mathbf{e}_j^\top\right\| \\
&\leq 6c_{\|\cdot\|}^2\max\{|z|\,,|z'|,|\epsilon|\}\cdot\max_{\substack{i,j\in\{1,2\}\\(i,j)\neq(1,1)}}\left\|\mathbf{e}_i\mathbf{e}_j^\top\right\|\,,
\end{aligned}
$$

Returning to Equation (54), applying the inequality above and the bound on $|w_{1,1}|$ (Lemma 25) we have:

$$
\begin{aligned}
\|W_{L:1}\| &\geq \frac{\left\|\mathbf{e}_1\mathbf{e}_1^\top\right\|}{c_{\|\cdot\|}}\left(\frac{|z|\cdot|z'|}{|\epsilon| + \sqrt{2\ell(W_{L:1})}} - |z| - |z'|\right) - 6c_{\|\cdot\|}^2\max\{|z|\,,|z'|,|\epsilon|\}\max_{\substack{i,j\in\{1,2\}\\(i,j)\neq(1,1)}}\left\|\mathbf{e}_i\mathbf{e}_j^\top\right\| \\
&\geq \frac{\left\|\mathbf{e}_1\mathbf{e}_1^\top\right\|}{c_{\|\cdot\|}}\cdot\frac{|z|\cdot|z'|}{|\epsilon| + \sqrt{2\ell(W_{L:1})}} - 8c_{\|\cdot\|}^2\max\{|z|\,,|z'|,|\epsilon|\}\cdot\max_{i,j\in\{1,2\}}\left\|\mathbf{e}_i\mathbf{e}_j^\top\right\|\,.
\end{aligned}
$$

Since $\|W_{L:1}\|$ is trivially lower bounded by zero, defining the constants

$$
a_{\|\cdot\|} := \frac{\left\|\mathbf{e}_1\mathbf{e}_1^\top\right\|}{c_{\|\cdot\|}}\,,\quad b_{\|\cdot\|} := \max\left\{\frac{a_{\|\cdot\|}\cdot|z|\cdot|z'|}{|\epsilon| + \min\{|z|\,,|z'|\}}, 8c_{\|\cdot\|}^2\max\{|z|\,,|z'|,|\epsilon|\}\max_{i,j\in\{1,2\}}\left\|\mathbf{e}_i\mathbf{e}_j^\top\right\|\right\}\,,
$$

allows us, on the one hand, to arrive at a bound of the form:

$$
\|W_{L:1}\| \geq a_{\|\cdot\|}\cdot\frac{|z|\cdot|z'|}{|\epsilon| + \sqrt{2\ell(W_{L:1})}} - b_{\|\cdot\|}\,,
$$

and on the other hand, to remove the previous assumption on the loss: in the case where $\ell(W_{L:1}) \geq \min\{z^2/2, z'^2/2\}$, the bound is non-positive and trivially holds. Noticing this is exactly Equation (15) (recall we omitted the time index $t$), concludes this part of the proof.

### G.8.2 Proof of Equation (16) (upper bound for effective rank)

Derivation of the upper bound for effective rank (Definition 1) is initially done under the assumption that $\ell(W_{L:1}) < \min\{z^2/8, z'^2/8\}$. We then remove this assumption, establishing a bound that holds for all loss values.

The bounds on $\sigma_1(W_{L:1})$ and $\sigma_2(W_{L:1})$ in Lemma 26 give:

$$\rho_1(W_{L:1}) = \frac{\sigma_1(W_{L:1})}{\sigma_1(W_{L:1}) + \sigma_2(W_{L:1})}$$

$$\geq \frac{\sigma_1(W_{L:1})}{\sigma_1(W_{L:1}) + 4\left(|\epsilon| + \sqrt{2\ell(W_{L:1})}\right)}$$

$$= 1 - \frac{4\left(|\epsilon| + \sqrt{2\ell(W_{L:1})}\right)}{\sigma_1(W_{L:1}) + 4\left(|\epsilon| + \sqrt{2\ell(W_{L:1})}\right)}$$

$$\geq 1 - \frac{4\left(|\epsilon| + \sqrt{2\ell(W_{L:1})}\right)}{\frac{1}{\sqrt{2}} \cdot |w_{1,1}| + 4\,|\epsilon| + 3\sqrt{2\ell(W_{L:1})}}$$

$$\geq 1 - \frac{4\sqrt{2}\left(|\epsilon| + \sqrt{2\ell(W_{L:1})}\right)}{|w_{1,1}|} \,.$$

Additionally, under our assumption that $\ell(W_{L:1}) < \min\{z^2/8, z'^2/8\}$, the bound on $|w_{1,1}|$ in Lemma 25 can be simplified to:

$$|w_{1,1}| \geq \frac{(|z| - \sqrt{2\ell(W_{L:1})})(|z'| - \sqrt{2\ell(W_{L:1})})}{|\epsilon| + \sqrt{2\ell(W_{L:1})}} \geq \frac{\min\{|z|, |z'|\}^2}{4\left(|\epsilon| + \sqrt{2\ell(W_{L:1})}\right)} \,.$$

Combining the last two inequalities we have:

$$\rho_2(W_{L:1}) = 1 - \rho_1(W_{L:1}) \leq \frac{16\sqrt{2}\left(|\epsilon| + \sqrt{2\ell(W_{L:1})}\right)^2}{\min\{|z|, |z'|\}^2} \,.$$

It is now possible to see that, in accordance with Subsection 3.4, the smaller $|\epsilon|$ is compared to $\min\{|z|, |z'|\}$, the closer to zero $\rho_2(W_{L:1})$ becomes as the loss is minimized. Let $h\left(\rho_2(W_{L:1})\right) := -\rho_2(W_{L:1}) \cdot \ln\left(\rho_2(W_{L:1})\right) - (1 - \rho_2(W_{L:1})) \cdot \ln\left(1 - \rho_2(W_{L:1})\right)$ denote the binary entropy function, and recall that the effective rank of the product matrix defined to be $\mathrm{erank}(W_{L:1}) := \exp\{h\left(\rho_2(W_{L:1})\right)\}$. As in the proof of Theorem 1 (Subappendix G.5.2), we may bound the exponent on the interval $[0, \ln(2)]$ by the linear function intersecting it at these points. That is,

$$\mathrm{erank}(W_{L:1}) \leq 1 + \frac{1}{\ln(2)} \cdot h\left(\rho_2(W_{L:1})\right) \,.$$

From Lemma 9 it holds that $h\left(\rho_2(W_{L:1})\right) \leq 2\sqrt{\rho_2(W_{L:1})}$. Plugging this into the inequality above leads to:

$$\mathrm{erank}(W_{L:1}) \leq 1 + \frac{8 \cdot 2^{\frac{1}{4}}}{\ln(2) \cdot \min\{|z|, |z'|\}} \cdot \left(|\epsilon| + \sqrt{2\ell(W_{L:1})}\right)$$

$$\leq 1 + \frac{16}{\min\{|z|, |z'|\}} \cdot \left(|\epsilon| + \sqrt{2\ell(W_{L:1})}\right) \,,$$

where the second transition is a slight simplification of the constants $(2^{1/4}/\ln(2) < 2)$. As will be shown below, $\inf_{W' \in \widetilde{S}} \mathrm{erank}(W') = 1$. We may thus conclude:

$$\mathrm{erank}(W_{L:1}) \leq \inf_{W' \in \widetilde{S}} \mathrm{erank}(W') + \frac{16}{\min\{|z|, |z'|\}} \cdot \left(|\epsilon| + \sqrt{2\ell(W_{L:1})}\right) \,.$$

Notice that when $\ell(W_{L:1}) \geq \min\{z^2/8, z'^2/8\}$ the inequality trivially holds since the right-hand side is greater than 2 (the maximal effective rank for a $2 \times 2$ matrix). This establishes Equation (16) (time index is omitted).

It remains to prove that $\inf_{W' \in \widetilde{S}} \mathrm{erank}(W') = 1$. If $\epsilon \neq 0$, it is trivial since there exists $W' \in \widetilde{S}$ with $\mathrm{rank}(W') = 1$, meaning $\sigma_2(W') = 0$ and $\mathrm{erank}(W') = 1$. If $\epsilon = 0$, examining the squared singular values of $W' \in \widetilde{S}$ (Equation (52) with $(W')_{1,1}$ in place of $w_{1,1}$) reveals that $\lim_{(W')_{1,1} \to \infty} \sigma_2(W') = 0$, while $\lim_{(W')_{1,1} \to \infty} \sigma_1(W') = \infty$. Thus, there exists a matrix in $\widetilde{S}$ with effective rank arbitrarily close to 1. Since the effective rank of any matrix is at least 1, this implies that $\inf_{W' \in \widetilde{S}} \mathrm{erank}(W') = 1$.

### G.8.3 Proof of Equation (17) (upper bound for distance from infimal rank)

We claim that the infimal rank (Definition 2) of $\widetilde{\mathcal{S}}$ is 1. Since $z, z' \neq 0$, it cannot be 0. If $\epsilon \neq 0$, our claim is trivial since there exists $W' \in \widetilde{\mathcal{S}}$ with $\mathrm{rank}(W') = 1$. Otherwise, inspecting the squared singular values of a matrix $W' \in \widetilde{\mathcal{S}}$ (Equation (52) with $(W')_{1,1}$ in place of $w_{1,1}$), we can see that, when $\epsilon = 0$, taking $(W')_{1,1}$ to infinity drives the minimal singular value towards zero $(\lim_{(W')_{1,1} \to \infty} \sigma_2(W') = 0)$. Hence, the distance of $\widetilde{\mathcal{S}}$ from the set of matrices with rank 1 or less is 0 in this case as well.

The distance of the product matrix from the infimal rank of $\widetilde{\mathcal{S}}$ is therefore $D(W_{L:1}(t), \mathcal{M}_1) = \sigma_2(W_{L:1}(t))$. From Lemma 26 we have

$$D(W_{L:1}(t), \mathcal{M}_1) \leq 4\,|\epsilon| + \left( 4 + \frac{\sqrt{|z| \cdot |z'|}}{\min\{|z|, |z'|\}} \right) \sqrt{2\ell(t)},$$

for all $t \geq 0$.

### G.8.4 Robustness to change in observed locations

Lastly, we prove that the established bounds (Equations (15), (16) and (17)) are robust to a change in observed locations. Let $(i, j) \in [2] \times [2]$ be the unobserved entry's location. Following proof steps analogous to those in Lemmas 25 and 26 — while recalling our assumption of $\det(W_{L:1}(0))$ having same sign as $z \cdot z'$ if $i = j$ and opposite sign otherwise — yields identical bounds on the unobserved entry and singular values of $W_{L:1}$. Since the derivations of Equations (15), (16) and (17) in Subappendices G.8.1, G.8.2 and G.8.3, respectively, rely solely on the aforementioned bounds, the proof concludes. $\qquad\square$

### G.9 Proof of Lemma 1

For $l \in [L]$, let $W_l = U_l \Sigma_l V_l^\top$ be a singular value decomposition of $W_l$, *i.e.* $U_l, V_l \in \mathbb{R}^{d,d}$ are orthogonal matrices, and $\Sigma_l \in \mathbb{R}^{d,d}_{\geq 0}$ is diagonal holding the singular values of $W_l$ in non-increasing order. Define $\{W'_l \in \mathbb{R}^{d,d}\}_{l=1}^L$ by $W'_1 := W_1$, and:

$$W'_l := \prod_{2}^{r=l} \left[ U_r V_r^\top \right] \cdot U_1 \Sigma_1 U_1^\top \cdot \prod_{r=2}^{l-1} \left[ V_r U_r^\top \right] \quad, l = 2, 3, \dots, L\,.$$

First, for $l \in [L-1]$:

$$
\begin{aligned}
W'^\top_{l+1} W'_{l+1} &= \prod_{2}^{r=l} \left[ U_r V_r^\top \right] U_1 \Sigma_1 U_1^\top \prod_{r=2}^{l+1} \left[ V_r U_r^\top \right] \cdot \prod_{2}^{r=l+1} \left[ U_r V_r^\top \right] U_1 \Sigma_1 U_1^\top \prod_{r=2}^{l} \left[ V_r U_r^\top \right] \\
&= \prod_{2}^{r=l} \left[ U_r V_r^\top \right] U_1 \Sigma_1 U_1^\top \prod_{r=2}^{l-1} \left[ V_r U_r^\top \right] \cdot \prod_{2}^{r=l-1} \left[ U_r V_r^\top \right] U_1 \Sigma_1 U_1^\top \prod_{r=2}^{l} \left[ V_r U_r^\top \right] \\
&= W'_l W'^\top_l \,,
\end{aligned}
$$

*i.e.* $W'_1, W'_2, \dots, W'_L$ are balanced.

Second, by induction over $l \in [L]$, we prove that $\|W_l - W'_l\|_F \leq (l-1) \cdot \sqrt{\epsilon}$. For $l = 1$ this is trivial, as $W'_1 = W_1$ by definition. Assume that the bound holds for all $j < l$. Expressing $W_l$ and $W'_l$ in terms of $\{U_r, V_r, \Sigma_r\}_{r=1}^l$ yields:

$$\|W_l - W'_l\|_F = \left\| U_l \Sigma_l V_l^\top - \prod_{2}^{r=l} \left[ U_r V_r^\top \right] \cdot U_1 \Sigma_1 U_1^\top \cdot \prod_{r=2}^{l-1} \left[ V_r U_r^\top \right] \right\|_F.$$

The Frobenius norm is invariant to multiplication by orthogonal matrices. Thus, we may multiply by $V_l U_l^\top$ from the left:

$$
\begin{aligned}
\|W_l - W_l'\|_F &= \left\| V_l \Sigma_l V_l^\top - \prod_2^{r=l-1} \left[ U_r V_r^\top \right] \cdot U_1 \Sigma_1 U_1^\top \cdot \prod_{r=2}^{l-1} \left[ V_r U_r^\top \right] \right\|_F \\
&= \left\| \sqrt{W_l^\top W_l} - \sqrt{W_{l-1}' W_{l-1}'^\top} \right\|_F \\
&\leq \left\| \sqrt{W_l^\top W_l} - \sqrt{W_{l-1} W_{l-1}^\top} \right\|_F + \left\| \sqrt{W_{l-1} W_{l-1}^\top} - \sqrt{W_{l-1}' W_{l-1}'^\top} \right\|_F .
\end{aligned}
\tag{55}
$$

Since the unbalancedness magnitude of $W_1, W_2, \ldots, W_L$ is $\epsilon$, from the Powers-Størmer inequality (Lemma 4.1 in [69]) we know that:

$$
\left\| \sqrt{W_l^\top W_l} - \sqrt{W_{l-1} W_{l-1}^\top} \right\|_F \leq \sqrt{\left\| W_l^\top W_l - W_{l-1} W_{l-1}^\top \right\|_{nuclear}} \leq \sqrt{\epsilon} .
$$

Additionally, multiplying by $U_{l-1} V_{l-1}^\top$ from the right:

$$
\begin{aligned}
\left\| \sqrt{W_{l-1} W_{l-1}^\top} - \sqrt{W_{l-1}' W_{l-1}'^\top} \right\|_F &= \left\| U_{l-1} \Sigma_{l-1} U_{l-1}^\top - \prod_2^{r=l-1} \left[ U_r V_r^\top \right] U_1 \Sigma_1 U_1^\top \prod_{r=2}^{l-1} \left[ V_r U_r^\top \right] \right\|_F \\
&= \left\| U_{l-1} \Sigma_{l-1} V_{l-1}^\top - \prod_2^{r=l-1} \left[ U_r V_r^\top \right] U_1 \Sigma_1 U_1^\top \prod_{r=2}^{l-2} \left[ V_r U_r^\top \right] \right\|_F \\
&= \left\| W_{l-1} - W_{l-1}' \right\|_F \\
&\leq (l-2) \cdot \sqrt{\epsilon} ,
\end{aligned}
$$

where the last inequality is by the inductive assumption. Going back to Equation (55), we conclude:

$$
\|W_l - W_l'\|_F \leq \sqrt{\epsilon} + (l-2) \cdot \sqrt{\epsilon} = (l-1) \cdot \sqrt{\epsilon} .
$$

$\square$

### G.10   Proof of Lemma 2

Define $g : [0, T] \to \mathbb{R}_{\geq 0}$ by $g(t) := \|\theta(t) - \theta'(t)\|_2^2$. For any $t \in [0, T]$ it holds that:

$$
\begin{aligned}
\dot{g}(t) &= 2 \left\langle \theta(t) - \theta'(t), \dot{\theta}(t) - \dot{\theta}'(t) \right\rangle \\
&= -2 \left\langle \theta(t) - \theta'(t), \nabla f(\theta(t)) - \nabla f(\theta'(t)) \right\rangle .
\end{aligned}
$$

By the Cauchy-Schwartz inequality and smoothness of $f(\cdot)$, we have:

$$
\begin{aligned}
\dot{g}(t) &\leq 2 \|\theta(t) - \theta'(t)\|_2 \cdot \|\nabla f(\theta(t)) - \nabla f(\theta'(t))\|_2 \\
&\leq 2\beta \|\theta(t) - \theta'(t)\|_2^2 \\
&= 2\beta \cdot g(t) .
\end{aligned}
$$

Let $\bar{t} \in [0, T]$, and suppose that there exists $t_0 \in [0, \bar{t}]$ for which $g(t_0) = 0$. Consider the initial value problem induced by gradient flow over $f(\cdot)$ starting from the point $\theta(t_0) = \theta'(t_0)$. Since its solution on the interval $[t_0, \bar{t}]$ is unique (by the definition of $\theta(\cdot), \theta'(\cdot)$ there exist a solution lying within $\mathcal{D}$, and it not being unique would contradict, $e.g.$, Theorem 2.2 in [81]), it holds that $\theta(t) = \theta'(t)$ for any $t \in [t_0, \bar{t}]$. That is, Equation (19) trivially holds. Now assume that $\forall t \in [0, \bar{t}] : g(t) > 0$. Then:

$$
\frac{\dot{g}(t)}{g(t)} \leq 2\beta \implies \int_{t=0}^{\bar{t}} \frac{\dot{g}(t)}{g(t)} dt \leq 2\beta \bar{t} \implies \ln(g(\bar{t})) - \ln(g(0)) \leq 2\beta \bar{t} \implies g(\bar{t}) \leq g(0) \cdot \exp(2\beta \bar{t}) .
$$

Taking the square root of both sides in the latter inequality concludes the proof.   $\square$

## G.11 Proof of Proposition 5

We note that the following proof does not depend on the dimensions of the deep matrix factorization $(d_0, d_1, \ldots, d_L$; see Section 2). That is, Proposition 5 holds for arbitrary dimensions, and is not limited to the square case where $d_0 = d_1 = \ldots = d_L$.

Let $R' > 0$, and define $\mathcal{D}_{R'} := \{(W_1, W_2, \ldots, W_L) : \|(W_1, W_2, \ldots, W_L)\|_F < R'\}$. For any $(W_1, W_2, \ldots, W_L)$ and $(W_1', W_2', \ldots, W_L')$ in $\mathcal{D}_{R'}$:

$$\|\nabla\phi(W_1, W_2, \ldots, W_L) - \nabla\phi(W_1', W_2', \ldots, W_L')\|_F$$

$$= \sqrt{\sum_{l=1}^{L} \left\| \prod_{r=l+1}^{L} W_r^\top \cdot \nabla\ell(W_{L:1}) \cdot \prod_{r=1}^{l-1} W_r^\top - \prod_{r=l+1}^{L} W_r'^\top \cdot \nabla\ell(W_{L:1}') \cdot \prod_{r=1}^{l-1} W_r'^\top \right\|_F^2}. \quad (56)$$

Since $(\nabla\ell(W_{L:1}))_{i,j} = (W_{L:1})_{i,j} - b_{i,j}$ if $(i,j) \in \Omega$, and otherwise $(\nabla\ell(W_{L:1}))_{i,j} = 0$, we have that $\|\nabla\ell(W_{L:1}) - \nabla\ell(W_{L:1}')\|_F \le \|W_{L:1} - W_{L:1}'\|_F$ and $\|\nabla\ell(W_{L:1})\|_F \le \|W_{L:1}\|_F + B$. For each $l \in [L]$, Lemma 12 and sub-multiplicativity of the Frobenius norm then yield:

$$\left\| \prod_{r=l+1}^{L} W_r^\top \cdot \nabla\ell(W_{L:1}) \cdot \prod_{r=1}^{l-1} W_r^\top - \prod_{r=l+1}^{L} W_r'^\top \cdot \nabla\ell(W_{L:1}') \cdot \prod_{r=1}^{l-1} W_r'^\top \right\|_F$$

$$\le R'^{L-2}(R'^L + B) \cdot \sum_{r=1}^{L} \|W_r - W_r'\|_F + R'^{L-1} \cdot \|\nabla\ell(W_{L:1}) - \nabla\ell(W_{L:1}')\|_F$$

$$\le R'^{L-2}(R'^L + B) \cdot \sum_{r=1}^{L} \|W_r - W_r'\|_F + R'^{L-1} \|W_{L:1} - W_{L:1}'\|_F$$

$$\le R'^{L-2}(R'^L + B) \cdot \sum_{r=1}^{L} \|W_r - W_r'\|_F + R'^{2L-2} \cdot \sum_{r=1}^{L} \|W_r - W_r'\|_F$$

$$= \left(2R'^{2L-2} + BR'^{L-2}\right) \cdot \sum_{r=1}^{L} \|W_r - W_r'\|_F .$$

Plugging the inequality above into Equation (56), we conclude:

$$\|\nabla\phi(W_1, W_2, \ldots, W_L) - \nabla\phi(W_1', W_2', \ldots, W_L')\|_F$$

$$\le \sqrt{L \cdot (2R'^{2L-2} + BR'^{L-2})^2 \cdot \left(\sum_{l=1}^{L} \|W_l - W_l'\|_F\right)^2}$$

$$\le \sqrt{L^2 \cdot (2R'^{2L-2} + BR'^{L-2})^2 \cdot \sum_{l=1}^{L} \|W_l - W_l'\|_F^2}$$

$$= LR'^{L-2}\left(2R'^L + B\right) \cdot \|(W_1, W_2, \ldots, W_L) - (W_1', W_2', \ldots, W_L')\|_F ,$$

where the second inequality is by $\sum_{l=1}^{L} \|W_l - W_l'\|_F \le (L \cdot \sum_{l=1}^{L} \|W_l - W_l'\|_F^2)^{0.5}$. That is, the objective $\phi(\cdot)$ is $LR'^{L-2}\left(2R'^L + B\right)$-smooth over $\mathcal{D}_{R'}$. For any $\bar{t} \in [0, T]$, from Lemma 2 we have that:

$$\|\theta(\bar{t}) - \theta'(\bar{t})\|_F \le \|\theta(0) - \theta'(0)\|_F \cdot \exp\left(LR'^{L-2}\left(2R'^L + B\right) \cdot \bar{t}\right) .$$

Notice that $R$ is finite (it is the supremum of a continuous function over a compact domain). Since the inequality above holds for all $R' > R$, taking the limit $R' \to R^+$ yields Equation (20). $\square$

## G.12 Proof of Lemma 3

For any $l \in [L]$:

$$\dot{W}_l(t) = -\prod_{r=l+1}^{L} W_r(t)^\top \cdot \nabla\ell(W_{L:1}(t)) \cdot \prod_{r=1}^{l-1} W_r(t)^\top \quad , \ \forall t \ge 0 .$$

This implies that for any $l \in [L-1]$:

$$\dot{W}_l(t)W_l(t)^\top = W_{l+1}(t)^\top \dot{W}_{l+1}(t) \quad , \forall t \geq 0 \, .$$

Adding the transpose of the latter equality to itself, we have:

$$\tfrac{d}{dt}(W_l(t)W_l(t)^\top) = \tfrac{d}{dt}(W_{l+1}(t)^\top W_{l+1}(t)) \quad , \forall t \geq 0 \, .$$

Hence, $W_{l+1}(t)^\top W_{l+1}(t) - W_l(t)W_l(t)^\top = W_{l+1}(0)^\top W_{l+1}(0) - W_l(0)W_l(0)^\top$ for all $t \geq 0$. Taking the nuclear norm of both sides and maximizing over $l \in [L-1]$ yields the desired result. $\square$

### G.13 Proof of Theorem 3

The proof of Theorem 1 (Subappendix G.5) relies solely on the fact that $\det(W_{L:1}(t))$ remains positive through time. In particular, it establishes that the bounds on (quasi-)norms, effective rank and distance from infimal rank (Equations (8), (9) and (10) respectively) are guaranteed to hold for any $t \geq 0$ for which $\det(W_{L:1}(t)) > 0$. Therefore, if $\det(W_{L:1}(t)) > 0$ for all $t \geq 0$, the proof concludes. Otherwise, let $T \in [0, \infty)$ be the initial time for which $\det(W_{L:1}(T)) = 0$. Formally, define:

$$T := \inf \{t | t \in [0, \infty) \text{ and } \det(W_{L:}(t)) = 0\} \, .$$

Since $\det(W_{L:1}(t))$ is continuous in $t$, the set on the right hand side is non-empty, closed, and bounded from below. Thus, $T$ is well defined (*i.e.* $-\infty < T < \infty$), $\det(W_{L:1}(T)) = 0$, and $\det(W_{L:1}(t)) > 0$ for any $t \in [0, T)$ (recall that by assumption the determinant is positive at initialization). That is, the results of Theorem 1 hold over $[0, T)$. Let:

$$R := \begin{cases} \left[ \frac{(1-\sqrt{\ell_{init}})^4}{2^{16}} \cdot \ln\left(\frac{1}{\epsilon}\right) \right]^{1/6} - 1 & , \text{if depth } L = 2 \\[2ex] \left[ \frac{(1-\sqrt{\ell_{init}})^4}{2^{2L+8}L^4} \cdot \frac{1}{\epsilon^{1/32}} \right]^{1/(4L-2)} - 1 & , \text{if depth } L \geq 3 \end{cases} \, . \tag{57}$$

The proof proceeds in two parts. First, assuming that $\max_{l \in [L]} \|W_l(t)\|_F \leq R$ for any $t \in [0, T]$, Subappendix G.13.1 establishes Equation (21) by deriving a lower bound on $T$. Otherwise, if there exists $t \in [0, T]$ with $\max_{l \in [L]} \|W_l(t)\|_F > R$, Subappendix G.13.2 shows that Equations (22), (23) and (24) jointly hold for some $\bar{t} \in [0, t]$, completing the proof.

#### G.13.1 Proof of Equation (21) (if weight matrices are bounded by $R$)

Assume that $\max_{l \in [L]} \|W_l(t)\|_F \leq R$ for any $t \in [0, T]$. The loss during gradient flow is monotonically non-increasing (Lemma 17). This implies that $\ell(W_{L:1}(t)) \leq \ell_{init}$ for all $t \geq 0$. Hence, $\min\{((W_{L:1}(t))_{1,2} - 1)^2, ((W_{L:1}(t))_{2,1} - 1)^2\} \leq \ell_{init}$ and

$$\begin{aligned} \sigma_1(W_{L:1}(t)) &\geq \frac{1}{\sqrt{2}} \|W_{L:1}(t)\|_F \\ &\geq \frac{1}{\sqrt{2}} \max\{(W_{L:1}(t))_{1,2}, (W_{L:1}(t))_{2,1}\} \\ &\geq \frac{1 - \sqrt{\ell_{init}}}{\sqrt{2}} \, , \end{aligned} \tag{58}$$

for all $t \geq 0$. Define:

$$t_0 := \begin{cases} 0 & , \forall t \in [0, T] : \sigma_2(W_{L:1}(t)) < \frac{1-\sqrt{\ell_{init}}}{2} \\ \sup\left\{t | t \in [0, T] \text{ and } \sigma_2(W_{L:1}(t)) = \frac{1-\sqrt{\ell_{init}}}{2}\right\} & , \text{otherwise} \end{cases} \, .$$

By Lemma 5, $\sigma_2(W_{L:1}(t))$ is a continuous function of $t$. Combined with the fact that $\sigma_2(W_{L:1}(T)) = 0$ (recall the determinant is zero at $T$), we have that the set in the alternative case above is non-empty and compact. Thus, $t_0$ is well defined (*i.e.* $-\infty < t_0 < \infty$). If $t_0 = 0$, then $\sigma_2(W_{L:1}(t_0)) > 0$ since by assumption $\det(W_{L:1}(0)) > 0$. Otherwise, continuity of $\sigma_2(W_{L:1}(t))$ with respect to $t$ implies that $\sigma_2(W_{L:1}(t_0)) = (1 - \sqrt{\ell_{init}})/2 > 0$. Therefore, in either case $t_0 < T$, and $[t_0, T]$ is the interval preceding $T$ over which $\sigma_2(W_{L:1}(t)) \leq (1 - \sqrt{\ell_{init}})/2$.

Fix an arbitrary $\hat{t} \in [t_0, T]$. For conciseness, we hereafter omit the time index in functions of $t$ at $\hat{t}$, *e.g.* $W_{L:1}$ will be shorthand for $W_{L:1}(\hat{t})$. We now seek to derive a lower bound on $\frac{d}{dt}\sigma_2^2(W_{L:1})$.

Integrating said bound will yield the desired result. By Lemmas 3 and 1, there exist balanced matrices $W'_1, W'_2, \ldots, W'_L$ satisfying:

$$\|W_l - W'_l\|_F \le (l-1) \cdot \sqrt{\epsilon} \quad, \forall l \in [L]. \tag{59}$$

Applying Lemma 12, and noticing that $\max_{l \in [L]} \|W'_l\|_F \le R + (L-1) \cdot \sqrt{\epsilon} \le R + 1$, we bound the distance between the induced product matrices:

$$\|W_{L:1} - W'_{L:1}\|_F \le (R+1)^{L-1} \cdot \sum_{l=1}^{L}(l-1) \cdot \sqrt{\epsilon} \le \frac{1}{2}L^2(R+1)^{L-1} \cdot \sqrt{\epsilon}. \tag{60}$$

Consider a gradient flow path originating from $W'_1, W'_2, \ldots, W'_L$, where for simplicity we regard the initial time of this path as $\hat{t}$. For a balanced flow, Lemma 5 characterizes the movement of the product matrix's singular values.[21] Omitting the time index, by the Cauchy-Schwartz inequality and the fact that the Frobenius norm of an outer product between unit vectors is 1, we have that:

$$\left| \frac{d}{dt}\sigma_2(W'_{L:1})^2 \right| = \left| 2\sigma_2(W'_{L:1}) \cdot \frac{d}{dt}\sigma_2(W'_{L:1}) \right|$$

$$\le 2L \cdot \sigma_2(W'_{L:1})^{3-2/L} \cdot \|\nabla\ell(W'_{L:1})\|_F$$

$$\le 4L(R+1)^L \cdot \sigma_2(W'_{L:1})^{3-2/L},$$

where the last transition is by $\|\nabla\ell(W'_{L:1})\|_F \le \|W'_{L:1}\|_F + \sqrt{2} \le 2(R+1)^L$. Applying the singular values perturbation bound $\sigma_2(W'_{L:1}) \le \sigma_2(W_{L:1}) + \|W_{L:1} - W'_{L:1}\|_F$ (Corollary 8.6.2 in [33]), the bound in Equation (60), and Jensen's inequality, we arrive at:

$$\left| \frac{d}{dt}\sigma_2(W'_{L:1})^2 \right| \le 4L(R+1)^L \cdot \left[ \sigma_2(W_{L:1}) + \frac{1}{2}L^2(R+1)^{L-1} \cdot \sqrt{\epsilon} \right]^{3-2/L}$$

$$\le 4 \cdot 2^{2-2/L} L(R+1)^L \cdot \left[ \sigma_2(W_{L:1})^{3-2/L} + \left( \frac{1}{2}L^2(R+1)^{L-1} \cdot \sqrt{\epsilon} \right)^{3-2/L} \right]$$

$$\le 2^4 L(R+1)^L \cdot \sigma_2(W_{L:1})^{3-2/L} + 2 \cdot L^{7-4/L}(R+1)^{4L-5+2/L} \cdot \epsilon^{3/2-1/L}. \tag{61}$$

We turn our attention to $\left| \frac{d}{dt}\sigma_2(W_{L:1})^2 - \frac{d}{dt}\sigma_2(W'_{L:1})^2 \right|$. Recalling that $W_{L:1}$ is a 2-by-2 matrix, its minimal squared singular value can be written as:

$$\sigma_2(W_{L:1})^2 = \frac{1}{2}\left( \|W_{L:1}\|_F^2 - \sqrt{\|W_{L:1}\|_F^4 - 4\det(W_{L:1})^2} \right).$$

Differentiating with respect to time, while noticing that $(\|W_{L:1}\|_F^4 - 4\det(W_{L:1})^2)^{0.5} = \sigma_1(W_{L:1})^2 - \sigma_2(W_{L:1})^2$, we obtain:

$$\frac{d}{dt}\sigma_2(W_{L:1})^2 = \left\langle W_{L:1}, \frac{d}{dt}W_{L:1} \right\rangle - \frac{\|W_{L:1}\|_F^2 \left\langle W_{L:1}, \frac{d}{dt}W_{L:1} \right\rangle + 2\det(W_{L:1}) \left\langle A_{W_{L:1}}, \frac{d}{dt}W_{L:1} \right\rangle}{\sigma_1(W_{L:1})^2 - \sigma_2(W_{L:1})^2},$$

with $A_{W_{L:1}} := \begin{pmatrix} (W_{L:1})_{2,2} & -(W_{L:1})_{2,1} \\ -(W_{L:1})_{1,2} & (W_{L:1})_{1,1} \end{pmatrix}$. The same derivation holds for $\frac{d}{dt}\sigma_2(W'_{L:1})$, with $W'_{L:1}$ in place of $W_{L:1}$. Applying the triangle inequality, $\left| \frac{d}{dt}\sigma_2(W_{L:1})^2 - \frac{d}{dt}\sigma_2(W'_{L:1})^2 \right|$ is upper bounded by the sum of the expressions in Equations (62), (63) and (64) below:

$$\left| \left\langle W_{L:1}, \frac{d}{dt}W_{L:1} \right\rangle - \left\langle W'_{L:1}, \frac{d}{dt}W'_{L:1} \right\rangle \right|, \tag{62}$$

$$\left| \frac{\|W_{L:1}\|_F^2 \left\langle W_{L:1}, \frac{d}{dt}W_{L:1} \right\rangle}{\sigma_1(W_{L:1})^2 - \sigma_2(W_{L:1})^2} - \frac{\|W'_{L:1}\|_F^2 \left\langle W'_{L:1}, \frac{d}{dt}W'_{L:1} \right\rangle}{\sigma_1(W'_{L:1})^2 - \sigma_2(W'_{L:1})^2} \right|, \tag{63}$$

$$\left| \frac{2\det(W_{L:1}) \left\langle A_{W_{L:1}}, \frac{d}{dt}W_{L:1} \right\rangle}{\sigma_1(W_{L:1})^2 - \sigma_2(W_{L:1})^2} - \frac{2\det(W'_{L:1}) \left\langle A_{W'_{L:1}}, \frac{d}{dt}W'_{L:1} \right\rangle}{\sigma_1(W'_{L:1})^2 - \sigma_2(W'_{L:1})^2} \right|. \tag{64}$$

Lemmas 28, 29 and 30 (with the aid of Lemma (27)) derive upper bounds for the expressions in Equations (62), (63) and (64), respectively. Then, putting it all together, we arrive at:

$$\left| \frac{d}{dt}\sigma_2(W_{L:1})^2 - \frac{d}{dt}\sigma_2(W'_{L:1})^2 \right| \le 6L^3(R+1)^{4L-3} \cdot \sqrt{\epsilon} + \frac{384L^3(R+1)^{8L-3}}{(1-\sqrt{\ell_{init}})^4} \cdot \sqrt{\epsilon}$$

$$+ \frac{768L^3(R+1)^{8L-3}}{(1-\sqrt{\ell_{init}})^4} \cdot \sqrt{\epsilon}$$

$$\le \frac{1158L^3(R+1)^{8L-3}}{(1-\sqrt{\ell_{init}})^4} \cdot \sqrt{\epsilon}.$$

Going back to Equation (61), this implies that:

$$\left| \frac{d}{dt}\sigma_2(W_{L:1})^2 \right| \le \left| \frac{d}{dt}\sigma_2(W'_{L:1})^2 \right| + \left| \frac{d}{dt}\sigma_2(W_{L:1})^2 - \frac{d}{dt}\sigma_2(W'_{L:1})^2 \right|$$

$$\le 2^4 L(R+1)^L \cdot \left(\sigma_2(W_{L:1})^2\right)^{3/2-1/L} + \frac{2^{11}L^7(R+1)^{8L-3}}{(1-\sqrt{\ell_{init}})^4} \cdot \sqrt{\epsilon}.$$

We are now in a position to achieve the sought-after lower bound on $T$. If $L = 2$, according to Lemma 16:

$$\sigma_2(W_{L:1}(T))^2 \ge \left( \frac{2^7 L^6(R+1)^{7L-3}}{(1-\sqrt{\ell_{init}})^4} \cdot \sqrt{\epsilon} + \sigma_{init}^2 \right) \cdot e^{-2^4 L(R+1)^L(T-t_0)} - \frac{2^7 L^6(R+1)^{7L-3}}{(1-\sqrt{\ell_{init}})^4} \cdot \sqrt{\epsilon},$$

where $\sigma_{init} := \min\{\sigma_2(W_{L:1}(0)), (1-\sqrt{\ell_{init}})/2\} = \sigma_2(W_{L:1}(t_0))$. Due to the fact that $T$ is the initial point in time for which $\det(W_{L:1}(T)) = 0$, the right hand side must be non-positive, in which case:

$$T \ge \frac{1}{2^4 L(R+1)^L} \left[ \frac{1}{2}\ln\left(\frac{1}{\epsilon}\right) - \ln\left(\frac{2^7 L^6(R+1)^{7L-3}}{(1-\sqrt{\ell_{init}})^4 \sigma_{init}^2}\right) \right].$$

Since $\ln\left(2^7 L^6(R+1)^{7L-3}/(1-\sqrt{\ell_{init}})^4\sigma_{init}^2\right) \cdot 2^{-4}L^{-1}(R+1)^{-L} \le \ln\left(e/(1-\sqrt{\ell_{init}})\sigma_{init}\right)$, plugging in the value of $R$ (Equation (57)) leads to the desired result (case $L = 2$ in Equation (21)).

Similarly, for depth $L \ge 3$, Lemma 16 gives the following lower bound:

$$\sigma_2(W_{L:1}(T))^2$$

$$\ge \frac{1}{b_{L,R}^{\frac{2L}{3L-2}} \left[ b_{L,R}^{\frac{2L}{3L-2}}(1/2-1/L)(T-t_0) + \left( a_{L,R}^{\frac{2L}{3L-2}} + b_{L,R}^{\frac{2L}{3L-2}} \cdot \sigma_{init}^2 \right)^{\frac{2-L}{2L}} \right]^{\frac{2L}{L-2}}} - \left( \frac{a_{L,R}}{b_{L,R}} \right)^{\frac{2L}{3L-2}}$$

$$\ge \frac{1}{b_{L,R}^{\frac{2L}{3L-2}} \left[ b_{L,R}^{\frac{2L}{3L-2}} \cdot T + b_{L,R}^{\frac{2-L}{3L-2}} \cdot \sigma_{init}^{\frac{2-L}{L}} \right]^{\frac{2L}{L-2}}} - \left( \frac{a_{L,R}}{b_{L,R}} \right)^{\frac{2L}{3L-2}},$$

where $a_{L,R} := \frac{2^{11}L^7(R+1)^{8L-3}}{(1-\sqrt{\ell_{init}})^4} \cdot \sqrt{\epsilon}$ and $b_{L,R} := 2^4 L(R+1)^L$. Since $\det(W_{L:1}(T)) = 0$, the lower bound must be non-positive, in which case:

$$T \ge b_{L,R}^{-\frac{2L}{3L-2}} \cdot a_{L,R}^{-\frac{L-2}{3L-2}} - b_{L,R}^{-1} \cdot \sigma_{init}^{-\frac{L-2}{L}}.$$

Noticing that $b_{L,R}^{-1} \le 2^{-(5L+5)}$, and replacing $a_{L,R}, b_{L,R}$ with their explicit expressions, we arrive at:

$$T \ge 2^{\frac{-19L+22}{3L-2}}(1-\sqrt{\ell_{init}})^{\frac{4L-8}{3L-2}} L^{\frac{-9L+14}{3L-2}}(R+1)^{-\frac{-10L^2+19L-6}{3L-2}} \cdot \epsilon^{-\frac{L-2}{6L-4}} - 2^{-(5L+5)}\sigma_{init}^{-\frac{L-2}{L}}.$$

To simplify the result, we may lower bound the exponent of $R$ by $-4L$. Then, plugging in the value of $R$ (Equation (57)), and applying straightforward bounds to the exponents of $2$, $(1 - \sqrt{\ell_{init}})$, $L$ and $\epsilon$, concludes this part of the proof (*i.e.* establishes the case $L \geq 3$ in Equation (21)):

$$
T \geq 2^{\frac{4L^2+16L}{2L-1} - \frac{19L-22}{3L-2}} (1 - \sqrt{\ell_{init}})^{\frac{4L-8}{3L-2} - \frac{16L}{4L-2}} L^{\frac{16L}{4L-2} - \frac{9L-14}{3L-2}} \cdot \epsilon^{\frac{L}{8(4L-2)} - \frac{L-2}{6L-4}} - 2^{-(5L+5)} \sigma_{init}^{-\frac{L-2}{L}}
$$

$$
\geq 2^{4L/3} (1 - \sqrt{\ell_{init}})^{-2} L \cdot \epsilon^{-\frac{3L-8}{32L-16}} - 2^{-(5L+5)} \sigma_{init}^{-\frac{L-2}{L}}.
$$

$\square$

#### G.13.1.1 Auxiliary Lemmas

**Lemma 27.** *In the context of the proof for Equation* (21) *(Subappendix G.13.1), the following inequalities hold:*

$$
\max \left\{ \left\| \tfrac{d}{dt} W_{L:1} \right\|_F, \left\| \tfrac{d}{dt} W'_{L:1} \right\|_F \right\} \leq 2L(R+1)^{3L-2}, \tag{65}
$$

$$
\left\| \tfrac{d}{dt} W_{L:1} - \tfrac{d}{dt} W'_{L:1} \right\|_F \leq 5L^3 (R+1)^{3L-3} \cdot \sqrt{\epsilon}, \tag{66}
$$

$$
\left| \sigma_1(W_{L:1})^2 - \sigma_2(W_{L:1})^2 - (\sigma_1(W'_{L:1})^2 - \sigma_2(W'_{L:1})^2) \right| \leq 2L^2 (R+1)^{2L-1} \cdot \sqrt{\epsilon}, \tag{67}
$$

$$
\left( \sigma_1(W_{L:1})^2 - \sigma_2(W_{L:1})^2 \right) \left( \sigma_1(W'_{L:1})^2 - \sigma_2(W'_{L:1})^2 \right) \geq (1 - \sqrt{\ell_{init}})^4 / 32. \tag{68}
$$

*Proof.* Starting with Equation (65), the derivative of $W_{L:1}$ with respect to time is:

$$
\frac{d}{dt} W_{L:1} = -\sum_{l=1}^{L} \prod_{l+1}^{r=L} W_r \prod_{r=l+1}^{L} W_r^\top \cdot \nabla\ell(W_{L:1}) \cdot \prod_{r=1}^{l-1} W_r^\top \prod_{1}^{r=l-1} W_r.
$$

Therefore:

$$
\left\| \frac{d}{dt} W_{L:1} \right\|_F \leq \sum_{l=1}^{L} \left\| \prod_{l+1}^{r=L} W_r \prod_{r=l+1}^{L} W_r^\top \cdot \nabla\ell(W_{L:1}) \cdot \prod_{r=1}^{l-1} W_r^\top \prod_{1}^{r=l-1} W_r \right\|_F
$$

$$
\leq 2L(R+1)^{3L-2},
$$

where the last transition is by sub-multiplicativity of the Frobenius norm, the fact that $\|W_r\|_F \leq R+1$ for $r \in [L]$, and $\|\nabla\ell(W_{L:1})\|_F \leq 2(R+1)^L$. The exact same bound can be derived for $\left\| \tfrac{d}{dt} W'_{L:1} \right\|_F$, completing the proof of Equation (65).

Moving on to Equation (66):

$$
\left\| \frac{d}{dt} W_{L:1} - \frac{d}{dt} W'_{L:1} \right\|_F \leq \sum_{l=1}^{L} \Bigg\| \prod_{l+1}^{r=L} W_r \prod_{r=l+1}^{L} W_r^\top \cdot \nabla\ell(W_{L:1}) \cdot \prod_{r=1}^{l-1} W_r^\top \prod_{1}^{r=l-1} W_r
$$

$$
- \prod_{l+1}^{r=L} W'_r \prod_{r=l+1}^{L} W_r'^\top \cdot \nabla\ell(W'_{L:1}) \cdot \prod_{r=1}^{l-1} W_r'^\top \prod_{1}^{r=l-1} W'_r \Bigg\|_F.
$$

Noticing that $\|\nabla\ell(W_{L:1}) - \nabla\ell(W'_{L:1})\|_F \leq \|W_{L:1} - W'_{L:1}\|_F$, according to Lemma 12 we may bound each term in the sum by $4(R+1)^{3L-3} \cdot \sum_{l=1}^{L} \|W_l - W'_l\|_F + (R+1)^{2L-2} \cdot \|W_{L:1} - W'_{L:1}\|_F$. Applying the bounds from Equations (59) and (60) then establishes Equation (66):

$$
\left\| \frac{d}{dt} W_{L:1} - \frac{d}{dt} W'_{L:1} \right\|_F \leq L \cdot \left[ 4L^2 (R+1)^{3L-3} \cdot \sqrt{\epsilon} + \frac{1}{2} L^2 (R+1)^{3L-3} \cdot \sqrt{\epsilon} \right]
$$

$$
\leq 5L^3 (R+1)^{3L-3} \cdot \sqrt{\epsilon}.
$$

Next, Equation (67) is straightforwardly derived using a perturbation bound on the singular values (Corollary 8.6.2 in [33]):

$$
\left| \sigma_1(W_{L:1})^2 - \sigma_1(W'_{L:1})^2 \right| = |\sigma_1(W_{L:1}) - \sigma_1(W'_{L:1})| \cdot |\sigma_1(W_{L:1}) + \sigma_1(W'_{L:1})|
$$

$$
\leq 2(R+1)^L \|W_{L:1} - W'_{L:1}\|_F
$$

$$
\leq L^2 (R+1)^{2L-1} \cdot \sqrt{\epsilon},
$$

where the last transition is by Equation (60). The same derivation shows that a similar inequality holds for $\sigma_2(\cdot)$, establishing Equation (67):

$$\left| \sigma_1(W_{L:1})^2 - \sigma_2(W_{L:1})^2 - (\sigma_1(W'_{L:1})^2 - \sigma_2(W'_{L:1})^2) \right|$$
$$\leq \left| \sigma_1(W_{L:1})^2 - \sigma_1(W'_{L:1})^2 \right| + \left| \sigma_2(W_{L:1})^2 - \sigma_2(W'_{L:1})^2 \right|$$
$$\leq 2L^2(R+1)^{2L-1} \cdot \sqrt{\epsilon}.$$

Lastly, recall that $\sigma_2(W_{L:1}(t)) \leq (1 - \sqrt{\ell_{init}})/2$ over $[t_0, T]$. Combined with Equation (58), this implies that $\sigma_1(W_{L:1})^2 - \sigma_2(W_{L:1})^2 \geq (1 - \sqrt{\ell_{init}})^2/4$. Then, by Equation (67) we have:

$$\sigma_1(W'_{L:1})^2 - \sigma_2(W'_{L:1})^2$$
$$\geq \sigma_1(W_{L:1})^2 - \sigma_2(W_{L:1})^2 - \left| \sigma_1(W_{L:1})^2 - \sigma_2(W_{L:1})^2 - (\sigma_1(W'_{L:1})^2 - \sigma_2(W'_{L:1})^2) \right|$$
$$\geq (1 - \sqrt{\ell_{init}})^2/4 - 2L^2(R+1)^{2L-1} \cdot \sqrt{\epsilon}.$$

Noticing that $\epsilon \leq \frac{(1 - \sqrt{\ell_{init}})^4}{2^{2L+8}L^4(R+1)^{4L-2}}$, in which case $2L^2(R+1)^{2L-1} \cdot \sqrt{\epsilon} \leq (1 - \sqrt{\ell_{init}})^2/8$, concludes the proof:

$$(\sigma_1(W_{L:1})^2 - \sigma_2(W_{L:1})^2)(\sigma_1(W'_{L:1})^2 - \sigma_2(W'_{L:1})^2) \geq \frac{(1 - \sqrt{\ell_{init}})^2}{4} \cdot \frac{(1 - \sqrt{\ell_{init}})^2}{8}.$$

$\square$

**Lemma 28.** *In the context of the proof for Equation* (21) *(Subappendix G.13.1):*

$$\left| \left\langle W_{L:1}, \frac{d}{dt}W_{L:1} \right\rangle - \left\langle W'_{L:1}, \frac{d}{dt}W'_{L:1} \right\rangle \right| \leq 6L^3(R+1)^{4L-3} \cdot \sqrt{\epsilon},$$

*and*

$$\left| \left\langle A_{W_{L:1}}, \frac{d}{dt}W_{L:1} \right\rangle - \left\langle A_{W'_{L:1}}, \frac{d}{dt}W'_{L:1} \right\rangle \right| \leq 6L^3(R+1)^{4L-3} \cdot \sqrt{\epsilon}.$$

*Proof.* Starting with the former inequality, adding and subtracting $\left\langle W'_{L:1}, \frac{d}{dt}W_{L:1} \right\rangle$, followed by the triangle and Cauchy-Schwartz inequalities, we have that $\left| \left\langle W_{L:1}, \frac{d}{dt}W_{L:1} \right\rangle - \left\langle W'_{L:1}, \frac{d}{dt}W'_{L:1} \right\rangle \right|$ is bounded by:

$$\|W_{L:1} - W'_{L:1}\|_F \cdot \left\| \frac{d}{dt}W_{L:1} \right\|_F + \|W'_{L:1}\|_F \cdot \left\| \frac{d}{dt}W_{L:1} - \frac{d}{dt}W'_{L:1} \right\|_F.$$

From Equation (60) we know that $\|W_{L:1} - W'_{L:1}\|_F \leq (1/2)L^2(R+1)^{L-1} \cdot \sqrt{\epsilon}$. Additionally, by sub-multiplicativity of the Frobenius norm, $\|W'_{L:1}\|_F \leq (R+1)^L$. Applying Equations (65) and (66) from Lemma 27, we conclude:

$$\left| \left\langle W_{L:1}, \frac{d}{dt}W_{L:1} \right\rangle - \left\langle W'_{L:1}, \frac{d}{dt}W'_{L:1} \right\rangle \right| \leq L^3(R+1)^{4L-3} \cdot \sqrt{\epsilon} + 5L^3(R+1)^{4L-3} \cdot \sqrt{\epsilon}$$
$$= 6L^3(R+1)^{4L-3} \cdot \sqrt{\epsilon}.$$

For $\left| \left\langle A_{W_{L:1}}, \frac{d}{dt}W_{L:1} \right\rangle - \left\langle A_{W'_{L:1}}, \frac{d}{dt}W'_{L:1} \right\rangle \right|$, similar proof steps establish the same upper bound since $\left\| A_{W_{L:1}} - A_{W'_{L:1}} \right\|_F = \|W_{L:1} - W'_{L:1}\|_F$ and $\left\| A_{W'_{L:1}} \right\|_F = \|W'_{L:1}\|_F$. $\square$

**Lemma 29.** *In the context of the proof for Equation* (21) *(Subappendix G.13.1):*

$$\left| \frac{\|W_{L:1}\|_F^2 \left\langle W_{L:1}, \frac{d}{dt}W_{L:1} \right\rangle}{\sigma_1(W_{L:1})^2 - \sigma_2(W_{L:1})^2} - \frac{\|W'_{L:1}\|_F^2 \left\langle W'_{L:1}, \frac{d}{dt}W'_{L:1} \right\rangle}{\sigma_1(W'_{L:1})^2 - \sigma_2(W'_{L:1})^2} \right| \leq \frac{384L^3(R+1)^{8L-3}}{(1 - \sqrt{\ell_{init}})^4} \cdot \sqrt{\epsilon}.$$

*Proof.* By Equation (68), it suffices to show that:

$$\frac{32}{(1 - \sqrt{\ell_{init}})^4} \left| (\sigma_1(W'_{L:1})^2 - \sigma_2(W'_{L:1})^2)\alpha_{W_{L:1}} - (\sigma_1(W_{L:1})^2 - \sigma_2(W_{L:1})^2)\alpha_{W'_{L:1}} \right|$$
$$\leq \frac{384L^3(R+1)^{8L-3}}{(1 - \sqrt{\ell_{init}})^4} \cdot \sqrt{\epsilon},$$

where $\alpha_{W_{L:1}} := \|W_{L:1}\|_F^2 \langle W_{L:1}, \frac{d}{dt} W_{L:1} \rangle$, and $\alpha_{W'_{L:1}}$ is defined similarly for $W'_{L:1}$. Focusing on the expression in absolute value, adding and subtracting $(\sigma_1(W_{L:1})^2 - \sigma_2(W_{L:1})^2)\alpha_{W_{L:1}}$, and applying the triangle inequality, leads to:

$$\frac{32}{(1-\sqrt{\ell_{init}})^4} \Big[ \big| \sigma_1(W'_{L:1})^2 - \sigma_2(W'_{L:1})^2 - (\sigma_1(W_{L:1})^2 - \sigma_2(W_{L:1})^2) \big| \cdot |\alpha_{W_{L:1}}| \tag{69}$$
$$+ \big| \sigma_1(W_{L:1})^2 - \sigma_2(W_{L:1})^2 \big| \cdot \big| \alpha_{W_{L:1}} - \alpha_{W'_{L:1}} \big| \Big].$$

We consider each of these terms separately. From Equation (67) in Lemma 27 we know that $\big| \sigma_1(W'_{L:1})^2 - \sigma_2(W'_{L:1})^2 - (\sigma_1(W_{L:1})^2 - \sigma_2(W_{L:1})^2 \big| \le 2L^2(R+1)^{2L-1} \cdot \sqrt{\epsilon}$. Equation (65) and the Cauchy-Schwartz inequality give:

$$|\alpha_{W_{L:1}}| \le \|W_{L:1}\|_F^3 \cdot \left\| \frac{d}{dt} W_{L:1} \right\|_F \le 2L(R+1)^{6L-2}.$$

Additionally, $\big| \sigma_1(W_{L:1})^2 - \sigma_2(W_{L:1})^2 \big| \le \sigma_1(W_{L:1})^2 + \sigma_2(W_{L:1})^2 = \|W_{L:1}\|_F^2 \le (R+1)^{2L}$. By adding and subtracting $\|W'_{L:1}\|_F^2 \langle W_{L:1}, \frac{d}{dt} W_{L:1} \rangle$, we may upper bound $\big| \alpha_{W_{L:1}} - \alpha_{W'_{L:1}} \big|$ by:

$$\left| \|W_{L:1}\|_F^2 - \|W'_{L:1}\|_F^2 \right| \left| \left\langle W_{L:1}, \frac{d}{dt} W_{L:1} \right\rangle \right| + \|W'_{L:1}\|_F^2 \left| \left\langle W_{L:1}, \frac{d}{dt} W_{L:1} \right\rangle - \left\langle W'_{L:1}, \frac{d}{dt} W'_{L:1} \right\rangle \right|$$

$$\le \big| \|W_{L:1}\|_F + \|W'_{L:1}\|_F \big| \cdot \big| \|W_{L:1}\|_F - \|W'_{L:1}\|_F \big| \cdot \|W_{L:1}\|_F \left\| \frac{d}{dt} W_{L:1} \right\|_F$$

$$+ \|W'_{L:1}\|_F^2 \left| \left\langle W_{L:1}, \frac{d}{dt} W_{L:1} \right\rangle - \left\langle W'_{L:1}, \frac{d}{dt} W'_{L:1} \right\rangle \right|$$

$$\le 2(R+1)^{2L} \big| \|W_{L:1}\|_F - \|W'_{L:1}\|_F \big| \cdot \left\| \frac{d}{dt} W_{L:1} \right\|_F$$

$$+ (R+1)^{2L} \left| \left\langle W_{L:1}, \frac{d}{dt} W_{L:1} \right\rangle - \left\langle W'_{L:1}, \frac{d}{dt} W'_{L:1} \right\rangle \right|$$

$$\le 2L^3(R+1)^{6L-3} \cdot \sqrt{\epsilon} + 6L^3(R+1)^{6L-3} \cdot \sqrt{\epsilon}$$

$$= 8L^3(R+1)^{6L-3} \cdot \sqrt{\epsilon}.$$

where the third transition is due to Equations (60), (65), and Lemma 28. Putting it all together, the expression in Equation (69) is upper bounded by:

$$\frac{32}{(1-\sqrt{\ell_{init}})^4} \Big[ 4L^3(R+1)^{8L-3} \cdot \sqrt{\epsilon} \cdot + 8L^3(R+1)^{8L-3} \cdot \sqrt{\epsilon} \Big]$$

$$= \frac{384L^3(R+1)^{8L-3}}{(1-\sqrt{\ell_{init}})^4} \cdot \sqrt{\epsilon}.$$

$\square$

**Lemma 30.** *In the context of the proof for Equation* (21) *(Subappendix G.13.1):*

$$\left| \frac{2 \det(W_{L:1}) \langle A_{W_{L:1}}, \frac{d}{dt} W_{L:1} \rangle}{\sigma_1(W_{L:1})^2 - \sigma_2(W_{L:1})^2} - \frac{2 \det(W'_{L:1}) \langle A_{W'_{L:1}}, \frac{d}{dt} W'_{L:1} \rangle}{\sigma_1(W'_{L:1})^2 - \sigma_2(W'_{L:1})^2} \right| \le \frac{768L^3(R+1)^{8L-3}}{(1-\sqrt{\ell_{init}})^4} \cdot \sqrt{\epsilon}.$$

*Proof.* The proof follows a line similar to that of Lemma 29. Applying the bound from Equation (68) in Lemma 27, we arrive at the following upper bound for the left hand side above:

$$\frac{64}{(1-\sqrt{\ell_{init}})^4} \left| (\sigma_1(W'_{L:1})^2 - \sigma_2(W'_{L:1})^2)\beta_{W_{L:1}} - (\sigma_1(W_{L:1})^2 - \sigma_2(W_{L:1})^2)\beta_{W'_{L:1}} \right|,$$

where $\beta_{W_{L:1}} := \det(W_{L:1}) \langle A_{W_{L:1}}, \frac{d}{dt} W_{L:1} \rangle$, and $\beta_{W'_{L:1}}$ is defined similarly for $W'_{L:1}$. Focusing on the expression in absolute value, we add and subtract $(\sigma_1(W_{L:1})^2 - \sigma_2(W_{L:1})^2)\beta_{W_{L:1}}$ and apply the triangle inequality:

$$\frac{64}{(1-\sqrt{\ell_{init}})^4} \Big[ \big| \sigma_1(W'_{L:1})^2 - \sigma_2(W'_{L:1})^2 - (\sigma_1(W_{L:1})^2 - \sigma_2(W_{L:1})^2) \big| \cdot |\beta_{W_{L:1}}| \tag{70}$$
$$+ \big| \sigma_1(W_{L:1})^2 - \sigma_2(W_{L:1})^2 \big| \cdot \big| \beta_{W_{L:1}} - \beta_{W'_{L:1}} \big| \Big].$$

We treat each of the four terms separately. From Equation (67) in Lemma 27 we know that:

$$\left| \sigma_1(W'_{L:1})^2 - \sigma_2(W'_{L:1})^2 - (\sigma_1(W_{L:1})^2 - \sigma_2(W_{L:1})^2 \right| \leq 2L^2(R+1)^{2L-1} \cdot \sqrt{\epsilon}.$$

Since $|\det(W_{L:1})| = \sigma_1(W_{L:1}) \cdot \sigma_2(W_{L:1}) \leq \|W_{L:1}\|_F^2$, the Cauchy-Schwartz inequality and Equation (65) from Lemma 27 give:

$$|\beta_{W_{L:1}}| \leq \|W_{L:1}\|_F^3 \left\| \frac{d}{dt} W_{L:1} \right\|_F \leq 2L(R+1)^{6L-2}.$$

Furthermore, $\left| \sigma_1(W_{L:1})^2 - \sigma_2(W_{L:1})^2 \right| \leq \|W_{L:1}\|_F^2 \leq (R+1)^{2L}$. The remaining term is $\left| \beta_{W_{L:1}} - \beta_{W'_{L:1}} \right|$. Adding and subtracting $\det(W'_{L:1}) \left\langle A_{W_{L:1}}, \frac{d}{dt} W_{L:1} \right\rangle$, it can be upper bounded by:

$$|\det(W_{L:1}) - \det(W'_{L:1})| \left| \left\langle A_{W_{L:1}}, \frac{d}{dt} W_{L:1} \right\rangle \right|$$

$$+ |\det(W'_{L:1})| \left| \left\langle A_{W_{L:1}}, \frac{d}{dt} W_{L:1} \right\rangle - \left\langle A_{W'_{L:1}}, \frac{d}{dt} W'_{L:1} \right\rangle \right|.$$

From Lemma 28 we have that $\left| \left\langle A_{W_{L:1}}, \frac{d}{dt} W_{L:1} \right\rangle - \left\langle A_{W'_{L:1}}, \frac{d}{dt} W'_{L:1} \right\rangle \right| \leq 6L^3(R+1)^{4L-3} \cdot \sqrt{\epsilon}$. Furthermore, Theorem 2.12 in [42] implies that:

$$|\det(W_{L:1}) - \det(W'_{L:1})| \leq 2 \|W_{L:1} - W'_{L:1}\|_F \max\{\|W_{L:1}\|_F, \|W'_{L:1}\|_F\}.$$

Thus, with the use of Equations (60) and (65), we have:

$$\begin{aligned}
\left| \beta_{W_{L:1}} - \beta_{W'_{L:1}} \right| &\leq 2L(R+1)^{4L-2} \cdot |\det(W_{L:1}) - \det(W'_{L:1})| + 6L^3(R+1)^{6L-3} \cdot \sqrt{\epsilon} \\
&\leq 2L^3(R+1)^{6L-3} \cdot \sqrt{\epsilon} + 6L^3(R+1)^{6L-3} \cdot \sqrt{\epsilon} \\
&= 8L^3(R+1)^{6L-3} \cdot \sqrt{\epsilon}.
\end{aligned}$$

Put together, the expression in Equation (70) is upper bounded by:

$$\begin{aligned}
&\frac{64}{(1 - \sqrt{\ell_{init}})^4} \left[ 4L^3(R+1)^{8L-3} \cdot \sqrt{\epsilon} + 8L^3(R+1)^{8L-3} \cdot \sqrt{\epsilon} \right] \\
&= \frac{768}{(1 - \sqrt{\ell_{init}})^4} L^3(R+1)^{8L-3} \cdot \sqrt{\epsilon}.
\end{aligned}$$

$\square$

### G.13.2   Proof of Equations (22), (23) and (24) (if weight matrices are not bounded by $R$)

Assume that there exists $t \in [0, T]$ with $\max_{l \in [L]} \|W_l(t)\|_F > R$. We examine the initial time at which the Frobenius norm of one of the weight matrices reaches $R$. Formally, define:

$$\bar{t} := \inf \left\{ \hat{t} | \hat{t} \in [0, t] \text{ and } \exists l \in [L] \text{ s.t. } \|W_l(\hat{t})\|_F = R \right\}.$$

Since $R \geq \max_{l \in [L]} \|W_l(0)\|_F$ — implied by the assumption on $\epsilon$ — and $W_1(t), W_2(t), \ldots, W_L(t)$ are continuous functions of $t$, the set on the right hand side is non-empty and compact. Hence, $\bar{t}$ is well defined (i.e. $-\infty < \bar{t} < \infty$), with $\max_{l \in [L]} \|W_l(\bar{t})\|_F = R$.

Next, we derive a lower bound on $|(W_{L:1}(\bar{t}))_{1,1}|$ — the absolute value of the unobserved entry. According to Lemmas 3 and 1, there exist balanced matrices $W'_1, W'_2, \ldots, W'_L$ satisfying:

$$\|W_l(\bar{t}) - W'_l\|_F \leq (l-1) \cdot \sqrt{\epsilon} \quad , \forall l \in [L]. \tag{71}$$

Applying Lemma 12, and noticing that $\max_{l \in [L]} \|W'_l\|_F \leq R + (L-1) \cdot \sqrt{\epsilon} \leq R+1$, we bound the distance between the induced product matrices:

$$\|W_{L:1}(\bar{t}) - W'_{L:1}\|_F \leq (R+1)^{L-1} \cdot \sum_{l=1}^{L} (l-1) \cdot \sqrt{\epsilon} \leq \frac{1}{2} L^2 (R+1)^{L-1} \cdot \sqrt{\epsilon}. \tag{72}$$

Since $W'_1, W'_2, \ldots, W'_L$ are balanced, they have the same singular values. In particular, this means that $\|W'_l\|_F = \|W'_{l'}\|_F$ for any $l, l' \in [L]$. Additionally, by Lemma 8 we know that $\sigma_1(W'_{L:1}) = \sigma_1(W'_1)^L$. Thus:

$$
\begin{aligned}
\|W'_{L:1}\|_F &\geq \sigma_1(W'_{L:1}) \\
&= \sigma_1(W'_1)^L \\
&\geq \left( \frac{1}{\sqrt{2}} \|W'_1\|_F \right)^L \\
&= 2^{-\frac{L}{2}} \max_{l \in [L]} \|W'_l\|_F^L \\
&\geq 2^{-\frac{L}{2}} \cdot \left( R - (L-1) \cdot \sqrt{\epsilon} \right)^L,
\end{aligned}
$$

where the last transition is by Equation (71) and the fact that $\max_{l \in [L]} \|W_l(\bar{t})\|_F = R$. The inequality above, combined with Equation (72), leads to:

$$
\begin{aligned}
\|W_{L:1}(\bar{t})\|_F &\geq \|W'_{L:1}\|_F - \frac{1}{2} L^2 (R+1)^{L-1} \cdot \sqrt{\epsilon} \\
&\geq 2^{-\frac{L}{2}} \cdot \left( R - (L-1) \cdot \sqrt{\epsilon} \right)^L - \frac{1}{2} L^2 (R+1)^{L-1} \cdot \sqrt{\epsilon}.
\end{aligned}
$$

From the definition of $R$ it holds that $\epsilon \leq \min \left\{ \left( \frac{(1-1/\sqrt{2}) \cdot R}{L-1} \right)^2, \frac{(R/2)^{2L}}{L^4 (R+1)^{2L-2}} \right\}$, and therefore:

$$
\|W_{L:1}(\bar{t})\|_F \geq \left( \frac{R}{2} \right)^L - \frac{1}{2} \left( \frac{R}{2} \right)^L = \frac{R^L}{2^{L+1}}. \tag{73}
$$

Recall that $\ell_{init} < \ell(0) = 1$. The loss during gradient flow is monotonically non-increasing with respect to time (Lemma 17), hence, $\ell(W_{L:1}(\bar{t})) < 1$. This leads to the following bounds on the observed entries:

$$
|(W_{L:1}(\bar{t}))_{1,2}| \leq 1 + \sqrt{2} \quad, \quad |(W_{L:1}(\bar{t}))_{2,1}| \leq 1 + \sqrt{2} \quad, \quad |(W_{L:1}(\bar{t}))_{2,2}| \leq \sqrt{2}. \tag{74}
$$

Applying these bounds, we obtain:

$$
\|W_{L:1}(\bar{t})\|_F \leq |(W_{L:1}(\bar{t}))_{1,1}| + \sqrt{2(1+\sqrt{2})^2 + 2} \leq |(W_{L:1}(\bar{t}))_{1,1}| + 4.
$$

Combined with Equation (73), we have that:

$$
|(W_{L:1}(\bar{t}))_{1,1}| \geq \frac{R^L}{2^{L+1}} - 4. \tag{75}
$$

We are now in a position to establish Equations (22), (23) and (24). Starting with Equation (22), for any quasi-norm $\|\cdot\|$ it holds that:

$$
\|W_{L:1}(\bar{t})\| \geq \frac{1}{c_{\|\cdot\|}} \left\| (W_{L:1}(\bar{t}))_{1,1} \mathbf{e}_1 \mathbf{e}_1^\top \right\| - \left\| W_{L:1}(\bar{t}) - (W_{L:1}(\bar{t}))_{1,1} \mathbf{e}_1 \mathbf{e}_1^\top \right\|, \tag{76}
$$

where $c_{\|\cdot\|} \geq 1$ is a constant for which $\|\cdot\|$ satisfies the weakened triangle inequality (see Footnote 2). Subsequent applications of the weakened triangle inequality, together with homogeneity of $\|\cdot\|$ and the bounds on the observed entries (Equation (74)), give:

$$
\left\| W_{L:1}(\bar{t}) - (W_{L:1}(\bar{t}))_{1,1} \mathbf{e}_1 \mathbf{e}_1^\top \right\| \leq 8 c_{\|\cdot\|}^2 \max_{i,j \in \{1,2\}} \left\| \mathbf{e}_i \mathbf{e}_j^\top \right\|.
$$

Plugging the inequality above into Equation (76), and lower bounding $|(W_{L:1}(\bar{t}))_{1,1}|$ according to Equation (75), we have:

$$
\begin{aligned}
\|W_{L:1}(\bar{t})\| &\geq \frac{\left\| \mathbf{e}_1 \mathbf{e}_1^\top \right\|}{c_{\|\cdot\|}} \cdot |(W_{L:1}(\bar{t}))_{1,1}| - 8 c_{\|\cdot\|}^2 \max_{i,j \in \{1,2\}} \left\| \mathbf{e}_i \mathbf{e}_j^\top \right\| \\
&\geq \frac{\left\| \mathbf{e}_1 \mathbf{e}_1^\top \right\|}{2^{L+1} c_{\|\cdot\|}} \cdot R^L - 12 c_{\|\cdot\|}^2 \max_{i,j \in \{1,2\}} \left\| \mathbf{e}_i \mathbf{e}_j^\top \right\|.
\end{aligned} \tag{77}
$$

The assumption on the size of $\epsilon$ implies that $R \geq 32$. Hence:

$$R \geq \begin{cases} \frac{1}{2}\left[\frac{(1-\sqrt{\ell_{init}})^4}{2^{16}} \cdot \ln\left(\frac{1}{\epsilon}\right)\right]^{1/6} & \text{, if depth } L = 2 \\[2ex] \frac{1}{2}\left[\frac{(1-\sqrt{\ell_{init}})^4}{2^{2L+8}L^4} \cdot \frac{1}{\epsilon^{1/32}}\right]^{1/(4L-2)} & \text{, if depth } L \geq 3 \end{cases}. \tag{78}$$

Applying Equation (78) to Equation (77), we obtain:

$$\|W_{L:1}(\bar{t})\| \geq \begin{cases} \frac{\|\mathbf{e}_1\mathbf{e}_1^\top\|(1-\sqrt{\ell_{init}})^{4/3}}{2^{31/3}c_{\|\cdot\|}} \cdot \ln\left(\frac{1}{\epsilon}\right)^{1/3} - 12c_{\|\cdot\|}^2 \max_{i,j\in\{1,2\}}\left\|\mathbf{e}_i\mathbf{e}_j^\top\right\| & \text{, if depth } L = 2 \\[3ex] \frac{\|\mathbf{e}_1\mathbf{e}_1^\top\|(1-\sqrt{\ell_{init}})^{\frac{2L}{2L-1}}}{2^{\frac{5L^2+4L-1}{2L-1}}L^{\frac{2L}{2L-1}}c_{\|\cdot\|}} \cdot \epsilon^{-\frac{L}{128L-64}} - 12c_{\|\cdot\|}^2 \max_{i,j\in\{1,2\}}\left\|\mathbf{e}_i\mathbf{e}_j^\top\right\| & \text{, if depth } L \geq 3 \end{cases}.$$

For clarity, we simplify the exponents in the lower bound above. In the case of $L = 2$, we use the fact that $2^{31/3} \leq 2^{11}$. For $L \geq 3$, noticing that $2L/(2L-1) \leq 6/5$ and $(5L^2 + 4L - 1)/(2L-1) \leq 4L$ completes the proof of Equation (22).

Next, we derive the upper bound for effective rank (Equation (23)). Let $h(p) := -p \cdot \ln(p) - (1-p) \cdot \ln(1-p)$ be the binary entropy function, defined over $[0,1]$. Recall that the effective rank of $W_{L:1}(\bar{t})$ is defined to be $\text{erank}(W_{L:1}(\bar{t})) := \exp\{h(\rho_2(W_{L:1}(\bar{t})))\}$, where $\rho_2(W_{L:1}(\bar{t})) := \sigma_2(W_{L:1}(\bar{t}))/(\sigma_1(W_{L:1}(\bar{t})) + \sigma_2(W_{L:1}(\bar{t})))$. Since the exponent function is convex, it is upper bounded on the interval $[0, \ln(2)]$ by the linear function that intersects it at these points. That is:

$$\text{erank}(W_{L:1}(\bar{t})) \leq 1 + \frac{1}{\ln(2)} \cdot h(\rho_2(W_{L:1}(\bar{t}))).$$

From Lemma 9 we know that $h(\rho_2(W_{L:1}(\bar{t}))) \leq 2\sqrt{\rho_2(W_{L:1}(\bar{t}))}$. Combined with the fact that $\inf_{W'\in\mathcal{S}}\text{erank}(W') = 1$ (Proposition 2), this leads to the following upper bound:

$$\text{erank}(W_{L:1}(\bar{t})) \leq \inf_{W'\in\mathcal{S}}\text{erank}(W') + \frac{2}{\ln(2)} \cdot \sqrt{\rho_2(W_{L:1}(\bar{t}))}. \tag{79}$$

We now lower bound $\rho_1(W_{L:1}(\bar{t})) := \sigma_1(W_{L:1}(\bar{t}))/(\sigma_1(W_{L:1}(\bar{t})) + \sigma_2(W_{L:1}(\bar{t})))$ as follows:

$$\begin{aligned} \rho_1(W_{L:1}(\bar{t})) &\geq \frac{\sigma_1(W_{L:1})}{\sigma_1(W_{L:1}) + 2^{L+2}/R^L + \sqrt{2\ell(W_{L:1}(\bar{t}))}} \\[1ex] &= 1 - \frac{2^{L+2}/R^L + \sqrt{2\ell(W_{L:1}(\bar{t}))}}{\sigma_1(W_{L:1}) + 2^{L+2}/R^L + \sqrt{2\ell(W_{L:1}(\bar{t}))}} \\[1ex] &\geq 1 - \frac{2^{L+2}/R^L + \sqrt{2\ell(W_{L:1}(\bar{t}))}}{R^L/2^{L+2} + 2^{L+2}/R^L} \\[1ex] &\geq 1 - \frac{2^{L+2}/R^L + \sqrt{2\ell(W_{L:1}(\bar{t}))}}{R^L/2^{L+2}} \\[1ex] &= 1 - \left(\frac{2^{L+2}}{R^L}\right)^2 - \sqrt{2\ell(W_{L:1}(\bar{t}))} \cdot \frac{2^{L+2}}{R^L}, \end{aligned}$$

where the first and second inequalities are due to Lemma 31. Since $2^{L+2}/R^L < 1$ and $\ell(W_{L:1}(\bar{t})) < 1$, it holds that $\rho_1(W_{L:1}(\bar{t})) \geq 1 - 2^{L+4}/R^L$, or equivalently, $\rho_2(W_{L:1}(\bar{t})) \leq 2^{L+4}/R^L$. Going back to Equation 79, we have:

$$\text{erank}(W_{L:1}(\bar{t})) \leq \inf_{W'\in\mathcal{S}}\text{erank}(W') + \frac{2^{L/2+3}}{\ln(2)\cdot R^{L/2}}.$$

Applying the lower bound on $R$ from Equation (78), we arrive at:

$$\text{erank}(W_{L:1}(\bar{t})) \leq \begin{cases} \inf_{W'\in\mathcal{S}}\text{erank}(W') + \frac{2^{23/3}}{\ln(2)(1-\sqrt{\ell_{init}})^{2/3}} \cdot \ln\left(\frac{1}{\epsilon}\right)^{-1/6} & \text{, if depth } L = 2 \\[3ex] \inf_{W'\in\mathcal{S}}\text{erank}(W') + \frac{2^{\frac{5L^2+14L-6}{4L-2}}L^{\frac{L}{2L-1}}}{\ln(2)(1-\sqrt{\ell_{init}})^{\frac{L}{2L-1}}} \cdot \epsilon^{\frac{L}{256L-128}} & \text{, if depth } L \geq 3 \end{cases}.$$

In the case of $L = 2$, we simplify the upper bound using the fact that $2^{23/3}/\ln(2) \leq 2^9$. For $L \geq 3$, noticing that $L/(2L-1) \leq 1$ and $(5L^2 + 14L - 6)/(4L - 2) \leq 2L + 4$ establishes Equation (23).

Finally, we turn our attention to the upper bound for distance from infimal rank (Equation (24)). By Proposition 2, the infimal rank of $\mathcal{S}$ is 1. Therefore, Lemma 31 yields:

$$D(W_{L:1}(\bar{t}), \mathcal{M}_1) = \sigma_2(W_{L:1}(\bar{t})) \leq \frac{2^{L+2}}{R^L} + \sqrt{2\ell(W_{L:1}(\bar{t}))}.$$

Applying the lower bound on $R$ from Equation (78), we arrive at:

$$D(W_{L:1}(\bar{t}), \mathcal{M}_{\mathrm{irank}(\mathcal{S})}) \leq \begin{cases} \frac{2^{34/3}}{(1-\sqrt{\ell_{init}})^{4/3}} \cdot \ln\left(\frac{1}{\epsilon}\right)^{-1/3} + \sqrt{2\ell(W_{L:1}(\bar{t}))} & , \text{if depth } L = 2 \\ \frac{2^{\frac{5L^2+6L-2}{2L-1}} L^{\frac{2L}{2L-1}}}{(1-\sqrt{\ell_{init}})^{\frac{2L}{2L-1}}} \cdot \epsilon^{\frac{L}{128L-64}} + \sqrt{2\ell(W_{L:1}(\bar{t}))} & , \text{if depth } L \geq 3 \end{cases}.$$

In the case of $L = 2$, we simplify the upper bound using the fact that $2^{34/3} \leq 2^{12}$. For $L \geq 3$, noticing that $2L/(2L-1) \leq 6/5$ and $(5L^2 + 6L - 2)/(2L - 1) \leq 3L + 4$ concludes the proof of Equation (24). $\qquad\square$

### G.13.2.1 Auxiliary Lemma

**Lemma 31.** *In the context of the proof for Equations (22), (23) and (24) (Subappendix G.13.2), the singular values of $W_{L:1}(\bar{t})$ fulfill:*

$$\sigma_1(W_{L:1}(\bar{t})) \geq \frac{R^L}{2^{L+2}} - \sqrt{2\ell(W_{L:1}(\bar{t}))} \quad , \quad \sigma_2(W_{L:1}(\bar{t})) \leq \frac{2^{L+2}}{R^L} + \sqrt{2\ell(W_{L:1}(\bar{t}))}. \qquad (80)$$

*Proof.* Define $W_{\mathcal{S}} := \begin{pmatrix} (W_{L:1}(\bar{t}))_{1,1} & 1 \\ 1 & 0 \end{pmatrix}$, the orthogonal projection of $W_{L:1}(\bar{t})$ onto the solution set $\mathcal{S}$ (Equation (7)). Suppose that $(W_{L:1}(\bar{t}))_{1,1} \geq 0$. The largest singular value of $W_{\mathcal{S}}$ can be written as:

$$\sigma_1(W_{\mathcal{S}}) = \max_{i=1,2} |\lambda_i(W_{\mathcal{S}})| = \frac{1}{2}\left((W_{L:1}(\bar{t}))_{1,1} + \sqrt{(W_{L:1}(\bar{t}))_{1,1}^2 + 4}\right).$$

Thus, lower bounding $(W_{L:1}(\bar{t}))_{1,1}$ with Equation (75), and noticing that $(W_{L:1}(\bar{t}))_{1,1}^2 + 4 \geq 16$, leads to $\sigma_1(W_{\mathcal{S}}) \geq R^L/2^{L+2}$. Analogously, for the smallest singular value of $W_{\mathcal{S}}$ we have:

$$\begin{aligned} \sigma_2(W_{\mathcal{S}}) &= \min_{i=1,2} |\lambda_i(W_{\mathcal{S}})| \\ &= \frac{1}{2}\left(\sqrt{(W_{L:1}(\bar{t}))_{1,1}^2 + 4} - (W_{L:1}(\bar{t}))_{1,1}\right) \\ &= \frac{2}{\sqrt{(W_{L:1}(\bar{t}))_{1,1}^2 + 4} + (W_{L:1}(\bar{t}))_{1,1}} \\ &\leq 2^{L+2}/R^L, \end{aligned}$$

where in the third transition we made use of the identity $a - b = \frac{a^2-b^2}{a+b}$ for $a, b \in \mathbb{R}$ such that $a + b \neq 0$. Corollary 8.6.2 in [33] yields the following singular values perturbation bound:

$$|\sigma_i(W_{L:1}(\bar{t})) - \sigma_i(W_{\mathcal{S}})| \leq \|W_{L:1}(\bar{t}) - W_{\mathcal{S}}\|_F \quad , i = 1, 2.$$

Equation (80) then readily follows from the fact that $\|W_{L:1}(\bar{t}) - W_{\mathcal{S}}\|_F = \sqrt{2\ell(W_{L:1}(\bar{t}))}$.

By similar arguments, Equation (80) holds when $(W_{L:1}(\bar{t}))_{1,1} < 0$ as well. $\qquad\square$