[Reviews · NeurIPS 2020]

Review 1

Summary and Contributions: Reconstruction of a low-rank matrix from its linear measurements is a canonical problem in machine learning and signal processing. There has been an intense effort to establish theoretical guarantees and design efficient algorithms for solving these problems. Of these, the most prominent two methods are: 1- Convex optimization approach - Nuclear-norm regularization. 2- Nonconvex factorization approach (aka Burer Monteiro splitting) - Reparametrization of the matrix variable X=UV’ for some tall matrices U and V, and optimization over the factors U and V. In particular, the non-convex factorization approach has received increasing attention due to the reduced arithmetic and storage costs. Recently, Gunasekar et al. (2017) reported a surprising observation, that the non-convex factorization approach (when solved with gradient descent) generalizes (i.e., recovers the low-rank matrix of interest) even when the factors U and V are full dimensional (i.e., not tall, hence UV’ does not impose an explicit low-rank structure). Note that the problem that we are considering is underdetermined, and there are infinitely many trivial solutions. Regardless, the gradient descent converges to the solution of interest. The existing theory fails to explain this phenomenon. Based on the empirical evidence, they conjectured that gradient descent when applied to factorized least-squares formulation converges to exhibits an implicit bias towards the minimum nuclear-norm solution (under some technical details regarding the step-size, initialization, convergence, etc.). In a follow-up paper, Arora et al. (2019) observed numerical setups where this conjecture fails, and proposed an alternative conjecture, claiming that the implicit bias is not towards the minimum nuclear-norm but the minimum rank. The paper under review joins this discussion, by presenting a simple example of (2 x 2) dimensional matrix completion problem where the gradient descent on factorized least-squares formulation drives the nuclear norm (and all norms and pseudonorms) towards infinity while decreasing the rank to the minimum, with probability 0.5. Hence, this example can be said to support Arora et al/ (2019). Along with the theoretical analysis, the paper also presents an empirical demonstration of this example. The paper also presents a procedure to design similar examples in higher dimensions. Moreover, the paper presents numerical experiments in tensor factorization and demonstrates (empirically) that gradient descent produces low-rank (in the context of tensors) solutions. Matrix factorization has a natural bond with deep learning, we can view matrix factorization as a depth-2 linear neural network. Similarly, tensor factorization corresponds to a certain type of nonlinear neural network. Based on these connections and the observations in the matrix and tensor factorization problems, the paper argues that “formalizing notions of rank … may be key to explaining generalization in deep learning.” --- update after the authors' response --- I have read the authors' response and other reviewers' comments carefully and taken them into account in my final evaluation. I kept my score unchanged.

Strengths: The topic is an excellent match for NeurIPS. Implicit regularization in matrix factorization is a young research topic (dates back to the 2017 paper by Gunasekar et al.) and an important body of prior work appeared in top machine learning conferences. It has a natural bond with implicit regularization in deep learning. The presentation quality is very high. I enjoyed reading the paper. The analysis is relatively simple to follow. Implicit regularization in matrix factorization (and in deep learning) is a hot topic, and I would expect this work to attract some attention at the conference and after.

Weaknesses: First of all, as already admitted in the paper by the authors, although the “counter”-example presented in this paper contextually supports low-rank implicit regularization against the low-nuclear norm, theoretically speaking, it does not disprove the conjecture of Gunasekar et al. (2017). This is because the conjecture has a technical condition, that requires the estimate X=UV’ to converge towards a solution to the original problem (consistent with the linear measurements.) Only then, the conjecture hypothesizes that gradient descent exhibits an implicit regularization towards the low-nuclear norm solution. On the other hand, X=UV’ in the example introduced in the paper under review diverges towards infinity. The authors chose to present this as a strength instead, arguing that the example disqualifies not only the nuclear norm but all norms and quasi norms (since all go to infinity when X diverges) from explaining the implicit regularization. In my opinion, however, an example where the variable is diverging towards infinity is practically less interesting.

Correctness: The claims seem correct to me. I have read through the main text very carefully. I also read the supplementary material but I did not check the proofs step by step. Overall, I did not notice any technical issues. The empirical test setup is clear and fair. Also, the code is provided in the supplements for reproducibility, but I did not check or test the code.

Clarity: The presentation quality is high, the paper is well written. However, in my opinion, the title should reflect the specific focus on “matrix factorization”. I particularly enjoyed that all technical results are presented along with a proof sketch. This makes it easier to see the complete image.

Relation to Prior Work: This paper is a follow-up on the prior works by Gunasekar et al. (2017) and Arora et al. (2019). Essentially, it joins into the discussion on potential explanations of the implicit regularization in matrix factorization. Gunasekar et al. (2017) conjectured that this implicit bias can be characterized as the minimum nuclear-norm solution. Two years later, Arora et al. (2019) proposed the minimization of rank as an alternative view in the lights of some numerical and theoretical analysis. The current paper presents a simple example that supports the latter. These immediate connections are discussed in a clear way in the paper. Some of the existing results on the topic are already mentioned in the introduction. A more detailed discussion on the related work is deferred to the appendix B. This appendix has good coverage of the existing literature on the implicit regularization in matrix factorization. The paper omits the overview of the broader literature on more conventional methods for matrix factorization (low-rank factorization, nuclear norm regularization, etc.) but I agree with the authors that this is out of the scope of this paper. The appendix also covers some works on the “implicit regularization in deep learning.” I am not an expert on this particular context, but to me, the coverage of this section seems adequate.

Reproducibility: Yes

Additional Feedback: Some other comments: - In my opinion, the main topic and the focus of presentation in this paper should be the “implicit regularization in matrix factorization.” Although the connections with deep learning are valid, I feel like that it would be clearer to keep these connections as remarks, rather than the main theme. - All solutions to the example in Section 3.1 is negative definite by definition. We already know from Arora et al. (2019) that the continuous dynamics of factorized gradient flow does not allow the sign change, i.e., the matrix remains psd (resp, nsd) if initialized as psd (resp., nsd). This plays a crucial role here. At the same time, the original problem template in Gunasekar et al., (2017) considers least-squares over psd cone, and considers symmetric factorization X=UU’, hence it avoids the sign discrepancy. Is it possible to extend your “counter”-example for the symmetric factorization problem?


Review 2

Summary and Contributions: The paper looks at the effect of running gradient descent from a small initialization on the nonconvex loss landscape of matrix completion in the overparameterized setting. They show that running gradient descent finds a low rank solution, but at the same time all the norms go to infinity. They thus settle down in the negative, a conjecture due to Gunasekar et al. In my opinion this result is important because Not all trained solutions generalize the same in deep learning (with non-linearities). Understanding the effect of optimization algorithms and reparameterization for simpler problems like matrix completion, etc can hopefully guide our search to understand better the same in more complex settings. The result is interesting for the matrix completion task as well, and suggests that if the underlying matrix (which we want to recover) was low rank, then optimizing a norm regularized square loss will not succeed.

Strengths: The paper is well written. All the arguments are based on a specific 2*2 matrix completion problem. Based on this example, the authors were able to come up with a fairly neat story. They also do some experiments justifying the theoretical constraints.

Weaknesses: I have a few general comments: 1. Is there any understanding of whether the trend will hold in the large learning rate setting. It seems that the paper approaches the problem for implicit regularization of gradient flow instead of gradient descent. Even though the experiments are done with large learning rate, I am not sure what would be the range of LR for this trend to hold. 2. What about the effect of dimension? Would the same effect, and the subsequent resolution of the conjecture due to Gunasekar et al. also hold in higher dimension, d >> 2 ? Can one embed the low dimensional construction in the higher dimensional setting? 3. Are there norms for which the constants a, b, and c as used in Theorem 1 will depend on the dimension?

Correctness: The proofs seem to be correct.

Clarity: Yes, the paper is well written.

Relation to Prior Work: Relation to prior work has been well explored. In fact, the paper tries to resolve a conjecture and some confusion around it in the prior work.

Reproducibility: Yes

Additional Feedback: I think the authors should not use the term deep learning for matrix factorization with linear neural networks. I think it attracts the wrong audience, and is not a good thing for the community :). ====== Post author feedback and discussion ====== I have read all the reviews and the feedback. I am decreasing my score by 1 point because (as replied by the authors in the feedback), the constants a, b, c, and d in Theorem 1 will depend on the dimension. Further, it is not clear why l_t should go to 0 for high rank matrices. This undermines the main result for high dimensional problems, understanding which is the ultimate goal of this endeavor.


Review 3

Summary and Contributions: The paper studies the ability to argue about implicit bias through (quasi-)norms for matrix completion problems. The authors examine a simple instance where one aims at completing a missing entry for a 2x2 matrix by fitting an L-layer linear NN. It is shown that (w.h.p.) for gradient flows corresponding to this problem, any (quasi-)norm grows in inverse proportion to the loss function. This shows that of the infinite set of global solutions, gradient flows converge to those with large (indeed, infinite) norm. The authors further propose two quantities which serve as proxies for the rank as an alternative implicit bias for this specific setting. The authors provide empirical results aimed at corroborating the theoretical analysis to the (discrete-time) gradient descent algorithm for tensor completion. --- update after the authors' response --- I have read the authors' response and other reviewers' comments carefully and taken them into account in my final evaluation. I kept my score unchanged.

Strengths: The recent surge in soluble non-convex problems led to a lot of interest in characterizing implicit bias. One dominating hypothesis (inspired by, e.g., least square problems) is that implicit bias is related to some form of implicit norm. The paper provides a very simple instance where no norm seems to give a good description of the solutions favored by gradient flows, and also provide an alternative rank-related explanation. The empirical results for tensor completion nicely demonstrate that the missing entry does indeed grow as should be expected by the analysis of the matrix completion.

Weaknesses: Although the question addressed by the paper is significant, it was shown already, though in different setting, that norms cannot provide a good explanation for implicit bias, e.g., "Can Implicit Bias Explain Generalization?". In that respect, the contribution of the paper is providing a simple instance for this under the matrix completion setting.

Correctness: The theoretical claims seem sound, as well as the presentation of the empirical results.

Clarity: The paper reads relatively well. I find some parts repetitive. See minor comments below.

Relation to Prior Work: As mentioned earlier, the recent work "Can Implicit Bias Explain Generalization?" and references therein (or other works which may refute norm-based implicit bias such as A. Suggala et al. (2018)) should be referred to and contrasted with more explicitly. It would also appear that a more comprehensive survey of recent works addressing implicit bias in other settings is required.

Reproducibility: Yes

Additional Feedback: Minor comments: -- Can you provide in the body of the paper more intuition as to why the missing entry grows? -- L47, -- [6] -- -- L113, the wording is not entirely clear, and the passage starting in this line is somewhat repetitive (and its content seemed to be repeated again in 4). -- Definition of effective rank seems to be missing


Review 4

Summary and Contributions: The paper is concerned with the matrix completion problem using matrix products (matrix factorization). The authors try to understand the behavior of norms and ranks of solutions to matrix completion discovered by optimization with gradient descent. The work is based on assumptions of initialization close to the origin and small learning rates. Two conjectures are presented (Conjecture 1 and Conjecture 2) from other works. The first one suggests that gradient descent converges to solutions with a small (minimal) nuclear norm. The second one states that when performing gradient descent on a matrix factorization there are cases where convergence of gradient descent to a solution with minimal norm is not guaranteed. These special cases are identified by a providing a specific set of observed entries. However, the paper clearly admits that Conjectures 1 and 2 are not necessarily formally contradictory due to technical subtleties (additional requirements in Conjecture 1). They are presented more as general perspectives on what kind of solutions we should expect with matrix factorization and gradient descent (small norms vs large norms). The authors support the second conjecture by providing a concrete example (a 2x2 matrix) where the norm of the solution tends to infinity. While this example is simple I think it is important that it was pointed out. Based on this observation authors suggest that gradient descent tends to find solutions with low rank rather than low norm, where rank is measured using an idea of continuous extension of rank called effective rank borrowed from other work and defined in the appendix. This claim is supported by Theorem 1. It shows that the norm of the matrix factorization is lower-bounded by an inverse of sqrt of the loss, which supports the idea that as the loss is decreased the norm increases. At the same time the theorem shows that the distance of the factorized matrix from the infimal rank of the solution set S is bounded from above by a sqrt of the loss, so as the loss is decreased the solution gets closer to solutions with lower ranks. However, authors admit that Theorem 1 applies to setting described in Subsection 3.1, which is a simple case. There is a question if the optimization converged to global optimum. Authors try to address this question in section 3.3. They provide a theoretical result for the case described in Subsection 3.1, however they diverge from the original assumption of near-origin initialization. Instead the initialization to identity is used to obtain the theoretical result in Proposition 4. I think that the divergence from the near-origin initialization should be clearly justified rather than just mentioned. It seems that other initialization schemes are left for future work which rises questions how relevant is the result in Proposition 4. The empirical justification of convergence to global optimum is provided in Section 4. Figure 1 illustrates how the absolute value of the missing entry in the 2x2 matrix increases while the loss decreases. However, it is not clear how Section 4.1 shows that the loss converges to zero (global optimum). It seems that the loss does not drop below 10^-4, which is relatively far from the precision (assuming single-precision floating numbers). Even after 2,000,000 iterations for depth 3 the loss is still around 10^-4. I think this is a concern that should be addressed. Furthermore, experiments in Section 4.1 are supposed to support the claim that as the loss decreases the absolute value of the missing entry goes to infinity. At the beginning of the section it says: "Specifically, we established that running gradient descent on the over-parameterized matrix completion objective in Equation (2), where the observed entries are those defined in Equation (6), leads the unobserved entry to diverge to infinity as loss converges to zero." However, while some runs with depth 3 and depth 4 clearly show rapid growth of the abs val of the missing entry, all of the runs in depth 2, two runs with depth 3 and 1 run with depth 4 seem to converge to a finite, well-defined value for the missing entry. Therefore, I am not sure if claims formed at the beginning of the paper are supported by experiments in Subsection 4.1 presented in Figure 1. I am also not sure if the loss minimized to 10^-4 is enough to claim that the global solution was found. Unfortunately, I do not find the experiments in Figure 1 strongly convincing. Furthermore, I think more experiments with other matrices than the 2x2 special case from the Subsection 3.1 would be very useful for this paper. Experimenting with other matrices would make the argument empirically convincing both in terms of diverging norms and in terms of discovering low-rank solutions. Experiments in Subsection 4.2 were performed on tensors. Figure 2 shows that tensors tend to low-rank solutions. It is a nice extension, but it does not compensate for missing experiments with more matrices.

Strengths: The paper is focused on an interesting question of what type of solutions are discovered by gradient descent for matrix completion problem using matrix product. The theory in the paper seems sound and provides good intuition about what kind of results should be expected. The writing is clear and easy to understand. The significance is more theoretical, because authors consider deep linear models (no non-linearities) which are significantly different than deep learning models used in practice. Nevertheless, the question of what type of solutions are obtained is interesting. I think this work is novel in trying to address Conjecture 1 and Conjecture 2. It is relevant to more theoretical part of the NeurIPS community.

Weaknesses: I think the empirical evaluation is a weak-spot of this paper. The experiments with the special case 2x2 matrix are not 100% convincing. I think the loss should be minimized to values closer to zero to claim global convergence. Moreover the rapid growth of the absolute value of the missing entry is visible only on some of the plots. Theory in the paper predicts that the abs val of the missing entry tends to infinity which is not necessarily the case looking at Figure 1. However, the experiments in Figure 1 confirm that the absolute value of the missing entry indeed grows as the loss drops, which means the norm of the solution tends to grow. This is an important example related to Conjecture 1. I was disappointed that no experiments with other matrices than the 2x2 special case were presented in the paper. I think such empirical evaluation is fundamental to make more general claims against Conjecture 1 and even more important to support Conjecture 2. Experiments with tensors are a nice addition, however the paper argues against Conjecture 1 and in favor on Conjecture 2 for regular matrices which was not supported by large enough evidence. One additional comment is that the paper at some points seems to formulate claims about deep learning models in general while experimenting only with deep linear models (no nonlinearities). I think it is important to provide at least some evidence that the observations made with deep linear models transfer to regular deep networks (with non-linearities) to support general claims about deep learning models. An experiment supporting that would be very useful.

Correctness: Some claims in the paper are too general for the provided empirical evaluation.

Clarity: Yes, the paper is well written. The writing is easily understandable and all the introduced notions are clearly defined and explained. The definition and formulas seem complete and are not missing indices or use undefined terms.

Relation to Prior Work: Yes, the paper clearly overviews the current state of research in matrix completion using matrix factorization. It clearly states existing conjectures that are central to the claims presented in the paper.

Reproducibility: Yes

Additional Feedback: -- Feedback after the rebuttal -- 1. I would like to thank authors for a clarification provided in "Theoretical result for convergence to zero loss" in their rebuttal (identity initialization with alpha scaling). 2. I also would like to thank authors for additional plots showing the behavior of the unobserved entry with longer runs in "Empirical support for convergence to zero loss" in the rebuttal. It was very helpful. 3. Thank you for addressing my questions in "Unobserved entry converging to finite value" in your rebuttal. 4. When it comes to "Experimentation with additional settings" part of the rebuttal what I had in mind was more experiments with matrices significantly different than the special case analyzed in the paper. In my opinion such experimentation would be helpful to support the general claim about the ranks of the solutions. 5. About "Extension to non-linear models" in the rebuttal: I understand authors' arguments about tensors being non-linear models. In the review I wrote: "One additional comment is that the paper at some point seems to formulate claims about deep learning models in general while experimenting only with deep linear models (no nonlinearities). I think it is important to provide at least some evidence that the observations made with deep linear models transfer to regular deep networks (with non-linearities) to support general claims about deep learning models. An experiment supporting that would be very useful." There may be a misunderstanding related to vocabulary. What I mean by 'non-linearities' are non-linear activation functions such as ReLUs or logistic functions typically found in deep learning models. By models with 'no non-linearities' I mean models that do not have non-linear activation functions. Indeed I did not mention that authors experiment with tensors. However, that would not change my main point. My main point is that the authors formulate claims about deep learning models in general, which comprises a wide set of models. Many of these models are significantly different than the ones used in the paper. Especially those best-performing ones like CNNs, ResNets etc. (which are used in practice). As authors point out themselves in the rebuttal for models with ReLUs (for example) it is hard to even formulate the idea of rank. Therefore I think forming conclusions about deep learning models in general raises questions. I think the same concern was also pointed out by other reviewers. Thank you for the rebuttal. The authors answered many of my questions which made me increase the score from 5 to 6.

[Author Response · NeurIPS 2020]

We thank reviewers for their time and effort!

## Reviewer 1

***Miscellaneous***   *(∗)* Thank you for the positive feedback! *(∗)* If one is willing to assume convergence of the unobserved entry, then our analysis admits a finite version. Namely, by Theorem 2 (*cf.* Subsection 3.4 and Appendix D), if the bottom right observation in the basic setting is increased from 0 to $\epsilon > 0$, then the limit of the unobserved entry is at least $\epsilon^{-1}$. *(∗)* We are not aware of a simple approach for adapting our analysis to symmetric matrix factorization; it is a direction we intend to investigate. *(∗)* We will modify the text to put more emphasis on matrix factorization.

## Reviewer 2

***Miscellaneous***   *(∗)* Thank you for the feedback and support! *(∗)* Extending our analysis to the regime of large learning rate is an interesting direction we intend to pursue. *(∗)* Our construction and theory indeed extend to higher dimensions — see Appendix C. *(∗)* In the high-dimensional analogue of Theorem 1 (not included in the paper) constants indeed depend on the dimension. *(∗)* We will modify the text to put more emphasis on matrix factorization.

## Reviewer 3

***Significance of our contribution***   The ability of norms to explain implicit regularization in matrix factorization is an important open question in the theory of deep learning (both supporting and opposing conjectures were made, with multiple recent works attempting to address them). Our main contribution — settling this open question (negatively) — does not follow from any prior work, in particular Dauber *et al.* (2020) or Suggala *et al.* (2018). These papers carefully construct highly specialized convex objectives on which gradient descent (or variants thereof) does not implicitly minimize certain norms. By this they refute the prospect of norms being implicitly minimized on **every** convex objective. To our knowledge, very few have endorsed this far-reaching prospect. Indeed, implicit regularization is conventionally viewed as stemming from a combination of optimizer and model (objective), not an optimizer on its own.

***Relation to prior work***   Prior work dealing with implicit regularization in matrix factorization and various other models is surveyed in Appendix B. We will add an account for "model-free" analyses such as the aforementioned.

***Miscellaneous***   *(∗)* Intuitively, what drives the unobserved entry towards infinity is the persistent positivity of the determinant — see proof sketch of Theorem 1. *(∗)* Definition of effective rank is provided in Appendix H.

## Reviewer 4

***Theoretical result for convergence to zero loss***   There seems to be a misunderstanding — Proposition 4 treats **scaled** identity initialization ($\alpha$ times identity, where $\alpha \in (0, 1]$ is allowed to be arbitrarily small), and thus does not deviate from the regime of near-zero initialization.

***Empirical support for convergence to zero loss***   In our experiments, the loss value $10^{-4}$ does **not** represent asymptotic convergence — we merely used it as a stopping criterion for maintaining reasonable run-times. Lower loss values are obtained if one is willing to accommodate longer runs. For example, top plot herein depicts a run with 50M iterations.

***Experimentation with additional settings***   We fear that portions of the paper may have gone unnoticed — beyond the basic setting defined in Subsection 3.1, various additional settings were treated, not only theoretically (Subsection 3.4, Appendixes C and D), but also empirically (Figure 4 in Appendix G). We will add more experimental figures to Appendix G, including ones obtained with different matrix dimensions (see example in bottom plot herein).

***Unobserved entry converging to finite value***   The fact that in some of our experiments the unobserved entry converges to finite value when theory predicts it should diverge to infinity, results from a non-perfect match between: *(i)* the theoretical scheme of gradient flow with balanced initialization; and *(ii)* its practical realization via gradient descent with small learning rate and near-zero initialization. (i) is a standard model for analyzing (ii), and is the subject of the formal conjectures on implicit regularization in matrix factorization. However, despite the fact that lower learning rate and smaller (or more balanced) initialization bring (ii) closer to (i) (as demonstrated in Figure 1), some degree of mismatch will always be present. As reviewer states, even when this leads unobserved entry to converge to finite value, a clear growth is always exhibited, in compliance with our theory, and in contrast to Conjecture 1.

***Extension to non-linear models***   We believe there is a misunderstanding — the whole point of our experiments with tensor factorization was to extend our conclusions beyond linear neural networks. Tensor factorization corresponds to a class of non-linear (polynomial) neural networks for which we can easily define the rank of input-output mappings, and accordingly examine whether that is implicitly minimized by gradient descent. As stated in the paper, in order to apply our conclusions to more conventional non-linear models (*e.g.* ReLU networks), formalizing a notion of rank for their input-output mappings is needed, and that may be a key step towards explaining generalization in deep learning.



[Meta-Review · NeurIPS 2020]

A strand of research that has emerged recently in theoretically understanding the success of deep learning suggests that implicit regularization due to the choice of optimization algorithms and other heuristics may play an important role. This paper dispels this notion by formally showing that in some problems (such as matrix completion using linear networks), implicit regularization drives all norms towards infinity. Authors also suggest that implicit regularization via minimization of rank may be more useful in terms of explaining generalization in deep learning. One of the concerns that reviewers expressed was that the theoretical results focused primarily on matrix factorization and learning linear networks. While the authors provide empirical evidence that their results may extend to nonlinear neural networks, the reviewers suggested that the paper’s positioning (and the title) would be more accurate if it were to focus on matrix problems rather than deep learning. The paper reads very well, and the results and insights in the paper are very compelling. Overall, a good paper. Accept!